# Drop Size Distribution Retrieval Using Dual-Polarization Radar Observations at C-Band and S-Band

Daniel Durbin[1], Yadong Wang[1], and Pao-Liang Chang[2]

[1]Electrical and Computer Engineering Department, Southern Illinois University Edwardsville, Illinois, USA
[2]Central Weather Bureau, Taipei, Taiwan

**Correspondence:** Daniel Durbin (ddurbin@siue.edu)

**Abstract.** Having knowledge of the drop size distribution (DSD) is of particular interest to researchers as it is widely applied to Quantitative Precipitation Estimation (QPE) methods. Polarimetric radar measurements have previously been utilized to derive DSD curve characteristics frequently modeled as a gamma distribution. Likewise, approaches using dual-frequency measurements have shown positive results. Both cases have relied on the need to constrain the relationship between the DSD parameters based on location or assumed weather conditions. This paper presents a methodology for retrieving the DSD parameters using the dual-frequency and polarimetric nature of measurements from a unique data set taken at co-located S-band and C-band dual-polarization radars. Using the reflectivity and differential phase measurements from each radar, an optimization routine employing particle swarm optimization (PSO) and T-Matrix computation of radar parameters is able to accurately retrieve the gamma distribution parameters without the constraints required in previous methods. Retrieved results are compared to known truth data collected using a network of OTT PARSIVEL disdrometers in Taiwan in order to assess the success of this procedure.

## 1   Introduction

Knowledge of the Drop Size Distribution (DSD) for a given location is an incredibly valuable piece of information. The most precise rainfall estimates can be achieved with an accurate assessment of the DSD. The applicability of deducing localized DSDs to Quantitative Precipitation Estimation (QPE) is evident. QPE provides society with many benefits. Being able to precisely identify or even forecast heavy precipitation can help emergency and water management services better deploy their resources and can alert others of dangerous flash floods, which can ultimately save lives.

The DSD provides critical information about the composition of a given volume of atmosphere in a region of interest. It is expressed as the number of particles for each drop size given as an equivalent sphere with a reference diameter, typically measured in millimeters. By knowing the DSD, important measures such as reflectivity ($Z$), rainfall rate, and total water content can be derived. These parameters are only as accurate as the DSD representation, which highlights the value of flexibility in the model.

The Marshall-Palmer distribution is a well-known model for describing the DSD of rain (Marshall and Palmer, 1948). It is based on an exponential distribution and assumes that the DSD is spatially and temporally homogeneous. The intercept parameter of the Marshall-Palmer distribution, which is set to 8000, determines the overall scale of the distribution.

However, research has shown that the Marshall-Palmer distribution is limited in its applicability to different types of precipitation and atmospheric conditions. Specifically, it has been found that the Marshall-Palmer distribution is only accurate in the stratiform precipitative region, which is a region of moderate, steady rainfall. As the rainfall rate increases and diverges from this mode of precipitation, the intercept factor exhibits a large degree of variability (Sauvageot and Lacaux, 1995). While it can be a useful approximation, it has limitations that should be taken into account when using it in applications such as QPE.

A larger number of DSDs can be expressed generically in the form of a gamma function given by Equation 1 where $N(D)$ (m$^{-3}$ mm$^{-1}$) is the concentration for each diameter $D$ (mm), and $\mu$ (unitless), $\Lambda$ (mm$^{-1}$), and $N_0$ are the shaping parameter, the exponential slope parameter, and the intercept parameter of the distribution, respectively (Ulbrich, 1983).

$$N(D) = N_0 D^\mu \exp(-\Lambda D) \tag{1}$$

While the gamma DSD provides more flexibility than the exponential or Marshall-Palmer distribution, its complexity can make it difficult to use for retrieval purposes, as three unknown parameters must be solved, and the model is still limited in its ability to account for the smallest drop sizes. In the past, researchers have attempted to simplify the process by assuming a relationship between these parameters.

Extensive research has been conducted to estimate DSDs, with many studies utilizing measurements taken at two frequencies. This is because using data from a single frequency is typically insufficient for accurately estimating the DSD and can only provide information about a single parameter at a specific location (Meneghini et al., 1997). Gorgucci and Baldini (2016) demonstrated this approach using Global Precipitation Measurement (GPM) dual-wavelength radar measurements. The surface reference technique is used to calculate path-integrated attenuation for each frequency so that a set of reflectivity and attenuation integral equations is formed. The system can then be solved for the DSD parameters that best match the measurements. The disadvantage of the approach is that it relies on assuming $\mu$ is a function of the median volume diameter (Gorgucci and Baldini, 2016).

Williams et al. (2014) also worked to show dual-wavelength approaches could be performed with GPM data. The under-constrainment issue was overcome by creating a distribution of standard deviations of the mean drop diameter using a large number of surface disdrometer measurements. Superior results were achieved relative to approaches that assume $\mu$ is a constant in order to overcome the ill-constrained nature of the retrieval (Williams et al., 2014). Many others have shown dual-frequency measurements are exploitable for discerning these properties (e.g., Mardiana et al., 2004; Chandrasekar et al., 2005; Eccles, 1979; Kozu et al., 1991; Kummerow et al., 1989; Marzoug and Amayenc, 1994; Meneghini et al., 1992).

In addition to leveraging the scattering variances across different frequencies, there have also been efforts to retrieve DSD parameters using dual-polarization measurements. These efforts capitalize on the characteristic oblateness of raindrops which would obviously not be available in the case of the previously mentioned vertical profiles measured by GPM. Brandes et al. used S-band measurements of reflectivity and differential reflectivity to seek the gamma parameters (Brandes et al., 2002). An

empirical relation is assumed between $\mu$ and $\Lambda$. While this relationship was applicable to both convective and stratiform rains, it was localized to the Florida region of the United States which was used for study. Similar approaches using horizontal and differential reflectivity have also been successful in experiments in Central Oklahoma but still rely on constrained relationships
of the parameters (Cao et al., 2010).

The aim of this present work is to solve for the three variables of the gamma drop size DSD. To achieve this, measurements taken at two frequencies with both vertical and horizontal polarizations are incorporated into optimization routines. This approach allows for the determination of DSD solutions that accurately represent the input reflectivity and specific differential phase information.

This paper is organized in the following manner. Section 2 conveys the methodology of this work. High-level design characteristics of the instruments used for data collection are provided, including the disdrometers and the two ground radars. Key elements of the data preprocessing routines are provided. Prior to concluding thoughts in Section 4, Section 3 demonstrates the results through multiple examples, comparing the disdrometer collected DSDs with the retrieved results from the proposed algorithm and an evaluation of the error when applied to QPE. A noticeable increase in accuracy is found when estimating the
rainfall rate using the proposed algorithm when compared to the traditional *Z-R* derived rainfall rate. Furthermore, we discuss and demonstrate a supplementary validation approach, utilizing C-band reflectivity data previously excluded from the proposed optimization routines. This approach serves as an external validation mechanism for the proposed algorithm and provides a comparison of the results obtained from the algorithm with those obtained from the C-band reflectivity data. The results demonstrate the effectiveness of the proposed algorithm in retrieving accurate DSDs and its potential usefulness in radar
systems that utilize more than one frequency.

## 2   Methods

### 2.1   Instrumentation

In the current work, the measurements of two co-located polarimetric radars, RCWF (S-band) and RCMD (C-band), are used in the algorithm development and validation. RCWF is an S-band dual-polarization radar that is part of Taiwan's operational
Multi-Radar-Multi-Sensor QPE system (Chang et al., 2021). To achieve a QPE accuracy within 10 percent, the reflectivity bias must be kept within 1 dBZ. This is accomplished by regularly calibrating the RCWF's reflectivity using a self-consistency algorithm (Le Loh et al., 2022). In contrast, RCMD is a C-band radar used experimentally and for research purposes rather than operationally.

A photo of these two radars is shown in Figure 1, where RCWF and RCMD are located on the right and left, respectively.
The Central Weather Bureau of Taiwan operates these two radars that provide real-time observations used for severe weather surveillance, quantitative precipitation estimation, hydrometeor classification, and precipitation microphysics studies. These radars play a critical role in monitoring and forecasting precipitation in Taiwan and provide key data that are useful in the research of preciptiative processes. The manufacturer/models of RCWF and RCMD are the Weather Surveillance Radar 1988 Doppler (WSR-88D) and Gematronik, respectively. More technical specifications about these two radars can be found in Table

1. As shown in Figure 1, these two radars are adjacent to each other, and the distance between them is much less than the range resolution (250 m). Since these two radars have the same polarization and location, the divergence of the observed differential phase ($\phi_{DP}$) and reflectivity ($Z$) can therefore be attributable to only the radar frequency difference. These measurement differences are the key variables utilized in the retrieval method.

To validate the performance of the proposed algorithm, DSD data collected by a laser-based optical system, the OTT Particle Size Velocity (PARSIVEL or Parsivel) disdrometer, is used as the ground truth. The Prasivel disdrometer derives the DSD through the measured particle's size and velocity. There are 32 diameter and velocity bins available to measure particles with size between 0.062 mm to 24.5 mm and with velocity between 0.05 m s$^{-1}$ to 20.8 m s$^{-1}$. The accuracy of the Parsivel disdrometer has been studied by Sheppard (2007); Jaffrain and Berne (2011).

Sampling with the OTT Parsivel may exhibit some degree of statistical variance, which is a common characteristic of all measurements. Microphysical events are inherently stochastic in nature and the physical sampling effects and noise also play roles in the deviations (Sheppard, 2007). The Parsivel used in this dataset has been compared to more accurate 2D disdrometer data in order to gauge the variance. The findings of Jaffrain et al. (2011) demonstrated that sampling uncertainty is minimal for small to moderate drop sizes but starts to escalate for larger classes (greater than 2.0 mm). Notably, the data set used in this study lacks a large representation of drop sizes exceeding 2 mm, which is advantageous as the smaller class measurements can be assumed to be more accurate(Jaffrain and Berne, 2011). Possible error sources of OTT Parsivel were investigated by Angulo-Martínez et al. (2018). It was found that uneven power distribution over the beamwidth or any time variation can adversely affect the accuracy. Other factors, such as the angle of the drop trajectory, coincidentally observed particles, and particles that intersect with only the edge will also lead to biases.

The locations of the two dual-polarization radars and OTT Parsivel disdrometers are depicted in Figure 2 with black and red markers, respectively. The complex terrain of Taiwan presents a challenge due to radar beam blockage. The Central Mountain Range (CMR) is visible in Figure 2, running from the north to the south of the island, with the highest peak exceeding 3800 m. To mitigate the effects of vertical variability in radar measurements, only data from the two lowest unblocked tilts are used in the DSD retrieval. Additionally, two rings with ranges of 25 km and 70 km are shown in Figure 2. The performance of the proposed approach was validated using only the disdrometers located within these two ranges. More details related to the validation data selection are provided in the following section.

The final sources of error we attempted to mitigate relate to the radar measurement itself. Observation bias is a primary concern, stemming from the inherent differences between radar and disdrometer measurements—where radars capture data over a large volume and disrometers provide point-specific observations leading to potential discrepancies in data interpretation. Additionally, vertical variations of DSDs pose a significant challenge due to the radar's observation volume being several hundred meters above the ground. This discrepancy is particularly problematic for all radar-based DSD retrieval methods. The various equipment heights and locations are shown in Table 2. To address this issue, we constrained our analysis to data from the two lowest elevation tilts of the radar scans. Although it does not fully eliminate the issue, this approach helps to reduce the impact of this error source. These aspects underscore the complexities involved in accurately retrieving DSDs and highlight the necessity of carefully considering these factors when analyzing radar and disdrometer data.

## 2.2 Radar and Disdrometer Data

### 2.2.1 Preprocessing of Radar and Disdrometer Data

The $Z$ and $\phi_{DP}$ fields from both C-band and S-band radars are the proposed parameters for DSD retrieval, and the qualities of both fields are examined and processed through a set of quality control procedures. The quality control process pertaining to the reflectivity field includes identifying and removing nonprecipitation radar echoes and smoothing along the radial direction. Any gates associated with correlation coefficient ($\rho_{HV}$) less than 0.98 are considered as possibly produced by nonprecipitative clutter, and reflectivity is excluded from the subsequent average operation. The obtained reflectivity is smoothed with a 4 km smoothing window along each radial direction. The raw $\phi_{DP}$ field of RCMD is processed with the new Selex–Gematronik family of digital receiver and signal processor (GDRX) (Bringi et al., 2005). The GDRX processes the raw field using the field unwrapping, "good data" mask application, and finite impulse response (FIR) filtering. The details of the procedure can be found in Bringi et al. (2005). A similar procedure is also applied on the raw $\phi_{DP}$ field from RCWF. Examples of processed reflectivity and differential phase fields are shown in Figure 3, where panels "a" ("c") and "b" ("d") show the reflectivity (raw differential phase) measured by S-band and C-band, respectively. The yellow arrow in panel "a" indicates the radial path from the radars to the disdrometer location of interest. It should be noted the reflectivity fields from both frequencies are consistent, and the difference is mainly caused by the attenuation. The differential phase fields, on the other hand, show significant difference, which indicates that the differential phase is more sensitive to the radar frequency. A detailed analysis of the impact from frequency on the differential phase and specific differential phase is presented in Section 3.1.

This figure serves as a visual representation of the data used in the DSD retrieval algorithm and demonstrates the differences in the data between the S-band and C-band radars. Figure 4 shows the pre-processed (dashed) and post-processed $Z$ (solid), without any attenuation correction, and $\phi_{DP}$ fields along the yellow arrow of Figure 3, contrasting the difference between the post-processed differential phase profiles (solid) with the profiles prior to unfolding and smoothing (dashed). The measurements from S-band and C-band are shown with blue and red color, respectively.

The DSD parameters are derived from the Parsivel disdrometer observations through the approach proposed by Raupach and Berne (Raupach and Berne, 2015). In this approach, data concerning individual raindrops, including their diameters, fall velocities, and the effective sampling areas of the instrument are recorded. The drops are binned into diameter classes, and the concentration is then calculated using Equation 2 where $N_i$ is the drop concentration for the $i$th equivolume diameter class, $S$ is the effective sampling area, $V$ is the particle velocity, and $\Delta D_i$ and $\Delta t$ are the class width and sampling period, respectively.

$$N_i = \frac{1}{\Delta D_i \Delta t} \sum_{j=1}^{M} \frac{1}{S_j V_j} \qquad (2)$$

### 2.2.2 Data selection

The algorithm development and validation were based on ten days of data collected by RCMD and RCWF. These days included June 1, 2017, June 11-17, 2017, January 7, 2018, and May 7, 2018. The majority of these days correspond to the Meiyu season

in Taiwan and represent events of light to moderate precipitation which were observed in the region covered by the two radars. These days provided data representative of multiple precipitation intensities.

One challenge in using both RCWF and RCMD is that they operate under different volume coverage patterns (VCPs). As a result, there can be a slight time lag between scans from these two radars when observing the same location. To minimize the retrieval biases caused by the DSD variation during the time lag, the time stamp differences between scans from the two radars were limited to within one minute. This time limitation helped ensure that the radar data adhere to an acceptable degree of synchronization.

In this work, the differences in both reflectivity and differential phase fields obtained from the S-band and C-band radars play critical roles in the DSD retrieval. Sufficient differences are expected from the two frequencies for differential phase. If the differences are too small, the algorithm may result in a biased or inaccurate retrieval of the DSD. Thus, it is crucial to carefully select the observation range such that enough differential phase difference has been experienced by the radar return.

Another important factor to consider is that biases in the retrieved DSD can accumulate along the range. This means that the farther the distance between the radar and the target area, the larger the potential error in the retrieved DSD. The presence of atmospheric attenuation is also a predictable issue in radar data processing, and it also underscores the importance of carefully selecting the range of interest when estimating the DSD.

To achieve reasonable results, the following criteria for candidate data are therefore used for the terminal gate of the retrieval which contains the disdrometer:

- 25 km < Range < 70 km

- $Z^S > 25$ dBZ

This set of criteria is designed to strike a balance between creating sufficient deviation between C and S-band differential phases for an accurate DSD retrieval, while also preventing excessive error accumulation as well as minimizing the vertical separation between the radar observation volume and ground location. Additionally, a reflectivity threshold of 25 dBZ is imposed to ensure that there is enough observable precipitation in the terminal gate where the disdrometer is located. These criteria are primarily chosen to increase the quality of the data for development and validation, however similar standards would need imposed if the algorithm were to be operationally applied to address the error accumulation and elevation differences.

## 2.3   Drop Size Distribution With an Artificial Intelligence Method

The flowchart of the proposed DSD retrieval algorithm is presented in Figure 5. Three variables: S-band reflectivity ($Z^S(r,\theta)$) and differential phase from both S- and C-band ($\phi_{DP}^S(r,\theta)$, $\phi_{DP}^S(r,\theta)$) are implemented as inputs, where $r$ and $\theta$ are the co-ordinate of range and azimuthal angle, respectively. The analysis exclusively employs S-band reflectivity as it is considered a more dependable variable than $Z^C$ in the dataset since RCWF is utilized operationally and is well calibrated. Although S-band reflectivity does experience atmospheric attenuation, $Z^C$ is much more vulnerable to this effect and is therefore not used as an input to the algorithm. Since $Z^C$ displays high correlation with $Z^S$, it is unlikely that the inclusion of C-band reflectivity

would furnish any additional useful insights into the retrieval. Moreover, the exclusion of $Z^C$ from the process serves as an additional validation parameter, as discussed further in Section 3.

The radar variables from a given gate are first preprocessed with the routine described in Section 2.2.1, and the processed data are then used to retrieve three parameters as described in Section 2.3.2.

### 2.3.1 T-Matrix Computation of Radar Parameters

In the retrieval procedure, a set of DSD parameters (Equation 1) is first initialized within commonly observed ranges of the parameters (Zhang et al., 2001):

- $10^2 < N_0 < 10^{10}$

- $0 < \mu < 10$

- $0 < \Lambda < 15$

With the initial parameters, the DSD ($N(D)$) is calculated, and radar variables ($Z^S$, $K_{DP}^S$, $K_{DP}^C$, and specific attenuation, $A$) are then calculated with the following equations integrated from 0 to a maximum diameter ($D_{\mathrm{max}}$) of 8 mm (Ryzhkov et al., 2013):

$$Z = \frac{4\lambda^4}{\pi^4 |K_w|^2} \int_0^{D_{\mathrm{max}}} |f_b^\pi(D)|^2 N(D)dD \tag{3}$$

$$K_{DP} = \frac{0.18\lambda}{\pi} \int_0^{D_{\mathrm{max}}} \mathrm{Re}\{f_b^0(D) - f_a^0(D)\}N(D)dD \tag{4}$$

$$A = 8.686 \times 10^{-3}\lambda \int_0^{D_{\mathrm{max}}} \mathrm{Im}\{f_b^0(D)\}N(D)dD \tag{5}$$

In these equations, vertical polarization scattering amplitudes, $f_a^0(D)$ ($f_a^\pi(D)$), and horizontal polarization scattering amplitudes, $f_b^0(D)$ ($f_b^\pi(D)$), are calculated with the T-Matrix method (Waterman, 1965), where 0 and $\pi$ indicate forward and backward directions respectively. These formulations make use of a simplification with zero canting angle which will introduce only a small error due to the limitation of elevation tilts. The dielectric constant of water is referenced at 10 degrees Celsius which is a reasonable temperature for the radar volume situated between the melting layer and the ground level temperature of Taiwan during these dates.

With the obtained $K_{DP}$ field, the $\phi_{DP}$ field from both the S-band and C-band radars were then calculated through Equation 6, where $\Delta R$ represents the range difference between the $i$th gate and the previous gate. $\phi_{DP}^{sys}$ is the system $\phi_{DP}$ be found in Table 1 for both radars.

$$\phi_{DP}(r_i,\theta) = 2 \cdot K_{DP}\Delta R + \phi_{DP}(r_{i-1},\theta) + \phi_{DP}^{sys} \tag{6}$$

## 2.3.2 Optimization Routine

The estimation of radar variables can be achieved by adjusting a parameterized DSD in order to reduce the difference between the estimated and observed values. As this difference decreases, the optimization problem gradually approaches a minimum value, ultimately resulting in the retrieval of the desired information. Essentially, the retrieval problem can be considered as an optimization problem where the goal is to find the optimal set of parameters that minimize the difference between the estimated and observed values.

Multiple methods exist for minimizing the error between the simulated radar variables and the measured variables. A relatively simple approach that was first used is the Gauss-Newton method. While it can quickly converge to a solution, the technique will often only find local minima rather than the global minimum of the solution space. Another early attempt used the genetic algorithm (GA), which very reliably found better solutions. The GA, however, was very computationally intensive and relied on fine-tuning of the crossover and mutation factors to efficiently solve for the DSD.

Particle Swarm Optimization (PSO) was ultimately used for this work since it is comparatively more efficient at seeking the most representative DSD. Figure 6 shows the organization of the PSO application. For a gate with preprocessed reflectivity and differential phase measurements, particles with random $N_0$, $\mu$, and $\Lambda$ are initialized. Ranges for the particle positions are chosen according to commonly observed intervals (Zhang et al., 2001). In this manner, a coordinate space is created in which each DSD parameter is a dimension.

The three coordinates of each particle position collectively define a DSD that is used to calculate $Z^S$ and $\phi_{DP}^{S,C}$. Various cost functions which measure the distance from the truth data were tried. The fitness function given by Equation 7 allows for relative weightings to be applied to each variable. Experimentally determined values ($\alpha = 5$, $\beta = \gamma = 1$) led to reliable retrievals. It should also be noted that $Z^S$ is expressed as a logarithmic value rather than in linear units.

$$\text{Cost} = \alpha \left| \frac{Z_{\text{simulated}}^S - Z_{\text{measured}}^S}{Z_{\text{measured}}^S} \right| + \beta \left| \frac{\phi_{DP,\text{ simulated}}^S - \phi_{DP,\text{ measured}}^S}{\phi_{DP,\text{ measured}}^S} \right| + \gamma \left| \frac{\phi_{DP,\text{ simulated}}^C - \phi_{DP,\text{ measured}}^C}{\phi_{DP,\text{ measured}}^C} \right| \tag{7}$$

For a gate's retrieval, the cost of every particle is calculated, and the iteration's current best solution as well as the global best solution of all iterations are recorded. The particle positions are then updated according to Equation 8 where $\epsilon$ is the local acceleration factor, $\zeta$ is the global acceleration factor, $r_1$ and $r_2$ are random numbers between zero and one that are generated for each particle, and $i$ denotes the current iteration. This allows the particles to move towards the current and global best solutions and possibly find better solutions along the path. The convergence speed must be weighed against the possibility of "over-shooting" viable candidate solutions. $\epsilon$ and $\zeta$ were determined experimentally, and values of 0.15 and 0.0015 were used for processing the overall data set.

$$\begin{bmatrix} N_0 \\ \mu \\ \Lambda \end{bmatrix}_{i+1} = \begin{bmatrix} N_0 \\ \mu \\ \Lambda \end{bmatrix}_{i} + \epsilon \left\{ \begin{bmatrix} N_0 \\ \mu \\ \Lambda \end{bmatrix}_{\text{iteration best}} - \begin{bmatrix} N_0 \\ \mu \\ \Lambda \end{bmatrix}_{i} \right\} r_1 + \zeta \left\{ \begin{bmatrix} N_0 \\ \mu \\ \Lambda \end{bmatrix}_{\text{global best}} - \begin{bmatrix} N_0 \\ \mu \\ \Lambda \end{bmatrix}_{i} \right\} r_2 \tag{8}$$

The swarm consists of five thousand particles which are allowed four hundred iterations for each retrieval. It should be noted that the setting of four hundred iterations is intended solely for prototype algorithm development, and computational efficiency has not yet been addressed. A simple iteration control algorithm could be implemented to terminate the computation once the root mean square error reaches the predefined threshold. Any embedded solution should take advantage of speed improvements gained from parameter tuning or even substituting a more efficient technique in place of PSO.

### 2.3.3 Retrieval Along Radial

The goal of the retrieval process is to obtain a DSD similar to the disdrometer measurement. To achieve this, the PSO retrieval algorithm is applied iteratively starting from the gate nearest the radar. Initially, the first gate is assumed unattenuated, and its measured reflectivity and differential phase are directly input into the PSO algorithm. After completing the retrieval for a gate, attenuation is calculated using Equation 5, and the reflectivity for the next farthest gate is adjusted. In this manner, each gate's input reflectivity accounts for attenuation experienced between the radar and that gate. This process, illustrated in Figure 7, continues at 4 km intervals until reaching the disdrometer location.

### 2.4 Simulation

The role of the additional frequency in providing extra information is one of the key questions addressed in this study. We demonstrate the viability of employing dual-polarization measurements at C-band and S-band for rainfall estimation through simulations. The algorithm was evaluated under ideal conditions by extracting disdrometer-recorded distributions and fitting each to a DSD with gamma parameters as discussed in Section 2.3.1. For each distribution, $Z$ and $K_{DP}$ values were calculated using Equations 3 and 4. A total of 700 distributions were generated.

These calculated $Z$ and $K_{DP}$ values were treated as true observations, devoid of confounding factors such as elevation differences, noise, or unaccounted attenuation discussed in Section 2.1. Rainfall rates were calculated using the fitted DSDs according to Equation 9 with $v(D)$ given by Equation 10 where $D$ is given in mm and $v(D)$ is in mm/s (Ulbrich, 1983). These rates are considered true if perfectly observed.

$$R = \frac{\pi}{6} \int\limits_0^{D_{\max}} D^3 N(D) v(D) dD \tag{9}$$

$$v(D) = 9650 - 10300 \exp(-0.6D) \tag{10}$$

The cost function, defined in Equation 7, was modified to operate directly on specific differential phase rather than differential phase, according to Equation 11 with simplified weights: $\alpha = \beta = \gamma = 1$.

$$\text{Cost} = \alpha \left| \frac{Z^S_{\text{retrieval}} - Z^S_{\text{true}}}{Z^S_{\text{true}}} \right| + \beta \left| \frac{K^S_{DP,\,\text{retrieval}} - K^S_{DP,\,\text{true}}}{K^S_{DP,\,\text{true}}} \right| + \gamma \left| \frac{K^C_{DP,\,\text{retrieval}} - K^C_{DP,\,\text{true}}}{K^C_{DP,\,\text{true}}} \right| \tag{11}$$

The retrieval process was conducted using this study's dual-frequency approach on a single gate. For comparison, retrievals were also performed using a single frequency by setting $\gamma$ to zero, thus discarding the C-band contribution. Additionally, retrievals incorporated a $\mu - \Lambda$ constraint (Equation 12, which Seela et al. demonstrated to be accurate for Taiwan rain systems during summer months) (Seela et al., 2018). Rain rates for each retrieval method were calculated and compared to true values
using relative absolute error, defined in Equation 13, where $R_d$ represents the truth rain rate, and $R$ denotes the rainfall rates from the methods being compared.

$$\Lambda = 0.0235\mu^2 + 0.472\mu + 2.394 \tag{12}$$

$$\text{RAE} = \frac{|R_d - R|}{R_d} \tag{13}$$

As a final comparative benchmark, rain rates were also calculated using varying *Z-R* relationships. Equation 14 represents
perhaps the most common meteorological radar relationship Ulbrich and Lee (1999), Equation 15 is specific to Taiwan as proposed by Chang et al. (Chang et al., 2021), and Equation 16 is derived from fitting calculated $Z$ and $R$ values found in the initial steps of the simulation.

$$Z = 300R^{1.4} \tag{14}$$

$$Z = 207R^{1.45} \tag{15}$$

$$Z = 324R^{1.35} \tag{16}$$

Figure 8 illustrates the cumulative distribution of errors for each method. In this plot, the cumulative portion of errors is shown on the y-axis, while the sorted error values of each method are displayed on the x-axis. As an example interpretation of

the plot, 90% of current study's errors (blue) are less than 0.2 in terms of RAE, while 90% of the Tawain $Z - R$ based errors (black dashed) are not contained until an error level of 0.78 RAE. Interpreting the plot from the constant RAE perspective, 65% of the current study's errors are below 0.1, while 60% of $\mu - \Lambda$ (purple) method's errors and 30% of the Tawain $Z - R$ (black) method's errors are below 0.1 RAE. The conclusion of the simulations can be drawn from the plots: the dual-frequency approach provides a modest improvement over single frequency retrievals, even with a relevant $\mu - \Lambda$ constraint, and a significant improvement compared to all *Z-R* based rates. Under these ideal simulated conditions, the median RAE of the dual-frequency approach was 0.0623 while the $\mu - \Lambda$ single-frequency approach was 0.0725. The median RAE of the best performing (at the 50th percentile) *Z-R* relationship ($Z = 207R^{1.45}$), was 0.1861.

At the parameter level, the benefit of an additional frequency can also be investigated while still using the gamma model assumption. Distributions with varying $\mu$ and $\Lambda$ values, while keeping $N_0$ constant, can be used to calculate specific differential phase values at two wavelengths (10.0 cm for S-band and 5.0 cm for C-band). These values are depicted in Figure 9. This simulation highlights that specific differential phase values exhibit distinct separations at the two wavelengths. This simulation underscores the potential of using dual-polarization measurements at C and S radar bands for DSD retrieval.

## 3    Performance Evaluation

The evaluation of the proposed algorithm was carried out through both qualitative and quantitative methods. Initially, the retrieved DSDs were subjectively compared to the DSDs measured by the disdrometers at the times nearest to the radar scans. This qualitative comparison provides a preliminary measure of the algorithm's overall accuracy as adjustments were made to its parameters. A subsequent discussion of the role of using two frequencies rather than a single frequency approach highlights the value of using the C-band reflectivity (unused in the retrieval) for further validation of the along-radial results. Following the fine-tuning of the cost function and PSO parameters, a more rigorous quantitative analysis was performed. This involved calculating the rainfall rates based on the disdrometer-recorded DSDs, which were considered as the benchmark, and comparing these rates to those obtained from the retrieved DSDs. Additionally, the calculated rainfall rates were contrasted with those derived using the traditional *Z-R* relationship methodology, offering a comprehensive evaluation of the algorithm's performance.

For the performance validation of this study, data spanning ten days from 2017 and 2018 were utilized. A potential case is defined as one plan position indicator (PPI) scan of each radar. The stringent criteria for time synchronization narrowed the dataset to 167 cases that not only met the synchronization requirements of the radar scans but also had the requisite data quality and fell within the disdrometer range requirement.

### 3.1    Qualitative Assessment

Figure 10 showcases a representative selection of the retrieved DSDs the algorithm generated as compared to the disdrometer data. Cases were systematically sampled across the dataset's collection period to accurately reflect the algorithm's retrieval performance. These cases strictly adhere to the time synchronization criteria established by our data quality standards and

exhibit reflectivity values at the terminal gate that exceed the minimum threshold. The radar scan time tags are specified with precision down to the second, whereas disdrometer measurements are recorded on a per-minute basis. Consequently, both the preceding and subsequent disdrometer timestamps are presented (red and green, respectively), as well as the closest time (blue). This approach acknowledges instances where these adjacent timestamps may more accurately represent the radar data.

Six of these eight cases show a high degree of agreement across the spectrum of drop sizes, with particularly strong corre-
lation observed for the measurements of moderate drop sizes. It can be found from Figure 10 that the retrieved DSD fits the disdromter observations very well for larger drops ($D > 0.5$ mm). Deviations can be found in cases such as 20170613 - 08:04:49 and 20180107 - 09:26:32 for smaller drops ($D < 0.5$ mm). This is a predictable result given the retrieval input parameters are heavily dominated by contributions of larger diameters. Not all disdrometer sizes need to be equally prioritized for accurate fitting. Examination of Equations 3 and 4 reveals that the contribution to radar parameters from each diameter increases with
the size of the drop. Furthermore, previous research evaluating the accuracy of disdrometers across various drop sizes indicated that drops of 0.6 mm and larger are the first reliably measured sizes by optical disdrometers (Tokay et al., 2001). This finding supports the notion that inaccuracies in measuring smaller drop sizes do not significantly impact the calculations of reflectivity or DSD-derived metrics such as rain rate and attenuation. While some researchers have proposed that the goal of a retrieval should be to fit the predominance of the size spectrum (Adirosi et al., 2013), it is clearly more important to represent medium
to large diameters when the integrated parameters are the focus.

To ensure the accuracy of the retrieval algorithm, one can utilize the C-band parameters obtained from the retrieved DSDs to validate the S-band parameters. To accomplish this, the attenuation that is calculated from the retrieved DSDs can be applied to correct the C-band reflectivities affected by the large attenuation experienced over the path. Figure 11 suggests that the S-band reflectivity is less affected by attenuation as expected. The blue dashed lines in the plots represent the raw reflectivity values
measured by the S-band radar, which demonstrates the relatively low attenuation at this wavelength. The blue solid line in the plot represents the attenuation-corrected reflectivity values measured by the S-band radar. The corrected C-band reflectivities are shown as the solid red line in each plot and are seen to match well with the S-band values. This indicates that the attenuation factor derived from the retrieved DSD is effective in converting the C-band reflectivities to equivalent S-band values. By using this correction factor, we can confirm that the retrieved DSD accurately predicts the atmospheric effects at both radar bands,
and thus the corrected C-band reflectivity serves as a "sanity check" for the along-radial retrieval. The input (dashed) and retrieved (solid) differential phase profiles are also provided for each example.

### 3.2  Quantitative Assessment

A comprehensive quantitative analysis was conducted on the 167 cases, utilizing the disdrometers in Table 2, with a particular focus on the algorithm's accuracy in QPE, where the importance of accuracy is most prominently showcased.
At the parameter level, Figure 12 demonstrates the retrieval routine produces DSDs corresponding to radar parameters which effectively match the inputs. The x-axis represents the inputs to the retrieval based on measurements, taking into account at-tenuation in the case of $Z$. The y-axis represents the parameters output by the algorithm retrievals for the final gate containing the disdrometer. Metrics for the retrieval pairs are indicated for each parameter: the mean bias (MB), root-mean-square error

(RMSE), and the correlation coefficient (CC). These are defined as follows: the mean bias $\text{MB} = \frac{\langle X_{\text{Retrieved}} \rangle}{\langle X_{\text{Observed}} \rangle}$, the root-mean-square error $\text{RMSE} = \left\langle (X_{\text{Retrieved}} - X_{\text{Observed}})^2 \right\rangle^{1/2}$, and the correlation coefficient $\text{CC} = \frac{\langle (X_{\text{Retrieved}} - \langle X_{\text{Retrieved}} \rangle)(X_{\text{Observed}} - \langle X_{\text{Observed}} \rangle) \rangle}{\sigma_{X_{\text{Retrieved}}} \sigma_{X_{\text{Observed}}}}$, where angle brackets indicate the mean, $X_{\text{Retrieved}}$ is the algorithm output for the given parameter ($Z^S$, $K_{DP}^S$, $K_{DP}^C$), $X_{\text{Observed}}$ is the observed input value for the parameter, and $\sigma$ is the standard deviation of the parameter.

In the context of QPE, rainfall rates were calculated using the retrieved DSDs according to Equation 9 with $v(D)$ given by Equation 10 as in Section 2.4. For comparison, the S-band reflectivity data was applied using the Equation 15, the region-specific relationship also referenced in Section 2.4 (Chang et al., 2021). The Parsivel data was used to calculate the ground truth via Equation 17 where $D_j$ is the equivolume diameter of the $j$th recorded drop, $S$ is the effective sampling area, and $\Delta t$ is the sampling period (Raupach and Berne, 2015).

$$R_d = \frac{6\pi \times 10^4}{\Delta t} \sum_{j=1}^{M} \frac{D_j^3}{S_j^2} \tag{17}$$

The evaluation involved comparing the rainfall rates derived from the *Z-R* relationship, that of a $\mu - \Lambda$ constrained (Equation 12) S-band only retrieval, and that of this study's retrieval algorithm against the ground truth values from the Parsivel disdrometer, whose biases were discussed in Section 2. This comparison was based on RAE following the same approach as discussed in Section 2.4.

Figure 13 features a plot of the cumulative distribution of RAE for all three methods. The retrieval algorithm's accuracy is depicted by the blue-circles line, while the $\mu - \Lambda$ constrained retrieval is shown in purple, and the *Z-R* relationship benchmark is indicated with the dashed black line. This visualization highlights that the proposed method of estimating rainfall rates using retrieved DSD parameters significantly enhances accuracy over a region-specific *Z-R* relationship (Chang et al., 2021). Specifically, the median RAE for the *Z-R* method is 0.76, while the retrieval results correspond to a median of 0.53, marking a notable improvement of 30.3 % in this study's context. The dual-frequency approach is comparable to the single-frequency constrained approach and the only claimed relative benefits are not needing to ascertain a regional $\mu - \Lambda$ relationship or cases in which the precipitation deviates from the assumed relationship.

It is important to acknowledge a notable limitation observed in the performance of the retrieval algorithm, as reflected in the resulting cases with large RAE. Specifically, the algorithm has the potential to significantly overestimate the rainfall rate. Closer examination reveals that such discrepancies arise from ill-posed conditions for retrieval. Although incorporating multiple frequencies enhances the capability to retrieve the DSD, the specific differential phase and reflectivity values must still exhibit a consistent correlation. Deviations from this correlation can mislead the algorithm, yielding suboptimal outcomes. Therefore, incorporating self-consistency relations between the input variables stands out as a promising direction for enhancing algorithmic accuracy in future research endeavors. (Park et al., 2005; Giangrande et al., 2013; Gorgucci et al., 1999; Reinoso-Rondinel et al., 2018)

## 4 Conclusions

A novel approach to retrieving the DSD using PSO has been discussed. The outlined dual-frequency dual-polarization method can provide significantly improved estimates for rainfall compared to $Z - R$ relationship estimates and offers modest improvement to retrievals utilizing relevant $\mu - \Lambda$ constraints. While the retrieval is unconstrained regarding the gamma distribution parameters, the price is the additional data needed. The authors predict that radar systems utilizing more than one frequency will continue to become more commonplace and producing the required synchronized measurements will be more feasible with the adoption of phased array systems. The value of dual-polarization in radars is already universally accepted. Algorithms such as the one prototyped in this work will become more valuable as radar systems produce data with this type of increased diversity.

There are several limitations in this research which have been briefly mentioned and which future work should address. A larger dataset or alternative preprocessing criteria should be assessed for long-term evaluation of the approach. While ten days of data were screened, the selection process for useful cases was highly discriminatory. Only measurements from the two radars which were synchronized within one minute became candidates. Of these, only data with corresponding disdrometer truth data and minimum terminal gate reflectivity were included in the processing pool. Preprocessing the measurements led to the greatest reduction in potential data. The majority of these cases were excluded because of data quality primarily due to unreliable differential phase profiles. Future work should exhaustively assess the approach by processing a larger amount of high quality data.

One focus of this project was to show that $N_0$, $\mu$, and $\Lambda$ are independently solvable when incorporating both multi-frequency and multi-polarization information. Restricted relationships between the variables are still highly useful. If the intended application of any retrieval algorithm can utilize a constraint with high confidence, the DSD retrieval process becomes more efficient. The tradeoff is that shaping characteristics may not be captured, which is why study of more flexible retrieval processes should continue.

The optimization approach involves various adjustable parameters, including swarm size, number of iterations, and acceleration coefficients. Fine-tuning these parameters could result in faster and more optimal results. Moreover, adapting the algorithm to function as an embedded application for field testing is also a promising area for further development.

*Code and data availability.*

The datasets and source code used in this study are available from the corresponding author upon request.

*Author contributions.*

Dr. Yadong Wang and Daniel Durbin originated the initial concept of utilizing dual-frequency, dual-polarization measurements for DSD retrieval and developed preliminary algorithms to demonstrate its feasibility. Daniel Durbin further applied the

PSO technique, processed the data set, and prepared the drafts of this paper. Dr. Yadong Wang and Daniel Durbin collaborated on the development of the final algorithm. Dr. P.-L. Chang provided and processed radar data from the Central Weather Bureau (CWB) and was further involved in algorithm discussion and article authoring. The CWB provided the unique data used in the project.

*Competing interests.*

The authors declare that there is no conflict of interest.

*Acknowledgements.* The authors thank the radar engineers at CWB for helping collect and process the radar data used in the study.

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

| Radars | | |
|---|---|---|
| | RCWF | RCMD |
| Model | WSR-88D | Meteor 1700C |
| Peak Power | 700 KW | 250 KW |
| Operating Band | S-band | C-band |
| Wavelength | 10.5 cm | 5.3 cm |
| Longitude | 121.78° E | 121.78° E |
| Latitude | 25.07° N | 25.07° N |
| Beamwidth | 0.93° | 0.90° |
| Range Resolution | 250 m | 100 to 500 m |
| Minimum System PhiDP | 60° | 10° |
| PRF | 320-1300 Hz | 250-2000 Hz |
| VCP | 221 | 82 |

**Table 1.** Technical specifications of two polarimetric radars (RCWF and RCMD) used in the current work.

| Location/Station | Latitude | Longitude | Height (m) | Range (km) |
|---|---|---|---|---|
| RCWF/RCMD | 25.07 | 121.78 | 743 | N/A |
| 466930 | 25.17 | 121.54 | 659 | 26.8 |
| 466920 | 25.04 | 121.51 | 8 | 27.9 |
| 466910 | 25.19 | 121.52 | 765 | 29.2 |
| 466880 | 25.00 | 121.43 | 15 | 35.9 |
| 466950 | 25.63 | 122.07 | 5 | 68.6 |

**Table 2.** Equipment Location and height above mean sea level. Range is calculated as the haversine distance.

| PARSIVEL Disdrometer | |
| --- | --- |
| Manufacturer | OTT HydroMet |
| Sampling Area | 50 cm$^2$ |
| Drop Size Range | 0.06-24.5 mm |
| Velocity Range | 0.05-20.8 m/s |

**Table 3.** Disdrometer Characteristics

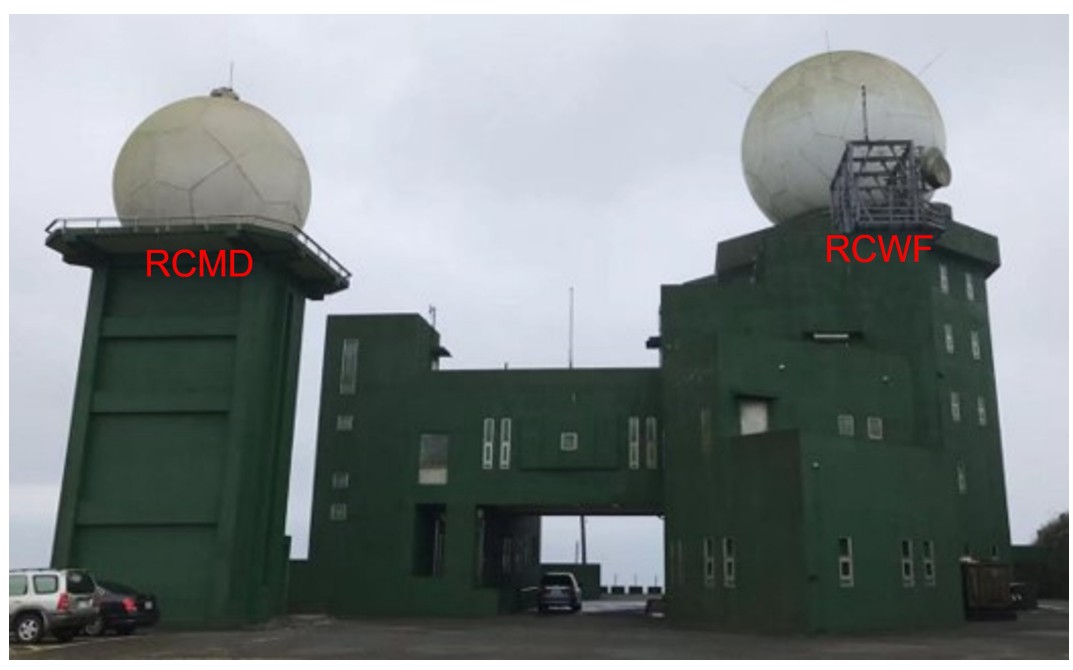

**Figure 1.** RCMD (C-band) left and RCWF (S-band) right at the Wufenshan weather station

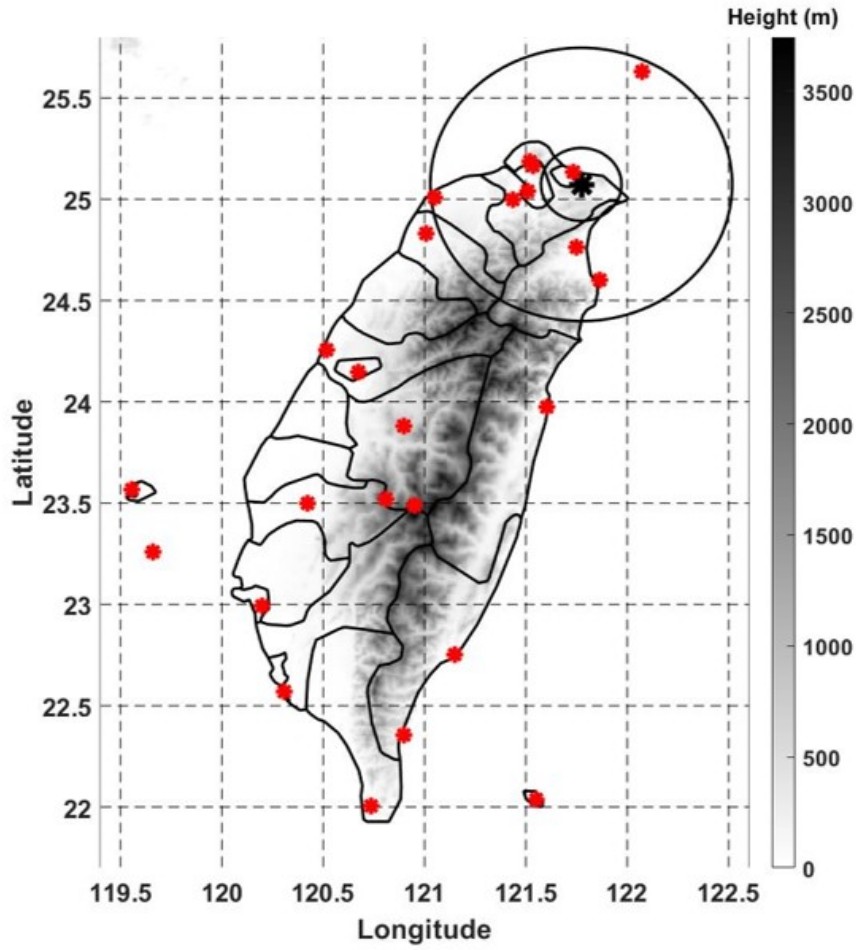

**Figure 2.** Taiwan - Instrument Locations. Radii (25 km and 70 km) are drawn around the radar location. Measurement stations with disdrometers are shown in red. Terrain height is indicated in grayscale throughout the map for reference.

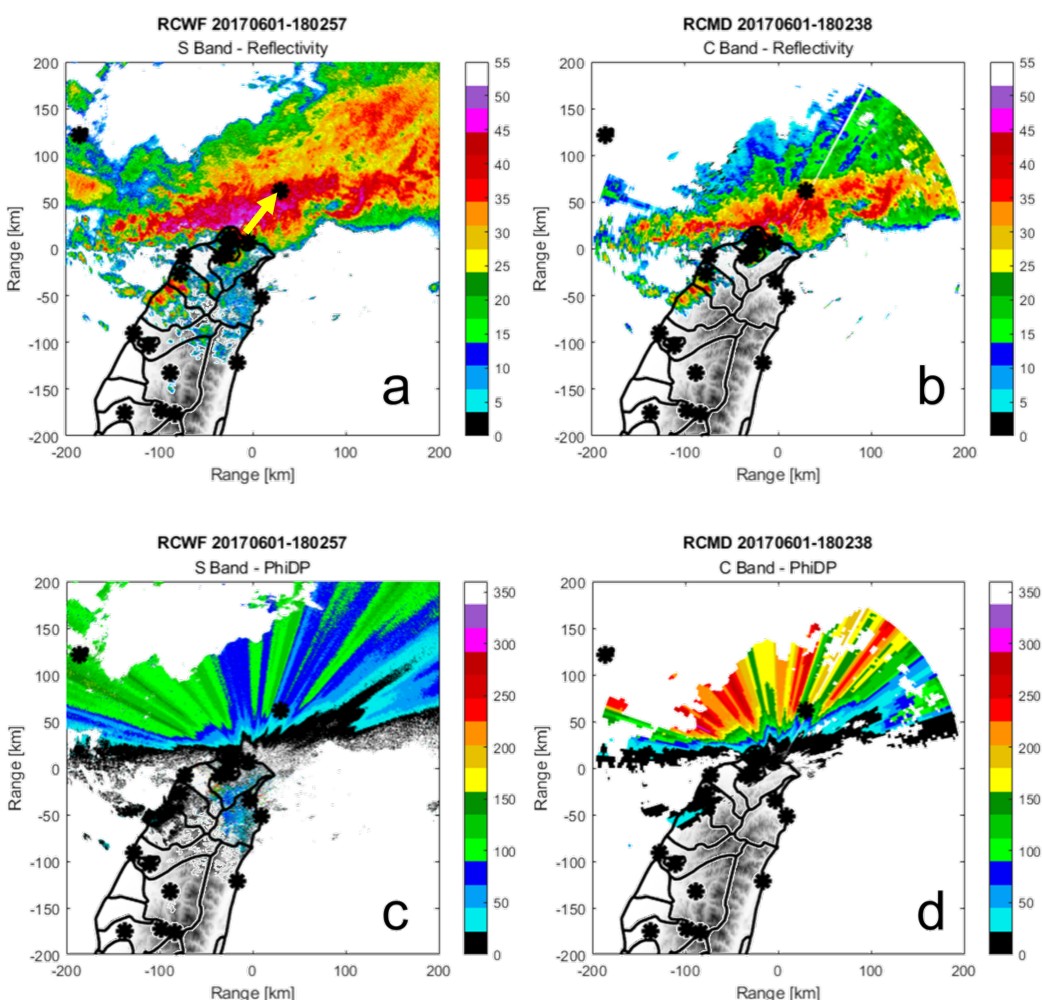

**Figure 3.** Panels a and b show the reflectivity measured in S-band and C-band, respectively. The yellow arrow in panel a indicates the radial path from the radars to the disdrometer location of interest. Panels c and d contain the raw differential phase measurements recorded in S-band and C-band.

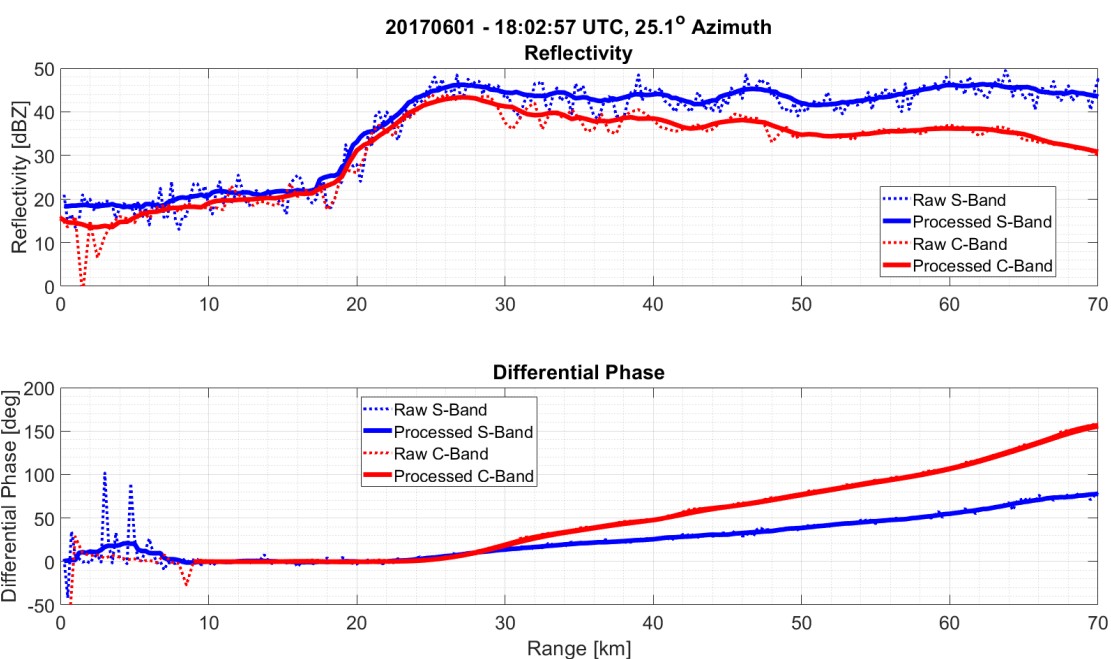

**Figure 4.** The data along the radial indicated in Figure 3 before and after processing. Raw radar data (dashed lines) are contrasted with post-processed data (solid lines). Processed C-band reflectivity is not used as a retrieval input but is useful for validation purposes as discussed in Section 3.1

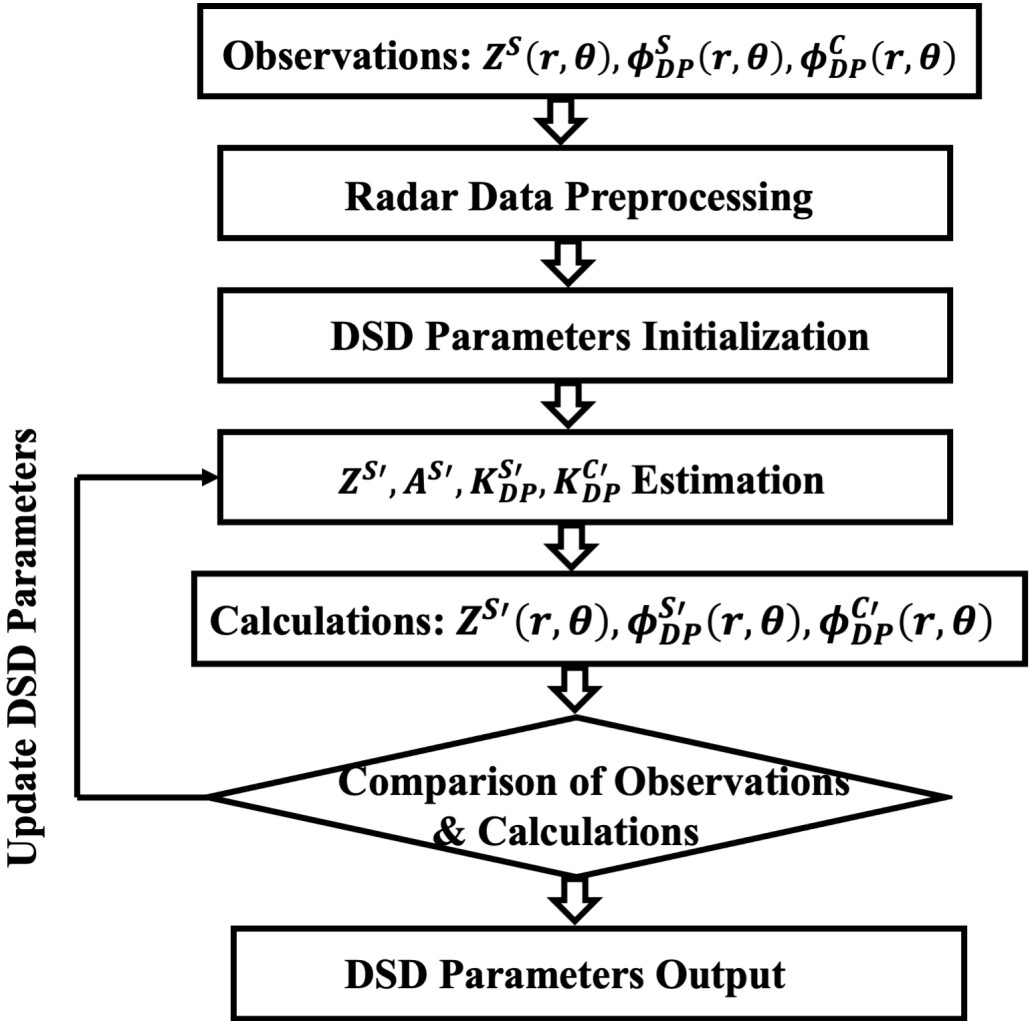

**Figure 5.** The algorithm applied to each gate seeks DSD parameters which produce reflectivity and estimated specific differential phase values that correspond to the radar reflectivity and differential phase values observed at the disdrometer location by RCMD and RCWF. The calculated values are indicated as $Z^{S'}$, $\phi_{DP}^{S'}$, etc. as opposed to the radar observed values indicated as $Z^{S}$, $\phi_{DP}^{S}$, etc.

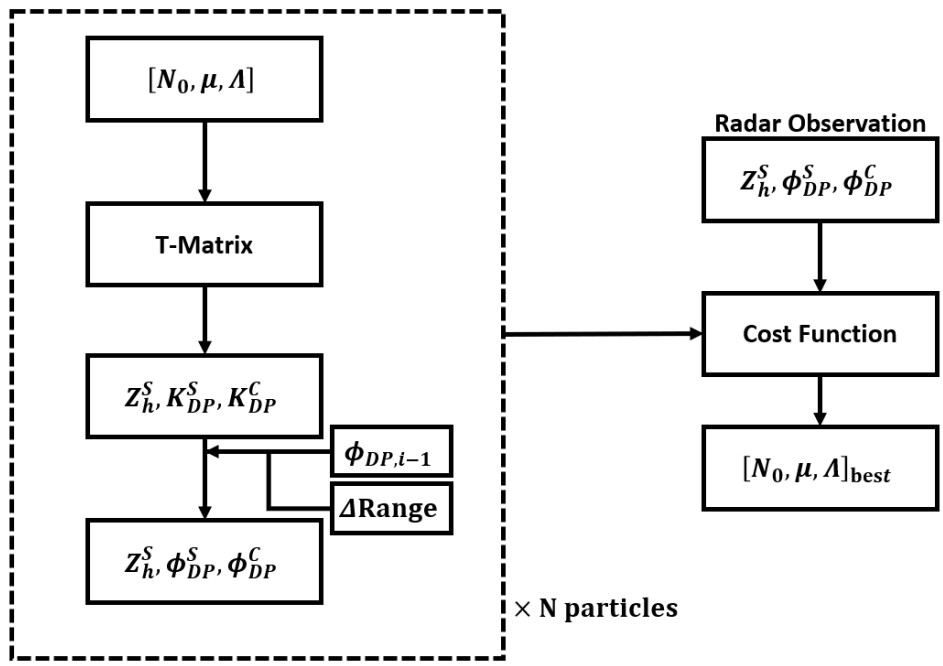

**Figure 6.** Particle Swarm Optimization for Gate of Interest. The left dashed box denotes the calculations for each particle which has an independent set of gamma distribution defining variables that is used to estimate reflectivity and specific differential phase values using the T-Matrix method. Range differentials and previously determined differential phase are used to produce estimated parameters which can then be applied to a cost function relative to the observed values.

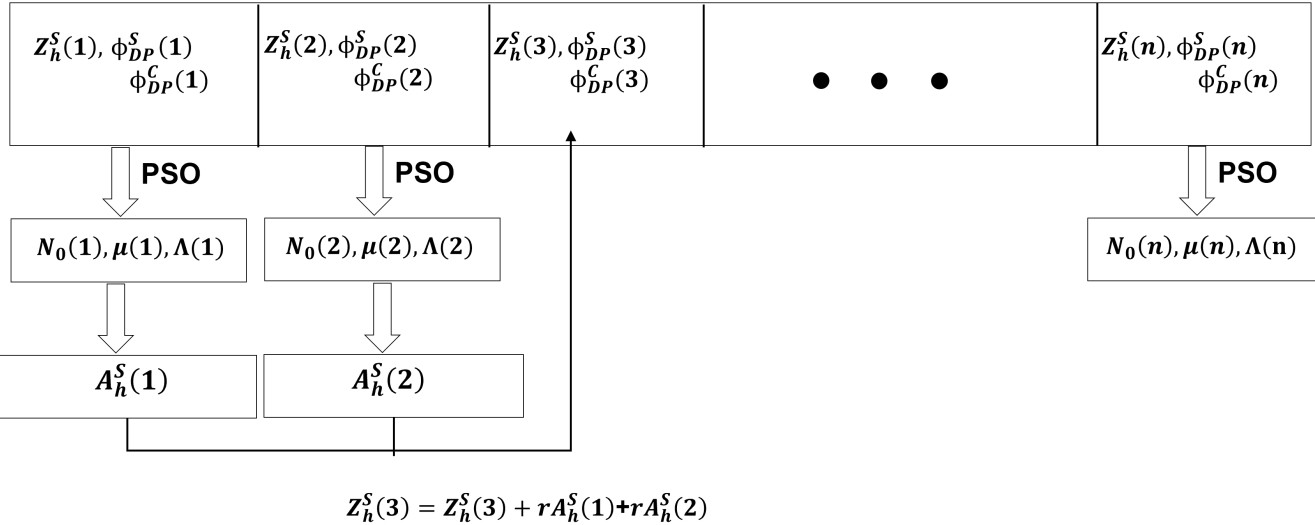

**Figure 7.** Retrieval Along Radial. The process begins with the first gate which is assumed to be unaffected by attenuation. The reflectivity and differential phase information serve as inputs to the PSO of each gate and produce DSD values that can be used to calculate the local attenuation. The next gate's observed reflectivity value is updated with any previously accumulated attenuation before being used in the next optimization. The process continues until the final gate's values are determined.

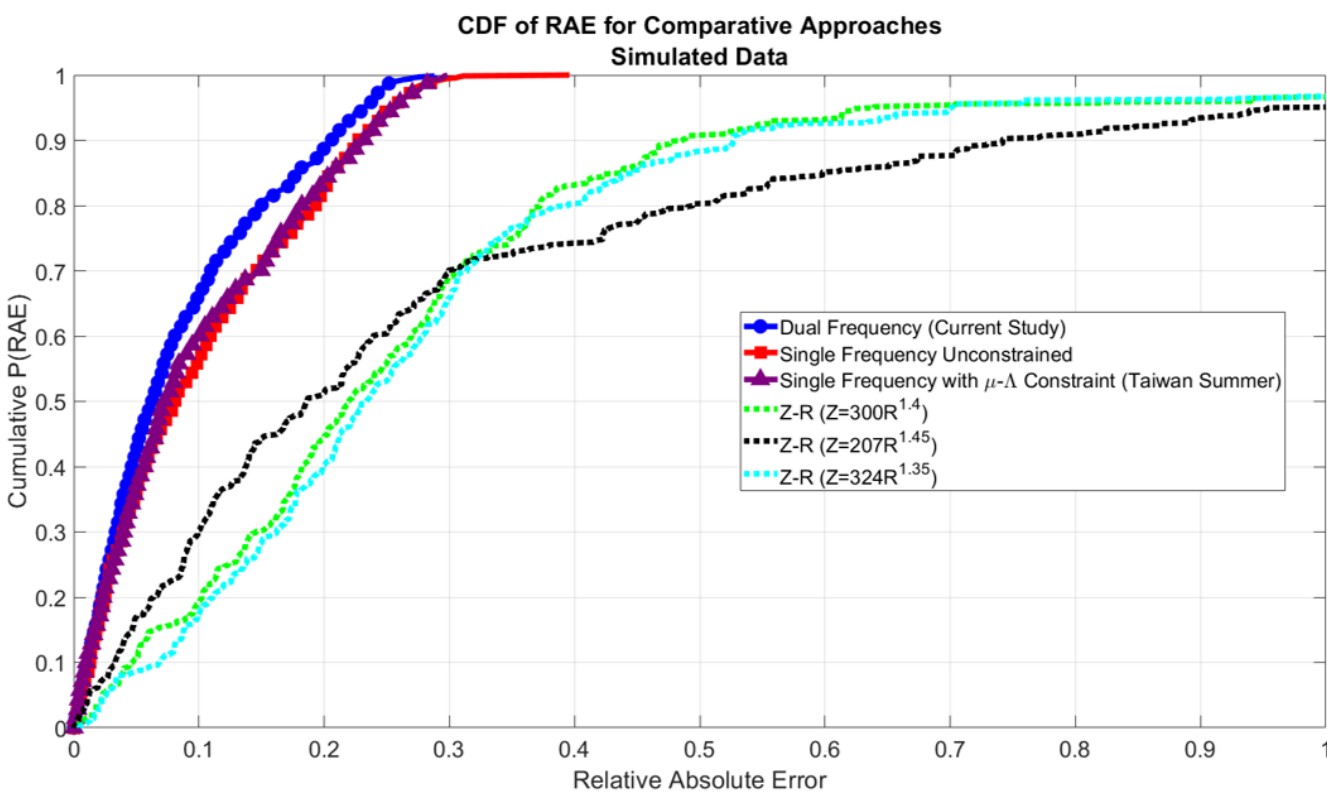

**Figure 8.** The distribution of errors (cumulative) associated with each approach are plotted. The results of this study's proposed methodology using two frequencies (blue circles) shows modest improvement when compared to single frequency retrievals with (purple triangle) and without (red square) assumed $\mu - \Lambda$ constraints. A more significant improvement is seen when compared to the various errors associated with *Z-R* rain rate estimations are indicated with dashed lines.

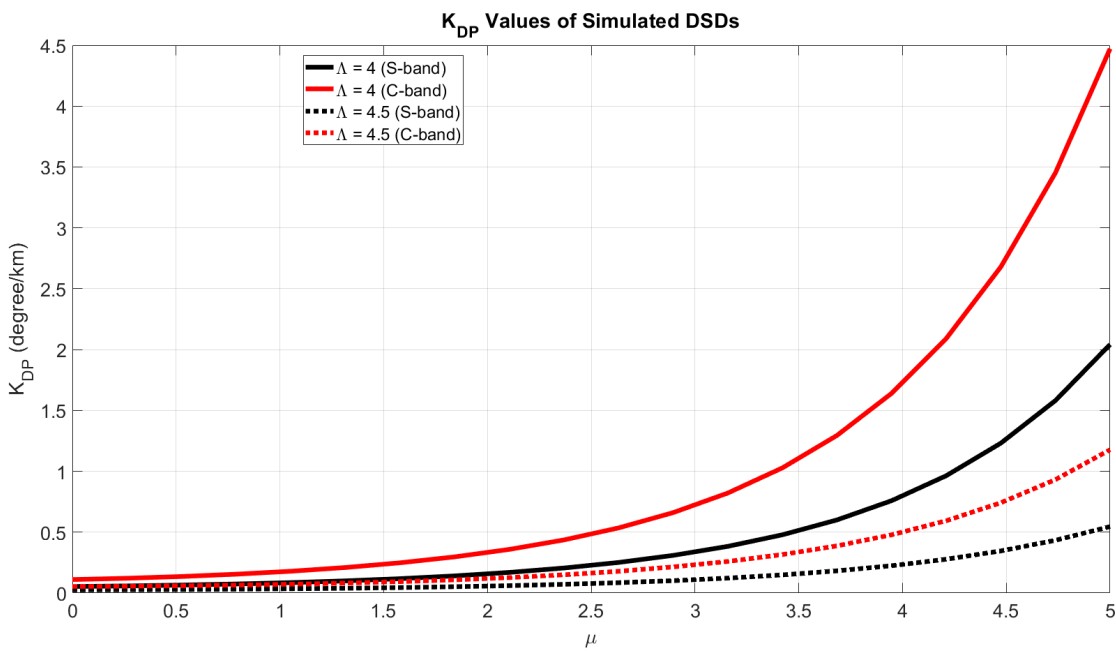

**Figure 9.** $K_{DP}$ is simulated for DSDs of varying $\mu$ and $\Lambda$ values with fixed $N_0 = 8000$. A separation between the $K_{DP}$ values is clear at the two radar bands.

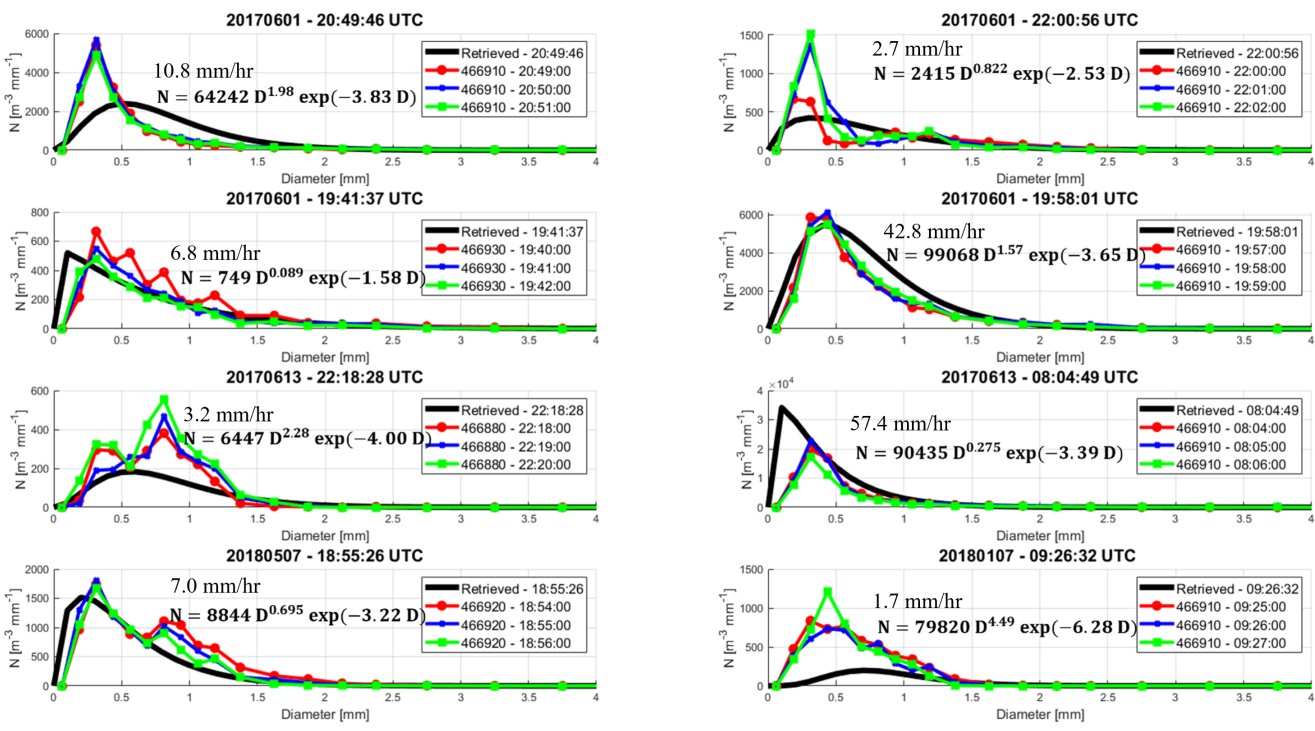

**Figure 10.** The retrieved DSD is shown in black on each plot with its gamma distribution equation. The DSD of the closest disdrometer record is plotted in blue while the previous record and next record time are shown in red and green, respectively. The median rainfall rate of the three disdrometer collections is indicated in mm/hr.

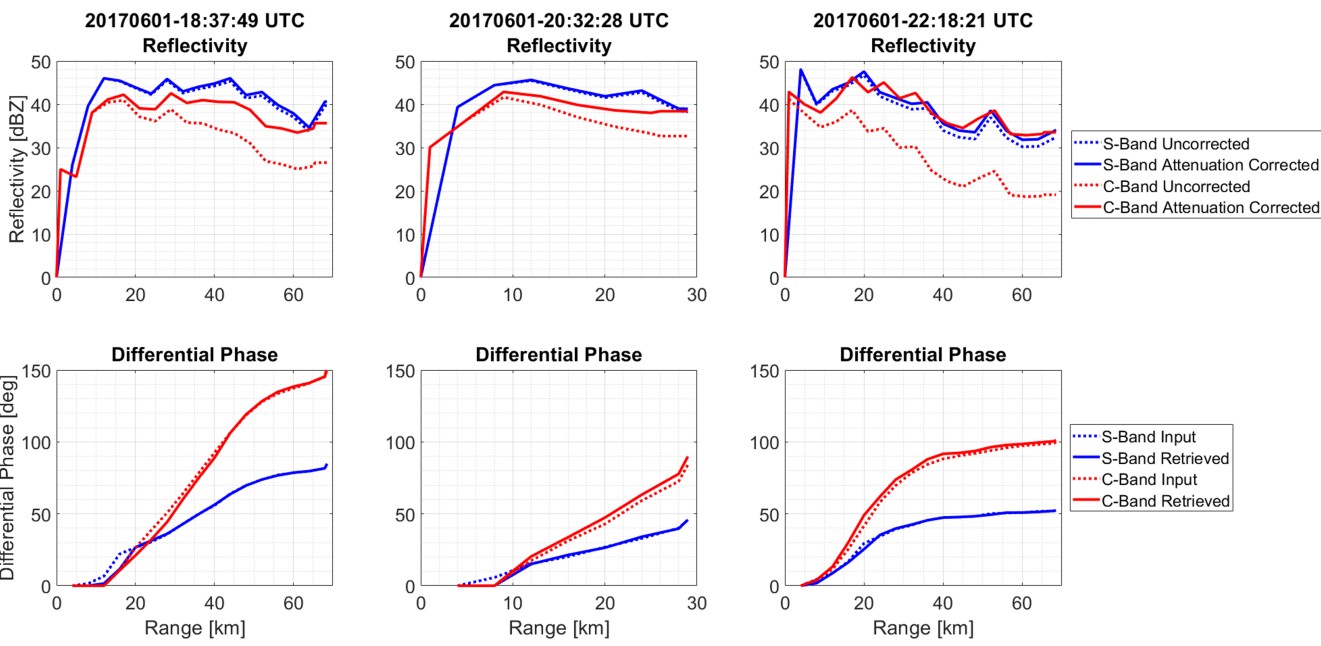

**Figure 11.** The S-band raw reflectivity (dashed blue line) and the attenuation-corrected reflectivity values indicated by solid blue. The corrected C-band reflectivity is shown with red solid lines relative to the uncorrected (dashed red) reflectivity. The input (dashed) and retrieved (solid) differential phase profiles of each case are included for each example.

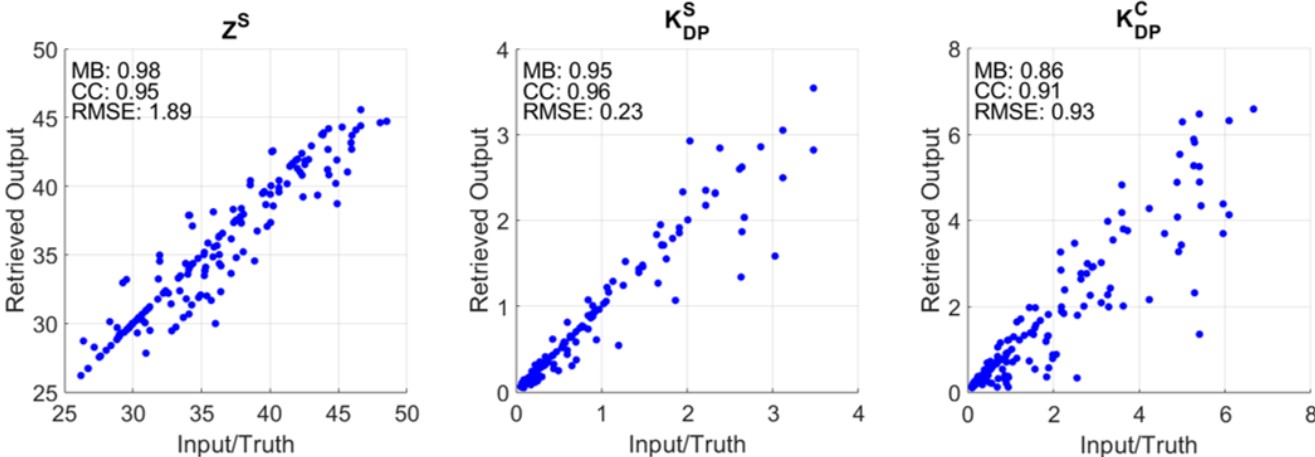

**Figure 12.** The retrieved calculated value of each parameter (y-axis) is shown relative to the algorithm's input (x-axis) based on the radar measurements at the final gate (disdrometer location). The results in terms of MB, CC, and RMSE are included.

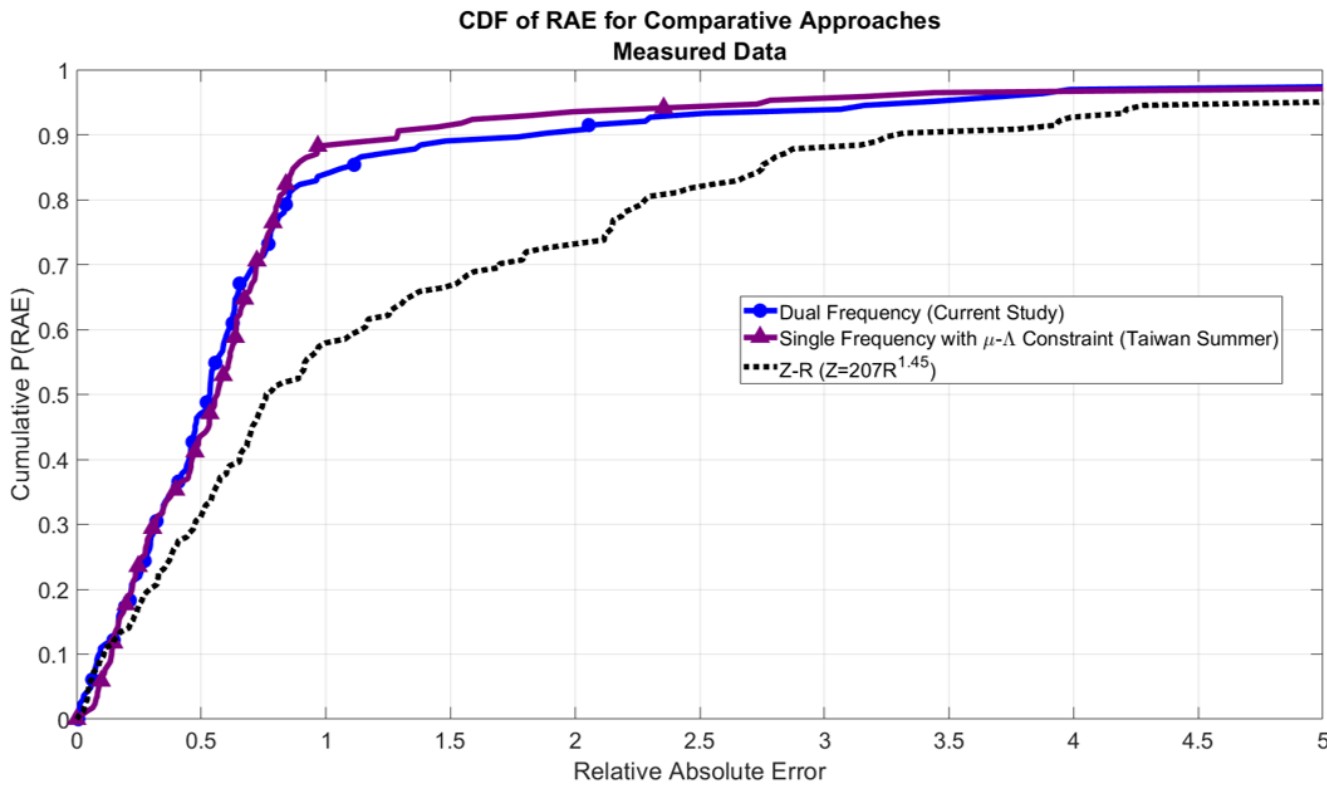

**Figure 13.** The performance of the dual-frequency approach (blue circles) shows roughly equivalent performance to the single frequency approach employing a relevant $\mu - \Lambda$ constraint (purple triangles). Both methods utilizing DSD information outperform the region-specific *Z-R* derived rainrates (black dash).