# Peer review of "Drop Size Distribution Retrieval Using Dual-Polarization Radar Observations at C-Band and S-Band"

_EGUsphere, 2023_

## Referee Comment (RC2)

**Review of "Drop Size Distribution Retrieval Using Dual Frequency Polarimetric Weather Radars"**

The authors conducted a study to retrieve DSD parameters and subsequently compared them with disdrometer measurements. However, the methodology was not sufficiently elucidated in the text. The manuscript fails to address several uncertainties, which are neither considered nor discussed thoroughly when comparing retrieved and measured DSD. The study appears incomplete, requiring more comprehensive analysis and comparisons between the retrieved and measured DSD. Therefore, I suggest that this manuscript can be accepted only if the authors make major revisions. Please refer to the provided comments and suggestions for guidance.

**General comments and suggestions:**

1) The authors should consider and discuss height difference between radar gates selected for the DSD retrieval and the disdrometers. The height of the radars are quite high.
2) It is better to briefly introduce the synoptic scale weather conditions for the cases you selected, especially about wind direction and speed, temperature, and relative humility. Evaporation, collision-coalescence will affect the DSD measured at disdrometer station as the raindrops fall from the height of "terminal gate".
3) I suggest performing the DSD retrieval at the terminal gate continuously for at least several radar scan to check robustness of the proposed method.
4) Unify the variable names such as $Z^S_{measured}$, $\Phi_{DP}{}^S$,measured (Eq.9), $Z^{S'}$, $\Phi_{DP}{}^{S'}$ etc. in the text and figures.

**Specific comments and suggestions:**

Line 5: What is the "assumptions of the collected data"? You should be more specific here.

Line 37: Delete word "remotely".

Line 60: Is the "phase" differential phase?

Line 61-71: The authors only depict the contents in section 2. What about section 3 and 4?

Line 74: Suggest changing the sentence to "the measurements of the two co-located polarimetric radars".

Line 93: Add year after "Jaffrain et al."

Line 98: Change "such and" to "such as".

Line 104: The description of the range rings "two rings with ranges of 20 km and 70 km are shown in Figure 2" is different from the one the Fig.2. Please make sure they are consistent.

Line 152: What is "$Z^S$"? reflectivity of S-band?  Change "dB" to "dBZ".

Line 155: What is "terminal gate"?

Line 154: Multiplier "2" is missed in front of $K_{DP}$ in Eq.6. Please also check your algorithm if $K_{DP}$ is multiplied by 2.

Line 195: Delete word "minimum".

Line 203: Change to "converge to a solution".

Line 211: What is the coordinates? Please give more specific description about "The three coordinates".

Line 207-220: I did not find the PSO method mentioned or applied in the referred article (Zhang et al., 2001). Please give more explanations on the word "particle" used in the PSO.

Line 215-222:   Are the parameters $\alpha$ and $\beta$ the same in Eq. 7 and 8?  What does index "i" stand for in Eq.8?

Line 226-238: Based on my understanding, the parameters of DSD can be retrieved at each radar gate individually. If just used measured $Z^S$ with less attenuation, $\Phi_{DP}{}^S$, and $\Phi_{DP}{}^C$ at "terminal gate" to retrieve directly instead of the way described here, what difference will make on retrieved DSD?

Line 246-251: Reorganize this part. Please also give the heights of radar "terminal gate" and the disdrometer stations and label them in Fig. 3.

Line 253-254: Rewrite this sentence.

Line 278-284: The "blue/red circles" and "blue/red triangles" are not found in Fig. 10.

Line 283-284: What is the "correction factor"? Please explain how do you "predict" what atmospheric effect.

Fig. 2: Suggest change the sentence in the caption to "radii are drawn around the radar location." In addition, delete "to indicate the data viability region."

Fig.3 and 4: At what elevation angle and time?

Fig. 5: Please define the $Z^{S'}$, $A^{S'}$, etc. in the caption.

Fig. 6: The $Z^S{}_h$, $\Phi^S{}_{DP}$, etc. in the dash line box should calculated. Please do not mix up with measured $Z^S{}_h$, $\Phi^S{}_{DP}$, etc. on the right side of figure.

---

## Author Comment (AC1)

Thank you for your question regarding our manuscript.

In selecting the smoothing window length for our study, we carefully considered the trade-offs between resolution and noise reduction. A longer smoothing window, while beneficial for reducing noise and bias, can adversely affect the resolution, leading to a loss of detail in the radar data. Conversely, a shorter window may not effectively mitigate these noise and bias elements, compromising the quality of the data.

Since there is no standard for filter length with respect to variable smoothing, we referenced the specific differential phase calculation methods utilized by Ryzhkov[1] which employ window lengths of 9 gates (2.25 km) and 25 gates (6.25 km). We opted for an approximate average between these two values in our analysis. This decision was based on the observation that within this range, the smoothing effect remains relatively stable against minor variations in the filter length.

However, it's important to highlight that window lengths exceeding 25 gates can lead to over-smoothing, potentially obscuring critical features in the radar data. Similarly, window lengths shorter than 9 gates might not be sufficient to effectively reduce noise. Thus, our chosen average aims to strike a balance, ensuring both adequate noise reduction and preservation of important radar data characteristics. This is easily seen in the following example from our dataset.

[Figure]

[1] A. V. Ryzhkov, S. E. Giangrande, V. M. Melnikov, and T. J. Schuur. Calibration issues of dual-polarization radar measurements. Journal of Atmospheric and Oceanic Technology, 22(8):1138–1155, 2005. doi: 10.1175/jtech1772.1.

---

## Author Comment (AC2)

This manuscript introduces a method using particle swarm optimization (PSO) to estimate the three parameters of a gamma shaped raindrop size distribution (DSD) along a radial beam using two side-by-side scanning radars operating at S- and C-band. There are several concerns in the manuscript that need to be addressed before this manuscript can be considered for publication.

Response: Thank you for reviewing our manuscript and improving our study with your suggestions. The valid criticism you voiced concerning our original manuscript primarily hinges on the lack of a robust statistical evaluation of our retrieval results. We have reprocessed our dataset in order to collect these statistics. We would like to address your comments and questions with the following in-line responses.

1. While the abstract states that the retrieval method "is able to accurately retrieve the gamma distribution parameters without the constraints required in previous methods" (line 9), the manuscript does not present any retrieved parameters, nor does it estimate the accuracy of any retrieved parameters. Thus, it is not possible to evaluate the accuracy of the proposed method because the necessary results are not presented in this manuscript.

Response: Thank you for highlighting the omission in our manuscript regarding the presentation of retrieved parameters and their accuracy.

To address this, we will update the manuscript to explicitly include following revisions:
a.) providing the retrieved gamma distribution parameter values for each example in the plot. The updated figures are shown as below. These figures replace the subjective examples in our original manuscript. In these new figures, the disdrometer observations from three continuous minutes are presented using red, blue and green lines, respectively. The retrieved DSD using the radar data of the closet moment is delineated with a black curve.

[Figure]

[Figure]

b.) Following the reviewer's suggestion, we quantitatively evaluated the performance of the proposed approach and look forward to including the results in the revised manuscript. In the quantitative evaluation, the rainfall rates were first estimated using three different approaches:

i.) using the retrieved DSD parameters following equation $R = \frac{\pi}{6} \int_0^{Dmax} D^3 N(D) v(D) dD$ (Bringi 20002, Zhang 2001, etc);

ii) using the S-band radar reflectivity ($Z$) following the WSR-88D $R$-$Z$ relationship, $Z = 300 R^{1.4}$ (Ulbrich and Lee 1999);

iii) using the DSD observed by the Parsivel disdrometer following equation $R = \frac{6 \pi \times 10^4}{\Delta t} \sum_{j=1}^{M} \frac{D_j^3}{S_j^{2DVD}}$ (Raupach and Berne, 2015).

The rainfall rates from i and ii were then compared with the iii, which was treated as the ground truth. In the comparison, the relative absolute error (RAE) was calculated as.

$$\epsilon = \frac{|R_d - R|}{R_d}$$

where $R_d$ are the rainfall rate estimated from the disdrometer as presented in approach iii, and $R$ are the evaluating rainfall rate from approach i or ii, respectively.

Total 167 cases were used in the analysis. The criteria of cases selection are:
1.) time difference between S- and C- band scan is within 1 minutes
2.) only the lowest two elevation angle (0.5° and 1.4°) are used.
3.) reflectivity > 25 dBZ
4.) 25 km < disdrometer range < 70 km

The time series plot presented below illustrates the RAE results for two different approaches. Approach i, our proposed method, is represented by the blue line, while Approach ii, which employs the conventional $R(Z)$ method, is indicated by the red line. The plot demonstrates that estimating rainfall rates using retrieved DSD parameters, as in our proposed approach, yields higher accuracy compared to the traditional $Z$-$R$ relationship. Specifically, the median RAE for the $Z$-$R$ approach stands at 0.72, which is notably reduced to 0.53 with our proposed method. This represents a significant improvement of 26.4% as observed in this study.

[Figure]

In the revision, the quantitative evaluation results and discussions will be added.

2.  As presented in this manuscript, the PSO method is essentially a random walk through the three DSD parameter space, and not an "optimization" method. A random walk (line 218) through the parameter space is a valid parameter estimation procedure if the neighborhood of solutions around the global solution are used to estimate an uncertainty of the retrieved parameters.

Response:  We appreciate your perspective on our use of the PSO method. While we understand your viewpoint that categorizes PSO as a random walk, we would like to clarify that PSO inherently possesses mechanisms that actively drive towards a global minimum, distinguishing it from a purely random walk approach. The intent behind selecting PSO was not to showcase it as a novel optimization technique but rather to utilize a method capable of overcoming the limitations commonly encountered in gradient-based approaches, namely convergence to local minima.

In earlier iterations of our work, we employed the traditional non-linear optimization method, such as Gauss-Newton (GN) algorithm and Levenberg-Marquardt (LM) algorithm. However, this type of method tend to converge at local minimum instead of global minimum. As an alternative, we tried unsupervised learning approaches such as genetic algorithm (GA) and PSO. They both yielded results superior to those achieved with a GN or LM algorithm. The choice to transition to PSO was primarily driven by its efficiency relative to our initial genetic algorithm code. It's important to note that our primary focus was on the effectiveness of the solution method rather than the optimization routine itself. We believe that several other methods could also be successfully employed for the solver block of our algorithm.

3.  It appears that the study is based on four radial samples made over two different disdrometers. The manuscript text states that nine (line 245) or ten (line 294) days of data were "used in the performance validation" (line 245), but only four reconstructed raindrop distributions are shown in Figure 9. Since no more retrieved parameters are shown in the manuscript, it must be assumed that those four radials were the only radials that were processed.

Response: We agree with regard to the noted deficiencies in our original manuscript, as noted here as well as the first comment. In order to address the absence of a statistical presentation of the data, we have reevaluated our dataset and will include the results shown in the response to comment 1 which includes the retrievals of 167 cases. More details please refer to the response to comment 1.

4.  It is major limitation of this work that retrieved DSD parameters are not presented in the manuscript. The four reconstructed raindrop distributions shown in Figure 9 do not show the retrieved slope parameter (lambda) nor the shape parameter (mu). Since three of the four constructed distributions (Fig. 8 a, b, and d) do not have the number concentration approaching zero at zero diameter, this suggests that the retrieved mu value is less than zero. It is unusual for the shape parameter to be negative in scanning radar retrievals due to the large scanning radar

sample volume. Which raises concern about the validity of the retrievals. The manuscript must show retrieved parameter statistics to evaluate the performance of the retrieval method.

Response: We had originally restricted our parameters along the bounds shown in Zhang's (2001) work, which allowed $\mu$ to be as low as negative 2. We have updated the constraint on $\mu$ to be no less than zero. We believe the trade off in possibly losing a few distributions with negative $\mu$ shown in Zhang is acceptable with the gained benefit of guarding against non-sensical distributions. We thank you for that note, and all cases have been reprocessed with the updated bounds. Additionally, we have included values of $\mu$, $\Lambda$, and $N_0$ in each subjective plot. Please see the updated subjective examples shown in response to your first comment.

5.  How much improvement in rainfall estimation does the proposed method provide compared to using one of the mu-lambda relationships proposed by Zhang et al. (2001), Brandes et al. (2002), or Cao et al. (2010)? If the new method does not produce comparable or better rainfall estimates than the mu-lambda constraints, then will this proposed method be an improvement to QPE? The mu-lambda constraints should be the baseline that the new method should aim to beat. In summary, without the manuscript showing results of the retrieved DSD parameters, it is not possible to evaluate the proposed retrieval method. The proposed method may produce results that are superior to results from imposing a mu-lambda constraint. But, as written, the manuscript does not show the evidence needed to verify the statements made in the manuscript.

Response: We acknowledge the oversight of not benchmarking our initial manuscript. In addition to the comparison with the Z-R derived rain rate, we have also conducted an analysis of our algorithm incorporating the $\mu$-$\Lambda$ relationship outlined in Zhang et al. (2001):

$$\mu = -0.016\,\Lambda^2 + 1.213\Lambda - 1.957$$

Different from the proposed method that retrieves three DSD parameters independently, this approach retrieves the DSD with above $\mu$-$\Lambda$ relationship. The PSO method is also used in this approach.

The following figure shows the rain rate derived from the proposed method's retrieval (blue), the disdrometer truth (black), *Z-R* relation (red), and the PSO retrieval with a $\mu$-$\Lambda$ constraint (green).

[Figure]

The accuracy of the $\mu$-$\Lambda$ constrained retrieval method is found to be comparable to that of our unconstrained retrieval technique. Evaluating performance with the RAE metric reveals that the constrained approach has a median error of 0.55, closely aligning with the unconstrained method's median error of 0.53. While a mean RAE evaluation might suggest enhanced performance with the mu-Lambda relationship, we believe using median values for assessment provides a truer reflection of performance, particularly due to the influence of outliers. To illustrate this, the attached figure displays error plots for all methods, with outliers truncated at a value of 8.

[Figure]

It should be noted that the approach proposed by Zhang et al. (2001) incorporates $Z_{DR}$ (Differential Reflectivity) in retrievals, a factor which we acknowledge could potentially enhance accuracy. However, we opted not to use $Z_{DR}$ due to its requirement for regular calibration, which introduces an element of uncertainty that we preferred to avoid. We acknowledge a $\mu$-$\Lambda$ constraint might be advantageous in various contexts, however, we found it to be non-essential for our specific application. The advantage of bypassing the need to establish this relationship through extensive prior data collection is clear. We will further investigate the impacts of the $\mu$-$\Lambda$ constraint in the retrieval results in the future work.

---

## Author Comment (AC3)

**Review of "Drop Size Distribution Retrieval Using Dual Frequency Polarimetric Weather Radars"**

The authors conducted a study to retrieve DSD parameters and subsequently compared them with disdrometer measurements. However, the methodology was not sufficiently elucidated in the text. The manuscript fails to address several uncertainties, which are neither considered nor discussed thoroughly when comparing retrieved and measured DSD. The study appears incomplete, requiring more comprehensive analysis and comparisons between the retrieved and measured DSD. Therefore, I suggest that this manuscript can be accepted only if the authors make major revisions. Please refer to the provided comments and suggestions for guidance.

**General comments and suggestions:**

1) The authors should consider and discuss height difference between radar gates selected for the DSD retrieval and the disdrometers. The height of the radars are quite high.

   Response: We acknowledge that the difference between radar volume and the disdrometer observation poses challenges in all radar-based retrieval algorithms. In our study, we have made concerted efforts to minimize the impact of this discrepancy. This was achieved by restricting our data collection to the lowest two radar elevation angles and confining the range of our retrievals to within 70 km.

   While these measures represent the best possible approach given the fixed positions of the equipment, we understand the importance of thoroughly discussing these inherent limitations. We will ensure that a comprehensive discussion of these height differences and their associated constraints is included in the revised version of our manuscript, providing a clear and honest assessment of the limitations of our study.

2) It is better to briefly introduce the synoptic scale weather conditions for the cases you selected, especially about wind direction and speed, temperature, and relative humility. Evaporation, collision-coalescence will affect the DSD measured at disdrometer station as the raindrops fall from the height of "terminal gate".

   Response: Thank you for pointing this out. The majority of our samples were collected in June 2017, during the "Meiyu" rain season, a period characterized by varying weather conditions from light drizzle to thunderstorms. The ground temperature is around $25^{\circ}$ C, and normally no strong wind is associated with the rainfall during the Meiyu season. To simplify the model, no evaporation, collision-coalescence are considered in this work. We will convey this information in the revised manuscript.

3) I suggest performing the DSD retrieval at the terminal gate continuously for at least several radar scan to check robustness of the proposed method.

Response: Thank you for the suggestion. In the revision, multiple scans were used in the performance evaluation, although not continuously due to the different VCPs of the radar and our time synchronization requirements.

We quantitatively evaluated the performance of the proposed approach and look forward to including the results in the revised manuscript. In the quantitative evaluation, the rainfall rates were first estimated using three different approaches:

i.) using the retrieved DSD parameters following equation $R = \frac{\pi}{6} \int_0^{D_{max}} D^3 N(D) v(D) dD$
(Bringi 20002, Zhang 2001, etc);
ii) using the S-band radar reflectivity (Z) following the WSR-88D R-Z relationship, $Z = 300 R^{1.4}$ (Ulbrich and Lee 1999);
iii) using the DSD observed by the Parsivel disdrometer following equation $R = \frac{6 \pi \times 10^4}{\Delta t} \sum_{j=1}^{M} \frac{D_j^3}{S_j^{2DVD}}$ (Raupach and Berne, 2015).

The rainfall rates from i and ii were then compared with the iii, which was treated as the ground truth. In the comparison, the relative absolute error (RAE) was calculated as.

$$\epsilon = \frac{|R_d - R|}{R_d}$$

where $R_d$ are the rainfall rate estimated from the disdrometer as presented in approach iii, and $R$ are the evaluating rainfall rate from approach i or ii, respectively.
Total 167 cases were used in the analysis. The criteria of cases selection are:
1.) time difference between S- and C- band scan is within 1 minutes
2.) only the lowest two elevation angle ($0.5°$ and $1.4°$) are used.
3.) reflectivity > 25 dBZ
4.) 25 km < disdrometer range < 70 km

The time series plot presented below illustrates the RAE results for two different approaches. Approach i, our proposed method, is represented by the blue line, while Approach ii, which employs the conventional R(Z) method, is indicated by the red line. The plot demonstrates that estimating rainfall rates using retrieved DSD parameters, as in our proposed approach, yields higher accuracy compared to the traditional Z-R relationship. Specifically, the median RAE for the Z-R approach stands at 0.72, which is notably reduced to 0.53 with our proposed method. This represents a significant improvement of 26.4% as observed in this study.

[Figure]

In the revision, the quantitative evaluation results and discussions will be added.

4) Unify the variable names such as ZSmeasured, KDP ,measured (Eq.9), ZS', KDPS' etc. in the text and figures.

Response: We will ensure all variables are properly defined and standardized in the revised manuscript.

**Specific comments and suggestions:**

Response: We greatly value your suggestions and agree that each one merits inclusion in our revised manuscript. In response to your specific comments, we have provided in-line responses, including the implementation of 'was-is' formatting where relevant. Thank you for your insightful contributions.

Line 5: What is the "assumptions of the collected data"? You should be more specific here.

We were referencing the mu-Lambda constraint which can vary depending on location and meteorological conditions. We will rewrite this to be explicit.

Line 37: Delete word "remotely".

*Extensive research has been conducted to  estimate the drop size distribution (DSD), with many studies utilizing measurements taken at two frequencies.*

Line 60: Is the "phase" differential phase?

*This approach allows for the determination of DSD solutions that accurately represent the input reflectivity and specific differential phase information.*

Line 61-71: The authors only depict the contents in section 2. What about section 3 and 4?

We will rectify this omission by including a brief synopsis of the additional two sections in the introductory material.

Line 74: Suggest changing the sentence to "the measurements of the two co-located polarimetric radars".

*In the current work, the measurements of the two co-located polarimetric radars, RCWF (S-band) and RCMD (C-band), are used in the algorithm development and validation.*

Line 93: Add year after "Jaffrain et al."

The findings of Jaffrain et al. (2011) demonstrated that sampling uncertainty

Line 98: Change "such and" to "such as".

*Other factors, such as  the angle of the drop trajectory, coincidentally observed particles, and particles that intersect with only the edge will also lead to biases.*

Line 104: The description of the range rings "two rings with ranges of 20 km and 70 km are shown in Figure 2" is different from the one the Fig.2. Please make sure they are consistent.

After checking the code to generate the plot, the range rings correspond to the ranges in the body of the text, and it is the "100 km" reference in the caption that is in error. Thank you for highlighting this. We will update it in the revised manuscript.

Line 152: What is "$Z^S$"? reflectivity of S-band?  Change "dB" to "dBZ".

$Z^S > 25$ dBZ

Line 155: What is "terminal gate"?

We later define terminal gate at line 230. We will move the definition (the gate at the disdrometer location) to line 155 where it is first used.

Line 154: Multiplier "2" is missed in front of $K_{DP}$ in Eq.6. Please also check your algorithm if $K_{DP}$ is multiplied by 2.

Thank you for highlighting this error. You are correct, and we will update Eq. 6 to correctly include the two-way factor. The algorithm does account for attenuation in both directions.

Line 195: Delete word "minimum".

the  system phi_dp

Line 203: Change to "converge to a solution".

While it can quickly converge  to a solution,

Line 211: What is the coordinates? Please give more specific description about "The three coordinates".

The three coordinates of the solution space are N_0, mu, and Lamda. We will rewrite this sentence to be more specific and make sure we haven't made too far a leap in relating the DSD parameters to a coordinate space.

Line 207-220: I did not find the PSO method mentioned or applied in the referred article (Zhang et al., 2001). Please give more explanations on the word "particle" used in the PSO.

Zhang 2001 was referenced for the range of values used in the solution space (second half of the paragraph). We will find a general reference to include in the revised manuscript that can point the reader to an overview of PSO that is consistent with our application. We do want to refrain from misleading the reader that the use of PSO is a major innovation. We contend that many other optimization approaches would work similarly so long as they can overcome local convergence issues.

Line 215-222: Are the parameters alpha and beta the same in Eq. 7 and 8? What does index "i" stand for in Eq.8?

These parameters are different and we should have used different variables in our initial submission to avoid any possible confusion. They are used as weighting factors of the cost function in Eq. 7 and as local/global convergence factors in Eq. 8. We will update this in the revised manuscript.

"i" in Eq. 8 refers to one iteration of the optimization routine as applied to a single gate. We will more thoroughly describe these definitions in the revised manuscript.

Line 226-238: Based on my understanding, the parameters of DSD can be retrieved at each radar gate individually. If just used measured $Z^S$ with less attenuation, $k_{DP}^S$, and $k_{DP}^C$ at "terminal gate" to retrieve directly instead of the way described here, what difference will make on retrieved DSD?

We retrieved the DSDs for the gates between the radar and the terminal gate solely to more accurately calculate the attenuation factor (also a T-matrix calculation using the DSD) at each position and correct for the accumulated attenuation for the final retrieval. While this may not be as important at S-band, it does become a necessary step for the disdrometer at station 466950 which is located near our 70 km limit.

Line 246-251: Reorganize this part. Please also give the heights of radar "terminal gate" and the disdrometer stations and label them in Fig. 3.

We will reorganize these lines to properly reference Fig 8 before beginning to talk about it. For the radar and disdrometer heights we would prefer to construct an additional table that will include the data you are requesting. This may be a cleaner approach than including it in the Fig. 3.

Line 253-254: Rewrite this sentence.
 First, the radar cross section, determined by the physical cross section, is inherently smaller for these drops, thereby exerting a lesser impact on both reflectivity and phase inputs compared to larger drops.

Line 278-284: The "blue/red circles" and "blue/red triangles" are not found in Fig. 10.

We had simplified the plot, but failed to update the manuscript body. Thank you for catching this. We will revise to reference the proper plot symbology.

Line 283-284: What is the "correction factor"? Please explain how do you "predict" what atmospheric effect.

We will rewrite this section to replace "correction factor" with "attenuation factor" or similar. As stated in response to your insightful comment on Line 226-238, the attenuation is calculated for the intermediate positions using their retrieved DSDs and the T-matrix method.

Fig. 2: Suggest change the sentence in the caption to "radii are drawn around the radar location." In addition, delete "to indicate the data viability region."

Figure 2. Taiwan - Instrument Locations. Radii (25 km and 70 km) are drawn around the radar location. Measurement stations with disdrometers are shown in red. Terrain height is indicated in grayscale throughout the map for reference.

Fig.3 and 4: At what elevation angle and time?

These all correspond to 0.5 degrees elevation. We will note this in the caption. The time is contained in the title of each subplot but may be missed. We will add "UTC" after each HHMMSS timetag to clarify.

Fig. 5: Please define the $Z^{S'}$, $A^{S'}$, etc. in the caption.

We will ensure all variables are clearly defined near their use when we edit the manuscript in response to your general comment 4.

Fig. 6: The $Z^S_h$, $k^S_{DP}$, etc. in the dash line box should calculated. Please do not mix up with measured $Z^S_h$, $k^S_{DP}$, etc. on the right side of figure.

We will revise the figure to explicitly state the values on the left side of the figures are calculated values and not the measured observed values.

---

## Author Comment (AC4)

The discussion paper by Durbin et al. looks quite attractive. There are several countries in the world with weather networks composed of C- and S- band with possible overlapping. A dual-frequency approach applied to C- and S- band radar collocated measurements can be of interest and the set-up of practically co-located radars used for the study is quite uncommon.

However, the manuscript lacks of and adequate description of the development of the technique and a thorough validation.

There are many examples in the literature of techniques aiming at estimating DSD with assumptions on gamma DSD and its parameters. The one proposed is based on Kdp at the two frequencies and the equivalent reflectivity factor at S-band, supposed to be not affected by attenuation due to precipitation.

Response: Thank you for reviewing our manuscript and improving our study with your suggestions. We'd like to address your comments and revised the manuscript following your suggestions.

Usually, algorithms are first investigated in ideal conditions (i.e. through T-matrix simulated measurements) and then the performance is investigated in the presence of different error sources (i.e. calibration, attenuation, spatial gradient, difference in volume sampling, etc.) and finally validated with real measurement and comparison with real data.

Response: As the reviewer suggested, we did study the possibility of using co-located dual-polarization radars to retrieve the DSD through simulation as the first step. In the simulation, we first calculated the backward/forward scattering amplitude using T-matrix method, and then calculated the reflectivity and specific differential phase fields from S-band and C-band for a given DSD assumption. The simulated results are shown as follows for the gamma model using various values of $\mu$ and $\Lambda$ with $N_0=8000$.

Figure 1

[Figure]

*Figure 1 –Simulated Reflectivity For Various Parameters –* The S and C-band returns are too similar to provide independent information for retrieval purposes.

Figure 2

[Figure]

*Figure 2 –KDP Variation For Various Parameters –* The S and C-band returns vary in KDP values which is a key premise of the algorithm. KDP values separate at the two frequencies at low and moderate values of $\Lambda$. The high values of $\Lambda$ do not provide useful separation as can be imagined by such a quickly collapsing distribution.

Figure 3

[Figure]

*Figure 3 – Attenuation Variation For Various Parameters* – As expected, the attenuation is much greater at C-band relative to S-band. Large separation in attenuation values are present for identical gamma parameters.

The simulations reveal a pronounced attenuation in the C-band reflectivity field, whereas the attenuation observed in the S-band reflectivity is significantly less pronounced. Furthermore, a clear separation of KDP values is present at each band. These findings underpin our selection of these specific variables and illustrate the methods through which they can be leveraged for retrieval processes.

Following the reviewer's suggestion, we will add simulation results into the revised manuscript.

In the first steps of the development missing is a thorough analysis of the impact of the error sources on the retrieval technique.

Response: We appreciate the reviewer's insightful observation regarding the absence of a comprehensive discussion regarding the impact of error sources on our retrieval technique. We acknowledge the oversight and recognize the importance of discussing the sources of error in our manuscript.

The primary sources of error stem from four aspects:

1. Observation Bias: The inherent discrepancy between radar (volume observation) and disdrometer (point observation) can introduce biases. This discrepancy arises because

radars observe a larger volume, whereas disdrometers measure at a specific point, leading to potential inconsistencies between the two measurements.

2. Vertical DSD Variations: Given that radar volumes are situated several hundred meters above the ground, discrepancies in DSDs observed by radar compared to those by disdrometers are anticipated. This issue represents a significant challenge in all radar-based DSD retrieval methods. To mitigate this, our dataset was limited to the two lowest elevation tilts of radar scans, although this approach only partially addresses the error source. This is the extent to which we can control any possible error from this source given the fixed locations of our equipment.

3. Measurement Errors: Errors from both radar and disdrometer measurements were considered. These are discussed in detail in the original manuscript.

4. Retrieval Error: A certain degree of preprocessing is required to remove noise in the data. While this is an essential step, it carries the possibility of smoothing out local phenomena which could introduce error. There is also a potential for error if the retrieval method converges to a local minimum rather than the global minimum, although we utilized an algorithm which is more likely to overcome this possibility.

In response to the reviewer's comments, we will incorporate a detailed discussion of these error sources in the revised manuscript to provide a clearer understanding of their impact on our retrieval technique.

Only a very limited comparison with a Parsivel is shown that could be influenced by many sources of errors, including the limitation of the Parsivel itself (authors honestly use "qualitatively" word). However, even for this case, the error associated with the retrieval is not provided, and the range of variability of retrieved data is not provided as well.

Moreover, the technique is not checked against techniques based on single frequency dual-polarization radar that can be found in the literature so that a reader can understand the advantage of having two radars insted of a cheaper single radar.

The text should be revised, since is not very clear on several parts.

Response: The main omission and criticism you have outlined is consistent with that of the other reviewers- the lack of a comprehensive performance assessment against an acceptable benchmark. This has prompted us to significantly expand the scope of our study.

To address this, we will update the manuscript to explicitly include following revision:

We quantitatively evaluated the performance of the proposed approach and look forward to including the results in the revised manuscript. In the quantitative evaluation, the rainfall rates were firstly estimated using three different approaches: i.) using the retrieved DSD parameters following equation $R = \frac{\pi}{6} \int_0^{D_{max}} D^3 N(D) v(D) dD$ (Bringi 20002, Zhang 2001, etc); ii) using the S-band radar reflectivity ($Z$) following the WSR-88D $R$-$Z$ relationship, $Z = 300\ R^{1.4}$ (Ulbrich and Lee 1999) and iii) using the DSD observed by the Parsivel disdrometer following equation $R = \frac{6\ \pi \times 10^4}{\Delta t} \sum_{j=1}^M \frac{D_j^3}{S_j^{2DVD}}$ (Raupach and Berne, 2015). The rainfall rates from i and ii were then

compared with the iii, which was treated as the ground truth. In the comparison, the relative absolute error (RAE) was calculated as.

$$\epsilon = \frac{|R_d - R|}{R_d}$$

where $R_d$ and R are the rainfall rate from iii and i/ii, respectively.

Total 167 cases were used in the analysis. The criteria of cases selection are:

1.) time difference between S- and C- band scan is within 1 minutes
2.) only the lowest two elevation angle ($0.5°$ and $1.4°$) are used.
3.) reflectivity>25
4.) 25<disdrometer range<70 km

The time series plot presented in the following figure illustrates the RAE results for two different approaches. Approach i, our proposed method, is represented by the blue line, while Approach ii, which employs the conventional R(Z) method, is indicated by the red line. The plot demonstrates that estimating rainfall rates using retrieved DSD parameters, as in our proposed approach, yields higher accuracy compared to the traditional Z-R relationship. Specifically, the median RAE for the Z-R approach stands at 0.72, which is notably reduced to 0.53 with our proposed method. This represents a significant improvement of 26.4% as observed in this study.

Figure 4

[Figure]

*Figure 4 – Retrieval Error Evaluation–* The retrieval algorithm's performance evaluated as RAE is shown in blue while the RAE associated with the Z-R derived rainrate is shown in red. Outliers are truncated at 8 in order to maintain a more useful vertical scale in the plot.

 In the revision, the quantitative evaluation results and discussions will be added.

Some specific issues are listed below:

a)    A title like "Drop Size Distribution Retrieval using joint dual-polarization radar observation at C- and S- band" could be more suited to the content of the study.

Response: Your suggestion is well taken. Previous dual frequency approaches relied on a much larger separation of frequencies (Ku and Ka bands for GPM for example). Specifying this study is conducted at S and C band is a valuable addition. We will work with the editor to explore the possibility of updating the title.

b)    About DSD parametric forms. The Marshall-Palmer is a particular case of a 2-parameter exponential DSD. The exponential was used before the Gamma become successful. It should be noticed that gamma is a model that has its own limitations in describing some natural DSD.

Response: In the revised manuscript, we will include a comprehensive discussion on the limitations of the Gamma distribution. This will cover its challenges in parametrically modeling certain extreme weather conditions and its shortcomings in accurately fitting smaller drop size classes recorded by disdrometers, particularly due to their limited capability in accounting for small drop sizes. Our discussion will aim to provide a clear understanding of these limitations, ensuring that the Gamma distribution's applicability and boundaries in DSD modeling are well articulated.

c)    In the introduction, authors are right on the need of assumptions on gamma parameters, although the early technique for Gamma DSD retrieval by Gorgucci et al. 2001 (https://doi.org/10.1175/1520-0469(2002)059%3C2373:EORSDP%3E2.0.CO;2) assumes independent gamma parameters. Also it should be pointed out that the dual frequency techniques described apply to applied to Ku-Ka frequencies for quasi vertical observation while dual polarization radar retrieval operates at quasi horizontal elevation angles based on oblateness of drops which is not seen at vertical incidence.

Response: We acknowledge that our initial assumption may have overestimated the readers' familiarity with the physical distinctions influencing observations across different parameters. While it may be obvious to you and the other reviewers, it may not be readily apparent to a more general audience. To address this, we intend to incorporate a concise explanation in the revised manuscript, closely reflecting the nuances you've highlighted regarding the application of gamma parameters and the operational differences between dual frequency and dual polarization radar techniques. This addition aims to clarify the basis for the observed discrepancies in a manner accessible to all readers.

d)    At line 132, authors say that the dataset include light to moderate precipitation. Actually there should be an influence of rain intensity on performance of the retrieval. In fact, in light rain

reflectivites at S- and C-band are not so different and Kdp is similar as well apart from the frequency scaling. In this case the contribution of the C-band freqeuncy should be negligible.

Response: We strongly concur with this observation and have accordingly established our data inclusion criteria to require a minimum reflectivity factor of 25 dBZ. We admit this threshold was chosen without conducting a comprehensive analysis to determine the lowest acceptable limit that would ensure discernible separation in KDP values across different bands, rather, we selected a threshold of 25 dBZ as a conservative measure which could be considered an upper limit for light precipitation.

We hypothesize that lower precipitation cases could still show a benefit from including two radars, due to practical considerations rather than any physical rationale. There were many cases we simply had to discard from our dataset simply due to data quality issues. Including inputs at both frequencies would increase the chances of a higher quality input being present for at least one band which the algorithm could use if the data quality standards were relaxed.

e)    At line 146, authors say that "This range effect is a predictable issue in radar data processing" What is the meaning of this statement ?

Response:  We were referring to attenuation as a function of range. This statement may not even be necessary, however we will update it as follows for clarity.

> *This means that the farther the distance between the radar and the target area, the larger the error in the retrieved DSD. This  attenuation effect is a predictable issue in radar data processing, and it underscores the importance of carefully selecting the range of interest when estimating the DSD.*

f)   Fig 4. The bias in reflectivity profile is a calibration error or is due to difference in elevation, time and so on ?

In this specific instance, we attribute the observed bias to the immediate attenuation of the C-band signal by proximate weather conditions, as illustrated in Figure 3 (of the original manuscript). This attenuation begins at the initial points of the radar beam's path. Conversely, in other scenarios, the C-band and S-band signals exhibit congruent returns up until they encounter weather phenomena at more distant gates, where the C-band signal is perturbed to a greater extent due to its susceptibility to attenuation.

We did take steps to mitigate calibration error. Given that the S-band radar (RCWF) is utilized operationally, while the C-band radar (RCMD) is primarily for research purposes, our expectations regarding the calibration of reflectivity measurements have been accordingly adjusted. This calibration challenge is a key factor in our decision to exclude C-band reflectivity from our algorithm. Instead, we have chosen to rely on KDP, which is less affected by

attenuation and calibration inaccuracies.

g) Fig 8. This is just visual inspection hampered by the linear scale for N. A meaningful comparison should be done in terms of DSD parameters for a meaningful dataset.

We acknowledge that visual inspection alone is insufficient to validate our methodology. To address this, we have compared the improvements in quantitative precipitation estimation (QPE) accuracy, using our method against Z-R derived rain rates, to fulfill the requirement for a statistical evaluation against recognized benchmarks. Nonetheless, we maintain that the subjective comparison of the retrieved DSDs with disdrometer data contributes valuable insights to our study.

Due to modifications in our methodology, which now restricts the solution space to only allow positive values of $\mu$ (previously within the range of $-2 < \mu < 12$), our earlier qualitative plots have become obsolete. The revised example plots, demonstrating gamma distribution parameters for each retrieval, are presented in the subsequent figure. For most cases, employing a linear y-scale has proven to offer a better dynamic range. Therefore, we prefer to maintain consistent scaling across different cases for clarity and comparability

Figure 5

[Figure]

*Figure 5 – Updated Example Plots–* The disdrometer distribution for the closest time to the radar scan is shown in blue as well as the distributions from the previous minute (red) and next minute (green). The gamma parameters of the retrieved distribution are indicated in the accompanying equation and the distribution itself is represented by the black line of each plot.

---

## Author Response (AR1)

Re: Drop Size Distribution Retrieval Using Dual Frequency Polarimetric Weather Radars

Dear Editorial Team of AMT,

We wish to express our gratitude to you and the AMT editorial staff for your efforts and assistance with our submitted manuscript. The constructive feedback and insightful comments received during the public discussion phase have helped in enhancing our manuscript, allowing us to present our methods with greater clarity and depth.

With minimal exceptions, which did not necessitate revisions on our part, we have diligently updated our manuscript to reflect the feedback received. A notable enhancement includes the addition of a comprehensive quantitative analysis of our proposed algorithm. This analysis demonstrates notable advantages over existing benchmarks, a development we were pleased to share with the referees during the public discussion. The undertaking of reprocessing our dataset for this analysis was substantial, and we are hopeful it will meet the reviewers' expectations.

Enclosed, please find our point-by-point response to each comment received, including our replies from the public discussion and the corresponding updates made to our manuscript where relevant.

Additionally, following a suggestion from one of the referees, we propose a title change for the article to "Drop Size Distribution Retrieval Using Dual-Polarization Radar Observations at C-Band and S-Band." We understand this request might pose administrative challenges and wish to proceed only if the change can be easily accommodated.

We eagerly await further guidance and hope our revisions align with the high standards of AMT. Thank you again for the opportunity to refine our work and for considering our submission.

Thank you,

Dan Durbin
Yadong Wang
Pao-Liang Chang

| **Reviewer:** CC1 – Vagner Castro | **Comment Number:** 1 |
|---|---|

**Comment:**

Dear Author,

I found your paper on "The Z and ΦDP fields from both C- and S-band radars for DSD retrieval" to be very interesting and informative. While I understand the overall methodology, I have a question about the specific details of the quality control procedures used for the reflectivity field.

In your paper, you mention that the obtained reflectivity is smoothed with a 4 km smoothing window along each radial direction. I was curious to learn more about how this specific window size was chosen. Were there any specific criteria used to determine this length, or is it a standard practice for this type of analysis"

Sincerely,

Vagner Castro

**Interactive Discussion Response Provided:**

Thank you for your question regarding our manuscript.

In selecting the smoothing window length for our study, we carefully considered the trade-offs between resolution and noise reduction. A longer smoothing window, while beneficial for reducing noise and bias, can adversely affect the resolution, leading to a loss of detail in the radar data. Conversely, a shorter window may not effectively mitigate these noise and bias elements, compromising the quality of the data.

Since there is no standard for filter length with respect to variable smoothing, we referenced the specific differential phase calculation methods utilized by Ryzhkov1 which employ window lengths of 9 gates (2.25 km) and 25 gates (6.25 km). We opted for an approximate averagebetween these two values in our analysis. This decision was based on the observation that within this range, the smoothing effect remains relatively stable against minor variations in the filter length.

However, it's important to highlight that window lengths exceeding 25 gates can lead to oversmoothing, potentially obscuring critical features in the radar data. Similarly, window lengths shorter than 9 gates might not be sufficient to effectively reduce noise. Thus, our chosen average aims to strike a balance, ensuring both adequate noise reduction and preservation of important radar data characteristics. This is easily seen in the following

example from our dataset.

[Figure]

**Manuscript Update:**

No update necessary. A comprehensive response was provided in the public discussion.

| Reviewer: | Comment Number: |
|---|---|
| RC1 | 2 |

**Comment:**

While the abstract states that the retrieval method "is able to accurately retrieve the gamma distribution parameters without the constraints required in previous methods" (line 9), the manuscript does not present any retrieved parameters, nor does it estimate the accuracy of any retrieved parameters. Thus, it is not possible to evaluate the accuracy of the proposed method because the necessary results are not presented in this manuscript.

**Interactive Discussion Response Provided:**

Thank you for highlighting the omission in our manuscript regarding the presentation of retrieved parameters and their accuracy.

To address this, we will update the manuscript to explicitly include following revisions:

a.)     providing the retrieved gamma distribution parameter values for each example in the plot. The updated figures are shown as below. These figures replace the subjective examples in our original manuscript. In these new figures, the disdrometer observations from three continuous minutes are presented using red, blue and green lines, respectively. The retrieved DSD using the radar data of the closet moment is delineated with a black curve.

[Figure]

Following the reviewer's suggestion, we quantitatively evaluated the performance of the proposed approach and look forward to including the results in the revised manuscript. In the quantitative evaluation, the rainfall rates were first estimated using three different approaches:

i.) using the retrieved DSD parameters following equation $R = \frac{\pi}{6} \int_0^{D_{max}} D^3 N(D) v(D) dD$
(Bringi 20002, Zhang 2001, etc);

ii) using the S-band radar reflectivity ($Z$) following the WSR-88D $R$-$Z$ relationship, $Z = 300\,R^{1.4}$ (Ulbrich and Lee 1999);

iii) using the DSD observed by the Parsivel disdrometer following equation $R = \frac{6\,\pi \times 10^4}{\Delta t} \sum_{j=1}^{M} \frac{D_j^3}{S_j^{2DVD}}$ (Raupach and Berne, 2015).

The rainfall rates from i and ii were then compared with the iii, which was treated as the ground truth. In the comparison, the relative absolute error (RAE) was calculated as.
$\epsilon = |R\_d - R| / R\_d$
where R_d are the rainfall rate estimated from the disdrometer as presented in approach iii, and R are the evaluating rainfall rate from approach i or ii, respectively.
Total 167 cases were used in the analysis. The criteria of cases selection are:
1.) time difference between S- and C- band scan is within 1 minutes
2.) only the lowest two elevation angle (0.5o  and 1.4 o ) are used.
3.) reflectivity > 25 dBZ
4.) 25 km < disdrometer range < 70 km

The time series plot presented below illustrates the RAE results for two different approaches. Approach i, our proposed method, is represented by the blue line, while Approach ii, which employs the conventional R(Z) method, is indicated by the red line. The plot demonstrates that estimating rainfall rates using retrieved DSD parameters, as in our proposed approach, yields higher accuracy compared to the traditional Z-R relationship. Specifically, the median RAE for the Z-R approach stands at 0.72, which is notably reduced to 0.53 with our proposed method. This represents a significant improvement of 26.4% as observed in this study.

[Figure]

In the revision, the quantitative evaluation results and discussions will be added.

**Manuscript Update:**

The Performance Evaluation section has undergone major revision in order to address this. Updates were made exactly as were outlined in the interactive discussion response.

Please see updated manuscript Section 3.2 beginning line 321 or redline version line 364.

| Reviewer:
RC1 | Comment Number:
3 |
|---|---|

**Comment:**

2. As presented in this manuscript, the PSO method is essentially a random walk through the three DSD parameter space, and not an "optimization" method. A random walk (line 218) through the parameter space is a valid parameter estimation procedure if the neighborhood of solutions around the global solution are used to estimate an uncertainty of the retrieved parameters.

**Interactive Discussion Response Provided:**

We appreciate your perspective on our use of the PSO method. While we understand your viewpoint that categorizes PSO as a random walk, we would like to clarify that PSO inherently possesses mechanisms that actively drive towards a global minimum, distinguishing it from a purely random walk approach. The intent behind selecting PSO was not to showcase it as a novel optimization technique but rather to utilize a method capable of overcoming the limitations commonly encountered in gradient-based approaches, namely convergence to local minima.

In earlier iterations of our work, we employed the traditional non-linear optimization method, such as Gauss-Newton (GN) algorithm and Levenberg-Marquardt (LM) algorithm. However, this type of method tend to converge at local minimum instead of global minimum. As an alternative, we tried unsupervised learning approaches such as genetic algorithm (GA) and PSO. They both yielded results superior to those achieved with a GN or LM algorithm. The choice to transition to PSO was primarily driven by its efficiency relative to our initial genetic algorithm code. It's important to note that our primary focus was on the effectiveness of the solution method rather than the optimization routine itself. We believe that several other methods could also be successfully employed for the solver block of our algorithm.

**Manuscript Update:**

No update was necessary for this comment, however a clarification was provided in the interactive discussion.

| Reviewer: | Comment Number: |
|---|---|
| RC1 | 4 |

**Comment:**

3. It appears that the study is based on four radial samples made over two different disdrometers. The manuscript text states that nine (line 245) or ten (line 294) days of data were "used in the performance validation" (line 245), but only four reconstructed raindrop distributions are shown in Figure 9. Since no more retrieved parameters are shown in the manuscript, it must be assumed that those four radials were the only radials that were processed.

**Interactive Discussion Response Provided:**

We agree with regard to the noted deficiencies in our original manuscript, as noted here as well as the first comment. In order to address the absence of a statistical presentation of the data, we have reevaluated our dataset and will include the results shown in the response to comment 1 which includes the retrievals of 167 cases. More details please refer to the response to comment 1.

**Manuscript Update:**

The dataset duration was corrected (line 145, redline line 157). Additional cases have been provided after reprocessing the dataset (Figure 8). A comprehensive quantitative assessment of 167 cases have also been added. Please see updated manuscript Section 3.2 beginning line 320 or redline version line 366.

| Reviewer: | Comment Number: |
|---|---|
| RC1 | 5 |

**Comment:**

4. It is major limitation of this work that retrieved DSD parameters are not presented in the manuscript. The four reconstructed raindrop distributions shown in Figure 9 do not show the retrieved slope parameter (lambda) nor the shape parameter (mu). Since three of the four constructed distributions (Fig. 8 a, b, and d) do not have the number concentration approaching zero at zero diameter, this suggests that the retrieved mu value is less than zero. It is unusual for the shape parameter to be negative in scanning radar retrievals due to the large scanning radar sample volume. Which raises concern about the validity of the retrievals. The manuscript must show retrieved parameter statistics to evaluate the performance of the retrieval method.

**Interactive Discussion Response Provided:**

We had originally restricted our parameters along the bounds shown in Zhang's (2001) work, which allowed $\mu$ to be as low as negative 2. We have updated the constraint on $\mu$ to be no less than zero. We believe the trade off in possibly losing a few distributions with negative $\mu$ shown in Zhang is acceptable with the gained benefit of guarding against non-sensical distributions. We thank you for that note, and all cases have been reprocessed with the updated bounds. Additionally, we have included values of $\mu$, $\Lambda$, and $N\_0$ in each subjective plot. Please see the updated subjective examples shown in response to your first comment.

**Manuscript Update:**

The bounds on mu have been updated (line 192, redline 216) and the dataset has been reprocessed. All gamma parameters have been added to each DSD plot (Figure 8).

| Reviewer: | Comment Number: |
|---|---|
| RC1 | 6 |

**Comment:**

5. How much improvement in rainfall estimation does the proposed method provide compared to using one of the mu-lambda relationships proposed by Zhang et al. (2001), Brandes et al. (2002), or Cao et al. (2010)? If the new method does not produce comparable or better rainfall estimates than the mu-lambda constraints, then will this proposed method be an improvement to QPE? The mu-lambda constraints should be the baseline that the new method should aim to beat. In summary, without the manuscript showing results of the retrieved DSD parameters, it is not possible to evaluate the proposed retrieval method. The proposed method may produce results that are superior to results from imposing a mu-lambda constraint. But, as written, the manuscript does not show the evidence needed to verify the statements made in the manuscript.

**Interactive Discussion Response Provided:**

We acknowledge the oversight of not benchmarking our initial manuscript. In addition to the comparison with the Z-R derived rain rate, we have also conducted an analysis of our algorithm incorporating the μ-Λ relationship outlined in Zhang et al. (2001):

$$\mu = -0.016\, \Lambda^2 + 1.213\Lambda - 1.957$$

Different from the proposed method that retrieves three DSD parameters independently, this approach retrieves the DSD with above μ-Λ relationship. The PSO method is also used in this approach.

The following figure shows the rain rate derived from the proposed method's retrieval (blue), the disdrometer truth (black), Z-R relation (red), and the PSO retrieval with a μ-Λ constraint (green).

[Figure]

The accuracy of the μ-Λ constrained retrieval method is found to be comparable to that of our unconstrained retrieval technique. Evaluating performance with the RAE metric reveals that the constrained approach has a median error of 0.55, closely aligning with the unconstrained method's median error of 0.53. While a mean RAE evaluation might suggest enhanced performance with the mu-Lambda relationship, we believe using median values for assessment provides a truer reflection of performance, particularly due to the influence of outliers. To illustrate this, the attached figure displays error plots for all methods, with outliers truncated at a value of 8.

**Manuscript Update:**
The Performance Evaluation section has undergone major revision in order to address this. Updates were made exactly as were outlined in the interactive discussion response.

Please see updated manuscript Section 3.2 beginning line 320 or redline version line 366.

| Reviewer: | Comment Number: |
|---|---|
| RC2 | 7 |

**Comment:**
1)      The authors should consider and discuss height difference between radar gates selected for the DSD retrieval and the disdrometers. The height of the radars are quite high.

**Interactive Discussion Response Provided:**
We acknowledge that the difference between radar volume and the disdrometer observation poses challenges in all radar-based retrieval algorithms.  In our study, we have made concerted efforts to minimize the impact of this discrepancy. This was achieved by restricting our data collection to the lowest two radar elevation angles and confining the range of our retrievals to within 70 km.
While these measures represent the best possible approach given the fixed positions of the equipment, we understand the importance of thoroughly discussing these inherent limitations. We will ensure that a comprehensive discussion of these height differences and their associated constraints is included in the revised version of our manuscript, providing a clear and honest assessment of the limitations of our study.

**Manuscript Update:**
The equipment elevations have been added in a table (Table 2). A discussion on height as an error source including our mitigation approach has been included in the revision (line 114, redline 124).

| Reviewer: RC2 | Comment Number: 8 |
|---|---|

**Comment:**

2)      It is better to briefly introduce the synoptic scale weather conditions for the cases you selected, especially about wind direction and speed, temperature, and relative humility. Evaporation, collision-coalescence will affect the DSD measured at disdrometer station as the raindrops fall from the height of "terminal gate".

**Interactive Discussion Response Provided:**

Thank you for pointing this out. The majority of our samples were collected in June 2017, during the "Meiyu" rain season, a period characterized by varying weather conditions from light drizzle to thunderstorms. The ground temperature is around 25o C, and normally no strong wind is associated with the rainfall during the Meiyu season. To simplify the model, no evaporation, collision-coalescence are considered in this work. We will convey this information in the revised manuscript.

**Manuscript Update:**

The general conditions corresponding to the data collection period have been added to the revision (line 145, redline 159).

| Reviewer: | Comment Number: |
|---|---|
| RC2 | 9 |

**Comment:**

3)     I suggest performing the DSD retrieval at the terminal gate continuously for at least several radar scan to check robustness of the proposed method.

**Interactive Discussion Response Provided:**

Thank you for the suggestion. In the revision, multiple scans were used in the performance evaluation, although not continuously due to the different VCPs of the radar and our time synchronization requirements.

We quantitatively evaluated the performance of the proposed approach and look forward to including the results in the revised manuscript. In the quantitative evaluation, the rainfall rates were first estimated using three different approaches:

i.) using the retrieved DSD parameters following equation $R = \frac{\pi}{6} \int_0^{D_{max}} D^3 N(D) v(D) dD$ (Bringi 20002, Zhang 2001, etc);

ii) using the S-band radar reflectivity ($Z$) following the WSR-88D $R$-$Z$ relationship, $Z = 300\ R^{1.4}$ (Ulbrich and Lee 1999);

iii) using the DSD observed by the Parsivel disdrometer following equation $R = \frac{6\ \pi \times 10^4}{\Delta t} \sum_{j=1}^{M} \frac{D_j^3}{S_j^{2DVD}}$ (Raupach and Berne, 2015).

ii) using the S-band radar reflectivity (Z) following the WSR-88D R-Z relationship, Z=300 R^1.4 (Ulbrich and Lee 1999);

iii) using the DSD observed by the Parsivel disdrometer following equation $R = \frac{6\ \pi \times 10^4}{\Delta t} \sum_{j=1}^{M} \frac{D_j^3}{S_j^{2DVD}}$ (Raupach and Berne, 2015).

The rainfall rates from i and ii were then compared with the iii, which was treated as the ground truth. In the comparison, the relative absolute error (RAE) was calculated as.

$$\epsilon = \frac{|R_d - R|}{R_d}$$

where R_d are the rainfall rate estimated from the disdrometer as presented in approach iii, and R are the evaluating rainfall rate from approach i or ii, respectively.

Total 167 cases were used in the analysis. The criteria of cases selection are:

1.) time difference between S- and C- band scan is within 1 minutes
2.) only the lowest two elevation angle ($0.5^o$ and $1.4^o$) are used.
3.) reflectivity > 25 dBZ
4.) 25 km < disdrometer range < 70 km

The time series plot presented below illustrates the RAE results for two different approaches. Approach i, our proposed method, is represented by the blue line, while Approach ii, which employs the conventional R(Z) method, is indicated by the red line. The plot demonstrates that

estimating rainfall rates using retrieved DSD parameters, as in our proposed approach, yields higher accuracy compared to the traditional Z-R relationship. Specifically, the median RAE for the Z-R approach stands at 0.72, which is notably reduced to 0.53 with our proposed method. This represents a significant improvement of 26.4% as observed in this study.

[Figure]

In the revision, the quantitative evaluation results and discussions will be added.

**Manuscript Update:**
Multiple back-to-back scans at the terminal gate are not present in our dataset due to the tight time synchronization requirements and data quality standards, however robustness has been included in the re-evaluation of the dataset. Please see updated manuscript Section 3.2 beginning line 320 or redline version line 366.

| Reviewer: | Comment Number: |
|---|---|
| RC2 | 10 |

**Comment:**
4)      Unify the variable names such as ZSmeasured, KDP ,measured (Eq.9), ZS', KDPS' etc. in the text and figures.

**Interactive Discussion Response Provided:**
We will ensure all variables are properly defined and standardized in the revised manuscript.

**Manuscript Update:**
Variables have been updated throughout text.

| Reviewer: RC2 | Comment Number: 11 |
| --- | --- |
| **Comment:** | |
| Line 5: What is the "assumptions of the collected data"? You should be more specific here. | |
| **Interactive Discussion Response Provided:** | |
| We were referencing the mu-Lambda constraint which can vary depending on location and meteorological conditions. We will rewrite this to be explicit. | |
| **Manuscript Update:** | |
| The assumptions have been added. Line 5, redline 5. | |

| Reviewer: RC2 | Comment Number: 12 |
| --- | --- |
| **Comment:** | |
| Line 37: Delete word "remotely". | |
| **Interactive Discussion Response Provided:** | |
| "Extensive research has been conducted to  estimate the drop size distribution (DSD), with many studies utilizing measurements taken at two frequencies." | |
| **Manuscript Update:** | |
| The word "remotely" has been deleted. Line 39, redline 43. | |

| Reviewer: RC2 | Comment Number: 13 |
| --- | --- |
| **Comment:** | |
| Line 60: Is the "phase" differential phase? | |
| **Interactive Discussion Response Provided:** | |
| "This approach allows for the determination of DSD solutions that accurately represent the input reflectivity and *specific differential* phase information." | |
| **Manuscript Update:** | |
| The text has been updated to note the reference was "specific differential phase." Line 63, redline 70. | |

| Reviewer: RC2 | Comment Number: 14 |
| --- | --- |
| **Comment:** | |
| Line 61-71: The authors only depict the contents in section 2. What about section 3 and 4? | |
| **Interactive Discussion Response Provided:** | |
| We will rectify this omission by including a brief synopsis of the additional two sections in the introductory material. | |
| **Manuscript Update:** | |
| Sections 3 and 4 have been added to the paragraph outlining the paper. Line 65, redline 76. | |

| Reviewer: RC2 | Comment Number: 15 |
|---|---|

**Comment:**
Line 74: Suggest changing the sentence to "the measurements of the two co-located polarimetric radars".

**Interactive Discussion Response Provided:**
"In the current work, the measurements of the two co-located polarimetric radars, RCWF (S-band) and RCMD (C-band), are used in the algorithm development and validation."

**Manuscript Update:**
The sentence has been updated per the comment. Line 78, redline 87.

| Reviewer: RC2 | Comment Number: 16 |
|---|---|

**Comment:**
Line 93: Add year after "Jaffrain et al."

**Interactive Discussion Response Provided:**
"The findings of Jaffrain et al. (2011) demonstrated that sampling uncertainty"

**Manuscript Update:**
The year has been added per the comment. Line 97, redline 106.

| Reviewer: RC2 | Comment Number: 17 |
|---|---|

**Comment:**
Line 98: Change "such and" to "such as".

**Interactive Discussion Response Provided:**
"Other factors, such as and the angle of the drop trajectory, coincidentally observed particles, and particles that intersect with only the edge will also lead to biases."

**Manuscript Update:**
The sentence has been updated per the commenter's suggestion. Line 102, redline 111.

| Reviewer: RC2 | Comment Number: 18 |
|---|---|

**Comment:**
Line 104: The description of the range rings "two rings with ranges of 20 km and 70 km are shown in Figure 2" is different from the one the Fig.2. Please make sure they are consistent.

**Interactive Discussion Response Provided:**
After checking the code to generate the plot, the range rings correspond to the ranges in the body of the text, and it is the "100 km" reference in the caption that is in error. Thank you for highlighting this. We will update it in the revised manuscript

**Manuscript Update:**
The caption has been updated to reflect the correct value of the range ring. Figure 2.

| Reviewer: | Comment Number: |
|---|---|
| RC2 | 19 |
| **Comment:** | |
| Line 152: What is "ZS"? reflectivity of S-band?  Change "dB" to "dBZ". | |
| **Interactive Discussion Response Provided:** | |
| "ZS > 25 dBZ" | |
| **Manuscript Update:** | |
| The unit has been corrected per the suggestion. Line 163, redline 181. | |

| Reviewer: | Comment Number: |
|---|---|
| RC2 | 20 |
| **Comment:** | |
| Line 155: What is "terminal gate"? | |
| **Interactive Discussion Response Provided:** | |
| We later define terminal gate at line 230. We will move the definition (the gate at the disdrometer location) to line 155 where it is first used. | |
| **Manuscript Update:** | |
| The definition of "terminal gate" has been moved to where it is first referenced. Line 166, Redline 185. | |

| Reviewer: | Comment Number: |
|---|---|
| RC2 | 21 |
| **Comment:** | |
| Line 154: Multiplier "2" is missed in front of KDP in Eq.6. Please also check your algorithm if KDP is multiplied by 2. | |
| **Interactive Discussion Response Provided:** | |
| Thank you for highlighting this error. You are correct, and we will update Eq. 6 to correctly include the two-way factor. The algorithm does account for attenuation in both directions. | |
| **Manuscript Update:** | |
| The code has been verified to correctly use the two-way factor. The equation has been corrected in the revision to clarify that the total attenuation is accounted for. Line 209, redline 234. | |

| Reviewer: | Comment Number: |
|---|---|
| RC2 | 22 |
| **Comment:** | |
| Line 195: Delete word "minimum". | |
| **Interactive Discussion Response Provided:** | |
| "the minimum  phi_dp" | |
| **Manuscript Update:** | |
| The word "minimum" has been deleted per the suggestion. Line 207, redline 232. | |

| Reviewer: | Comment Number: |
|---|---|
| RC2 | 23 |

**Comment:**
Line 203: Change to "converge to a solution".

**Interactive Discussion Response Provided:**
"While it can quickly converge on to a solution,  "

**Manuscript Update:**
The phrase has been updated per the suggestion. Line 217, redline 242.

| Reviewer: | Comment Number: |
|---|---|
| RC2 | 24 |

**Comment:**
Line 211: What is the coordinates? Please give more specific description about "The three coordinates".

**Interactive Discussion Response Provided:**
The three coordinates of the solution space are N_0, mu, and Lamda. We will rewrite this sentence to be more specific and make sure we haven't made too far a leap in relating the DSD parameters to a coordinate space.

**Manuscript Update:**
A clarifying statement has been added that complete the connection between the 3 gamma parameters and a coordinate system. Line 224, redline 249.

| Reviewer: | Comment Number: |
|---|---|
| RC2 | 25 |

**Comment:**
Line 207-220: I did not find the PSO method mentioned or applied in the referred article (Zhang et al., 2001). Please give more explanations on the word "particle" used in the PSO.

**Interactive Discussion Response Provided:**

Zhang 2001 was referenced for the range of values used in the solution space (second half of the paragraph). We will find a general reference to include in the revised manuscript that can point the reader to an overview of PSO that is consistent with our application. We do want to refrain from misleading the reader that the use of PSO is a major innovation. We contend that many other optimization approaches would work similarly so long as they can overcome local convergence issues.

**Manuscript Update:**
No update to the manuscript was required for this comment. A clarification was provided in the public discussion.

| Reviewer: | Comment Number: |
|---|---|
| RC2 | 26 |

**Comment:**

Line 215-222:   Are the parameters alpha and beta the same in Eq. 7 and 8?  What does index "i" stand for in Eq.8?

**Interactive Discussion Response Provided:**

These parameters are different and we should have used different variables in our initial submission to avoid any possible confusion. They are used as weighting factors of the cost function in Eq. 7 and as local/global convergence factors in Eq. 8. We will update this in the revised manuscript.

"i" in Eq. 8 refers to one iteration of the optimization routine as applied to a single gate. We will more thoroughly describe these definitions in the revised manuscript.

**Manuscript Update:**

The coefficients have been updated to remove any confusion. Some additional clarification has also been provided in the revision concerning subscripts of the variables.  Lines 226 to 240, redines 251-265.

| Reviewer: | Comment Number: |
|---|---|
| RC2 | 27 |

**Comment:**

Line 226-238: Based on my understanding, the parameters of DSD can be retrieved at each radar gate individually. If just used measured ZS with less attenuation, kDPS, and kDPC at "terminal gate" to retrieve directly instead of the way described here, what difference will make on retrieved DSD?

**Interactive Discussion Response Provided:**

We retrieved the DSDs for the gates between the radar and the terminal gate solely to more accurately calculate the attenuation factor (also a T-matrix calculation using the DSD) at each position and correct for the accumulated attenuation for the final retrieval. While this may not be as important at S-band, it does become a necessary step for the disdrometer at station 466950 which is located near our 70 km limit.

**Manuscript Update:**

No update to the manuscript was required for this comment. A clarification was provided in the public discussion.

| **Reviewer:** RC2 | **Comment Number:** 28 |
|---|---|
| **Comment:** Line 246-251: Reorganize this part. Please also give the heights of radar "terminal gate" and the disdrometer stations and label them in Fig. 3. | |
| **Interactive Discussion Response Provided:** We will reorganize these lines to properly reference Fig 8 before beginning to talk about it. For the radar and disdrometer heights we would prefer to construct an additional table that will include the data you are requesting. This may be a cleaner approach than including it in the Fig. 3. | |
| **Manuscript Update:** We have addressed this comment by adding a table of our equipment positions. Table 2. | |

| **Reviewer:** RC2 | **Comment Number:** 29 |
|---|---|
| **Comment:** Line 253-254: Rewrite this sentence. | |
| **Interactive Discussion Response Provided:**  First, the radar cross section, determined by the physical cross section, is inherently smaller for these drops, thereby exerting a lesser impact on both reflectivity and phase inputs compared to larger drops. | |
| **Manuscript Update:** We have completely removed this sentence. Upon reflection, we realize it is not necessary. We have retained the discussion on the importance of fitting various regions of the DSD. Beginning line 282, redline 321. | |

| **Reviewer:** RC2 | **Comment Number:** 30 |
|---|---|
| **Comment:** Line 278-284: The "blue/red circles" and "blue/red triangles" are not found in Fig. 10. | |
| **Interactive Discussion Response Provided:** We had simplified the plot, but failed to update the manuscript body. Thank you for catching this. We will revise to reference the proper plot symbology. | |
| **Manuscript Update:** We have corrected the text to correctly explain the plot symbology. Line 313-319, redlines 358-361. | |

| Reviewer: RC2 | Comment Number: 31 |
|---|---|

**Comment:**
Line 283-284: What is the "correction factor"? Please explain how do you "predict" what atmospheric effect.

**Interactive Discussion Response Provided:**
We will rewrite this section to replace "correction factor" with "attenuation factor" or similar. As stated in response to your insightful comment on Line 226-238, the attenuation is calculated for the intermediate positions using their retrieved DSDs and the T-matrix method.

**Manuscript Update:**
The term "correction factor" has been replaced with "attenuation factor." How the attenuation is calculated is already adequately addressed. Line 317, redline 363.

| Reviewer: RC2 | Comment Number: 32 |
|---|---|

**Comment:**
Fig. 2: Suggest change the sentence in the caption to "radii are drawn around the radar location."
In addition, delete "to indicate the data viability region."

**Interactive Discussion Response Provided:**

Figure 2. Taiwan - Instrument Locations. Radii (25 km and 70 km) are drawn around the radar location. Measurement stations with disdrometers are shown in red. Terrain height is indicated in grayscale throughout the map for reference.

**Manuscript Update:**
The caption has been updated per the suggestion. Figure 2.

| Reviewer: | Comment Number: |
|---|---|
| RC2 | 33 |
| **Comment:** | |
| Fig.3 and 4: At what elevation angle and time? | |
| **Interactive Discussion Response Provided:** | |
| These all correspond to 0.5 degrees elevation. We will note this in the caption. The time is contained in the title of each subplot but may be missed. We will add "UTC" after each HHMMSS timetag to clarify. | |
| **Manuscript Update:** | |
| The excluded data has been added to the figures and the time tags now include "UTC" to avoid any confusion. Figures 3 and 8. | |

| Reviewer: | Comment Number: |
|---|---|
| RC2 | 34 |
| **Comment:** | |
| Fig. 5: Please define the ZS', AS', etc. in the caption. | |
| **Interactive Discussion Response Provided:** | |
| We will ensure all variables are clearly defined near their use when we edit the manuscript in response to your general comment 4. | |
| **Manuscript Update:** | |
| Captions for Figures 5 and 6 have been updated to note which variables correspond to calculated vs. measured values. | |

| Reviewer: | Comment Number: |
|---|---|
| RC2 | 35 |
| **Comment:** | |
| Fig. 6: The ZSh, kSDP, etc. in the dash line box should calculated. Please do not mix up with measured ZSh, kSDP, etc. on the right side of figure. | |
| **Interactive Discussion Response Provided:** | |
| We will revise the figure to explicitly state the values on the left side of the figures are calculated values and not the measured observed values. | |
| **Manuscript Update:** | |
| Captions for Figures 5 and 6 have been updated to note which variables correspond to calculated vs. measured values. | |

| Reviewer: | Comment Number: |
|---|---|
| RC3 | 36 |

**Comment:**

Usually, algorithms are first investigated in ideal conditions (i.e. through T-matrix simulated measurements) and then the performance is investigated in the presence of different error sources (i.e. calibration, attenuation, spatial gradient, difference in volume sampling, etc.) and finally validated with real measurement and comparison with real data

**Interactive Discussion Response Provided:**

As the reviewer suggested, we did study the possibility of using co-located dual-polarization radars to retrieve the DSD through simulation as the first step. In the simulation, we first calculated the backward/forward scattering amplitude using T-matrix method, and then calculated the reflectivity and specific differential phase fields from S-band and C-band for a given DSD assumption. The simulated results are shown as follows for the gamma model using various values of μ and Λ with $N_0$=8000.

Figure 1

[Figure]

*Figure 1 –Simulated Reflectivity For Various Parameters –* The S and C-band returns are too similar to provide independent information for retrieval purposes.

Figure 2

[Figure]

*Figure 2 –KDP Variation For Various Parameters* – The S and C-band returns very in KDP values which is a key premise of the algorithm. KDP values separate at the two frequencies at low and moderate values of Λ. The high values of Λ do not provide useful separation as can be inferred by such a quickly collapsing distribution.

Figure 3

[Figure]

*Figure 3 – Attenuation Variation For Various Parameters* – As expected, the attenuation is much greater at C-band relative to S-band. Large separation in attenuation values are present for identical gamma parameters.

The simulations reveal a pronounced attenuation in the C-band reflectivity field, whereas the attenuation observed in the S-band reflectivity is significantly less pronounced. Furthermore, a clear separation of KDP values is present at each band. These findings underpin our selection of these specific variables and illustrate the methods through which they can be leveraged for retrieval processes.

Following the reviewer's suggestion, we will add simulation results into the revised manuscript.

**Manuscript Update:**
A simulation and resulting discussion have been added to the manuscript per the reviewer's suggestion. Figure 9, Line 290-299, relines 333-342.

| Reviewer:
RC3 | Comment Number:
37 |
|---|---|

**Comment:**

In the first steps of the development missing is a thorough analysis of the impact of the error sources on the retrieval technique.

**Interactive Discussion Response Provided:**

We appreciate the reviewer's insightful observation regarding the absence of a comprehensive analysis on the impact of error sources on our retrieval technique. We acknowledge the oversight and recognize the importance of discussing the sources of error in our manuscript.

The primary sources of error stem from four aspects:

1. Observation Bias: The inherent discrepancy between radar (volume observation) and disdrometer (point observation) can introduce biases. This discrepancy arises because radar observes a larger volume, whereas disdrometers measure at a specific point, leading to potential inconsistencies between the two measurements.
2. Vertical DSD Variations: Given that radar volumes are situated several hundred meters above the ground, discrepancies in DSDs observed by radar compared to those by disdrometers are anticipated. This issue represents a significant challenge in all radar-based DSD retrieval methods. To mitigate this, our dataset was limited to the two lowest elevation tilts of radar scans, although this approach only partially addresses the error source. This is the extent to which we can control any possible error from this source given the fixed locations of our equipment.
3. Measurement Errors: Errors from both radar and disdrometer measurements were considered. These are discussed in detail in the original manuscript.
4. Retrieval Error: A certain degree of preprocessing is required to remove noise in the data. While this is an essential step, it carries the possibility of smoothing out local phenomena which could introduce error. There is also a potential for error if the retrieval method converges to a local minimum rather than the global minimum, although we utilized an algorithm which is more likely to overcome this possibility.

In response to the reviewer's comments, we will incorporate a detailed discussion of these error sources in the revised manuscript to provide a clearer understanding of their impact on our retrieval technique.

**Manuscript Update:**

The section on sources of error has been revised to include the additional information we included in the public discussion. Lines 94-119, redlines 110-129.

| Reviewer: | Comment Number: |
|---|---|
| RC3 | 38 |

**Comment:**

Only a very limited comparison with a Parsivel is shown that could be influenced by many sources of errors, including the limitation of the Parsivel itself (authors honestly use "qualitatively" word). However, even for this case, the error associated with the retrieval is not provided, and the range of variability of retrieved data is not provided as well.

Moreover, the technique is not checked against techniques based on single frequency dual-polarization radar that can be found in the literature so that a reader can understand the advantage of having two radars insted of a cheaper single radar.

The text should be revised, since is not very clear on several parts.

**Interactive Discussion Response Provided:**

The main omission and criticism you have outlined is consistent with that of the other reviewers- the lack of a comprehensive performance assessment against an acceptable benchmark. This has prompted us to significantly expand the scope of our study.

To address this, we will update the manuscript to explicitly include following revision:

We quantitatively evaluated the performance of the proposed approach and look forward to including the results in the revised manuscript. In the quantitative evaluation, the rainfall rates were firstly estimated using three different approaches: i.) using the retrieved DSD parameters following equation $R = \frac{\pi}{6}\int_0^{Dmax} D^3 N(D)v(D)dD$ (Bringi 20002, Zhang 2001, etc); ii) using the S-band radar reflectivity ($Z$) following the WSR-88D $R$-$Z$ relationship, $Z = 300\ R^{1.4}$ (Ulbrich and Lee 1999) and iii) using the DSD observed by the Parsivel disdrometer following equation $R = \frac{6\ \pi \times 10^4}{\Delta t}\sum_{j=1}^{M}\frac{D_j^3}{S_j^{2DVD}}$ (Raupach and Berne, 2015). The rainfall rates from i and ii were then compared with the iii, which was treated as the ground truth. In the comparison, the relative absolute error (RAE) was calculated as.

$$\epsilon = \frac{|R_d - R|}{R_d}$$

where $R_d$ and R are the rainfall rate from iii and i/ii, respectively.

Total 167 cases were used in the analysis. The criteria of cases selection are:

1.) time difference between S- and C- band scan is within 1 minutes

2.) only the lowest two elevation angle (0.5° and 1.4° ) are used.

3.) reflectivity>25

4.) 25<disdrometer range<70 km

The time series plot presented below illustrates the RAE results for two different approaches. Approach i, our proposed method, is represented by the blue line, while Approach ii, which employs the conventional R(Z) method, is indicated by the red line. The plot demonstrates that estimating rainfall rates using retrieved DSD parameters, as in our proposed approach, yields higher accuracy compared to the traditional Z-R relationship. Specifically, the median RAE

for the Z-R approach stands at 0.72, which is notably reduced to 0.53 with our proposed method. This represents a significant improvement of 26.4% as observed in this study.

[Figure]

In the revision, the quantitative evaluation results and discussions will be added.

**Manuscript Update:**

The Performance Evaluation section has undergone major revision in order to address this. Updates were made exactly as were outlined in the interactive discussion response.

Please see updated manuscript Section 3.2 beginning line 320 or redline version line 366.

| Reviewer: | Comment Number: |
|---|---|
| RC3 | 39 |

**Comment:**

a)   A title like "Drop Size Distribution Retrieval using joint dual-polarization radar observation at C- and S- band" could be more suited to the content of the study.

**Interactive Discussion Response Provided:**

Your suggestion is well taken. Previous dual frequency approaches relied on a much larger separation of frequencies (Ku and Ka bands for GPM for example). Specifying this study is conducted at S and C band is a valuable addition. We will work with the editor to explore the possibility of updating the title.

**Manuscript Update:**

We have proposed an updated title but realize there may be limitations in updating it at this point in the peer review process. The title has been updated to "Drop Size Distribution Retrieval Using Dual-Polarization Radar Observations at C-Band and S-Band" on the front page.

| Reviewer: | Comment Number: |
|---|---|
| RC3 | 40 |

**Comment:**

b)    About DSD parametric forms. The Marshall-Palmer is a particular case of a 2-parameter exponential DSD. The exponential was used before the Gamma become successful. It should be noticed that gamma is a model that has its own limitations in describing some natural DSD.

**Interactive Discussion Response Provided:**

In the revised manuscript, we will include a comprehensive discussion on the limitations of the Gamma distribution. This will cover its challenges in parametrically modeling certain extreme weather conditions and its shortcomings in accurately fitting smaller drop size classes recorded by disdrometers, particularly due to their limited capability in accounting for small drop sizes. Our discussion will aim to provide a clear understanding of these limitations, ensuring that the Gamma distribution's applicability and boundaries in DSD modeling are well articulated.

**Manuscript Update:**

We believe we have adequately addressed the main drawback of the gamma distribution our work addresses - the need for additional parameterization. However, we have added an additional note on the ability for gamma models to accurately account for small diameters. Line 35, redline 41.

| Reviewer: | Comment Number: |
|---|---|
| RC3 | 41 |

**Comment:**

c)    In the introduction, authors are right on the need of assumptions on gamma parameters, although the early technique for Gamma DSD retrieval by Gorgucci et al. 2001 (https://doi.org/10.1175/1520-0469(2002)059%3C2373:EORSDP%3E2.0.CO;2) assumes independent gamma parameters. Also it should be pointed out that the dual frequency techniques described apply to applied to Ku-Ka frequencies for quasi vertical observation while dual polarization radar retrieval operates at quasi horizontal elevation angles based on oblateness of drops which is not seen at vertical incidence.

**Interactive Discussion Response Provided:**

We acknowledge that our initial assumption may have overestimated the readers' familiarity with the physical distinctions influencing observations across different parameters. While it may be obvious to you and the other reviewers, it may not be readily apparent to a more general audience. To address this, we intend to incorporate a concise explanation in the revised manuscript, closely reflecting the nuances you've highlighted regarding the application of gamma parameters and the operational differences between dual frequency and dual polarization radar techniques. This addition aims to clarify the basis for the observed discrepancies in a manner accessible to all readers.

**Manuscript Update:**

We have updated the manuscript to include the physical reasons for the information orthogonal polarizations provides. Line 53-55, redlines 58-61.

| Reviewer: | Comment Number: |
|---|---|
| RC3 | 42 |

**Comment:**

d)    At line 132, authors say that the dataset include light to moderate precipitation. Actually there should be an influence of rain intensity on performance of the retrieval. In fact, in light rain reflectivites at S- and C-band are not so different and Kdp is similar as well apart from the frequency scaling. In this case the contribution of the C-band freqeuncy should be negligible.

**Interactive Discussion Response Provided:**

We strongly concur with this observation and have accordingly established our data inclusion criteria to require a minimum reflectivity factor of 25 dBZ. We admit this threshold was chosen without conducting a comprehensive analysis to determine the lowest acceptable limit that would ensure discernible separation in KDP values across different bands, rather, we selected a threshold of 25 dBZ as a conservative measure which could be considered an upper limit for light precipitation.

We hypothesize that lower precipitation cases could still show a benefit from including two radars, due to practical considerations rather than any physical rationale. There were many cases we simply had to discard from our dataset simply due to data quality issues. Including inputs at both frequencies would increase the chances of a higher quality input being present for at least one band which the algorithm could use if the data quality standards were relaxed.

**Manuscript Update:**

No update was necessary for this comment, however a clarification was provided in the interactive discussion.

| Reviewer: | Comment Number: |
|---|---|
| RC3 | 43 |

**Comment:**

e)    At line 146, authors say that "This range effect is a predictable issue in radar data processing" What is the meaning of this statement ?

**Interactive Discussion Response Provided:**

We were referring to attenuation as a function of range. This statement may not even be necessary, however we will update it as follows for clarity.

This means that the farther the distance between the radar and the target area, the larger the error in the retrieved DSD. This range attenuation effect is a predictable issue in radar data processing, and it underscores the importance of carefully selecting the range of interest when estimating the DSD.

**Manuscript Update:**

We have clarified this sentence in the manuscript. Line 158, redline 177.

| Reviewer: | Comment Number: |
|---|---|
| RC3 | 44 |

**Comment:**

f)   Fig 4. The bias in reflectivity profile is a calibration error or is due to difference in elevation, time and so on ?

**Interactive Discussion Response Provided:**

In this specific instance, we attribute the observed bias to the immediate attenuation of the C-band signal by proximate weather conditions, as illustrated in Figure 3 (of the original manuscript). This attenuation begins at the initial points of the radar beam's path. Conversely, in other scenarios, the C-band and S-band signals exhibit congruent returns up until they encounter weather phenomena at more distant gates, where the C-band signal is perturbed to a greater extent due to its susceptibility to attenuation.

We did take steps to mitigate calibration error. Given that the S-band radar (RCWF) is utilized operationally, while the C-band radar (RCMD) is primarily for research purposes, our expectations regarding the calibration of reflectivity measurements have been accordingly adjusted. This calibration challenge is a key factor in our decision to exclude C-band reflectivity from our algorithm. Instead, we have chosen to rely on KDP, which is less affected by attenuation and calibration inaccuracies.

**Manuscript Update:**

No update was necessary for this comment, however a clarification was provided in the interactive discussion.

| Reviewer: | Comment Number: |
|---|---|
| RC3 | 45 |

**Comment:**

g)   Fig 8. This is just visual inspection hampered by the linear scale for N. A meaningful comparison should be done in terms of DSD parameters for a meaningful dataset.

**Interactive Discussion Response Provided:**

We acknowledge that visual inspection alone is insufficient to validate our methodology. To address this, we have compared the improvements in quantitative precipitation estimation (QPE) accuracy, using our method against Z-R derived rain rates, to fulfill the requirement for a statistical evaluation against recognized benchmarks. Nonetheless, we maintain that the subjective comparison of the retrieved DSDs with disdrometer data contributes valuable insights to our study.

Due to modifications in our methodology, which now restricts the solution space to only allow positive values of μ (previously within the range of -2 < μ < 12), our earlier qualitative plots have become obsolete. The revised example plots, demonstrating gamma distribution parameters for each retrieval, are presented in the subsequent figure. For most cases, employing a linear y-scale has proven to offer a better dynamic range. Therefore, we prefer to maintain consistent scaling across different cases for clarity and comparability.

[Figure]

**Manuscript Update:**
We have revised the example plots. Please see Figure 8.

| Reviewer: | Comment Number: |
|---|---|
| CC2-Davide Ori | 46 |

**Comment:**

1) A great emphasis is given to the fact that the proposed method does not require assumptions on the relation between DSD parameters. I would like to point out that this is not very convincing because:

a - assuming a gamma distribution for the DSD is already an assumption by itself. Why not a 4-parameter gamma, a log-normal distribution or perhaps a normalized gamma?

**Interactive Discussion Response Provided:**

Thank you for your insightful observations regarding the assumptions underlying our method. Upon reflection, the current language may overstate the flexibility of our approach. While our method avoids imposing a μ-Λ relationship, we recognize that choosing any specific distribution model, including the gamma distribution, inherently involves assumptions. We appreciate your pointing this out and will carefully revise the manuscript to eliminate any misleading implications about the absence of assumptions.

**Manuscript Update:**

No update was necessary for this comment, however a clarification was provided in the interactive discussion.

| Reviewer: CC2-Davide Ori | Comment Number: 47 |
|---|---|

**Comment:**

b - the parameters of an unnormalized gamma distribution are indeed mathematically co-dependent. As an example, one can see just the measuring units of N0 (which, by the way, have not been written in line 32), those should be 1/mm**(mu). Just by noticing that the measuring units of N0 depend on mu, one can realize that the parameters cannot be independent. Some of the drawbacks of using such a size distribution are discussed Testud 2001 and Illingworth 2002 among others.

**Interactive Discussion Response Provided:**

Response: Regarding the units for N0, we realize this omission and will make the necessary corrections to include them in the manuscript.

As to your question about our choice of the gamma distribution: our decision to utilize the three-parameter gamma distribution was driven by a balance between complexity and interpretability. We found it offers a higher degree of complexity that we can feasibly manage, while also providing intuitively interpretable features. This choice was made after considering alternative distributions and aligns with our goal of maintaining a manageable level of model complexity.

We totally agree with the reviewer that there are some drawbacks of using the gamma DSD. We will present these limitations in the revised manuscript and refer to the past works such as Testud 2001 and Illingworth 2002.

**Manuscript Update:**

We do not believe a major discussion on the general pros and cons of the gamma distribution relative to alternatives is warranted in the paper. Rather, our goal was to present a strategy for estimating DSDs within the framework of using a gamma model. We have however added sentences in the revision stating the limitations of the gamma model beyond the additional parameterization required. Line 35, redline 41.

| Reviewer: | Comment Number: |
|---|---|
| CC2-Davide Ori | 48 |

**Comment:**

2) The rationale for the selection of the used radar parameters is not clear. The term multifrequency radar is usually related to the leveraging of either differential scattering or differential absorption properties of the hydrometeors (see also the cited literature in the introduction), however here reflectivity at S-band is used (because it is considered unaffected by attenuation) and phase shift at S- and C-band. I am not sure if a signal difference in Kdp is to be expected at the S- and C- band at all apart from the expected 1/wavelength scaling for Rayleigh scatterers. Some points:

a - Due to the 1/wavelength scaling C-band Kdp is more sensitive than S-band, but at the same time, it does not contain additional information. This means that the retrieval of 3 parameters DSDs would be again ill-posed. One might also test dropping the least sensitive Kdp information (S-band) and see what happens to the results

**Interactive Discussion Response Provided:**

Thank you for your insightful questions regarding our selection of radar parameters. We acknowledge that our explanation in the manuscript may not have been sufficiently clear and will strive to clarify these points in the revision. We'd like to address this comment from following 2 aspects:

1.) We did not intend to imply that S-band returns are entirely unattenuated; rather, our point was that they are less affected by attenuation compared to C-band returns.

We will correct any misleading language in the manuscript to reflect this more accurately.

2.) We realized that that ideal dual-frequency technology should utilize the frequencies from different scattering region, such as the Ku-band and Ka-band radars used in the GPM. These two co-located S-band and C-band dual-polarization radars provide us unique data to investigate their usage in hydrometeorology study. Within different purposes, one goal is to understand whether S-band and C-band dual-polarization variables such as $Z_{DR}$ and $K_{DP}$ can reveal DSD information. We started this research from simulation, and we found that the differences in $K_{DP}$ between these bands are still significant. In the simulation, we first calculated the forward/backward scattering amplitude using T-matrix method, and then calculated their reflectivity ($Z$), attenuation ($A$), and specific differential phase ($K_{DP}$) using the gamma DSD assumption with $N_0 = 8000$. The simulated results as are demonstrated in Figure 1 of this reply. In this simulation, the value of $\mu$ changes from 0 to 9.5, and three values of $\Lambda$, 2, 4, and 6 are selected. From the plots we can find that both attenuation $A$ and specific differential phase $K_{DP}$ show significant difference from S-band and C-band radar especially for small $\Lambda$ or large $\mu$.

In this work, we also studied the necessity of including C-band variable. We found that the retrieval results become significantly worse if only S-band variables are used.

Based on the above study through simulation and real cases, we believe C-band $K_{DP}$ can provide extra information related to DSD.

Figure 1

[Figure]

*Figure 1 –Simulated Reflectivity For Various Parameters –* The S and C-band returns are too similar to provide independent information for retrieval purposes.

Figure 2

[Figure]

*Figure 2 –KDP Variation For Various Parameters* – The S and C-band returns vary in KDP values which is a key premise of the algorithm. KDP values separate at the two frequencies at low and moderate values of Λ. The high values of Λ do not provide useful separation as can be imagined by such a quickly collapsing distribution.

Figure 3

[Figure]

*Figure 3 – Attenuation Variation For Various Parameters* – As expected, the attenuation is much greater at C-band relative to S-band. Large separation in attenuation values are present for identical gamma parameters.

In the revised manuscript, we will add simulation results with discussion. We will also provide more discussion related to the impact of radar frequencies in DSD retrieval.

**Manuscript Update:**
A simulation and resulting discussion have been added to the manuscript which addresses the comment. Additional clarification has been provided in the public discussion. Figure 9, Line 290-299, relines 333-342.

| Reviewer: | Comment Number: |
|---|---|
| CC2-Davide Ori | 49 |

**Comment:**

b - it is not clear to me why Kdp at two frequencies is used and not some other polarimetric/multifrequency quantity. ZDR is a straightforward example, LDR if available. If one assumes S-band reflectivity to be unattenuated (which is also a core assumption in this study) then one can estimate differential attenuation at the S- and C-band which would also be a nice proxy for the total liquid water content. I would have expected a better explanation of why certain radar variables have been used and not others. Perhaps, one might have conducted a theoretical sensitivity study with T-matrix to identify the best choice of observations to include in the retrieval technique given a climatology of observed DSDs... just some suggestions.

**Interactive Discussion Response Provided:**

This is a very good suggestion. We realized that KDP may not be the optimal variables in the DSD retrieval. However, two reasons limit our selection for other variables. First, LDR is not available for both radars used in this work. Second, through our simulation, we knew that ZDR show more obvious features under different frequencies. However, the S-band radar (RCWF) is operationally calibrated, ensuring reliability in its measurements. In contrast, the C-band radar (RCMD) is used primarily for research, and we had concerns about the accuracy of its values especially the calibration bias. We believe that more uncertainties will be brought into the retrieval results with a questionable variable. That is the reason that we did not use ZDR in this work.   For any future work, we will closely work with the radar engineers to evaluate any ZDR calibration issues, and hopefully could include such a variable as it appears to be very promising.

In the revised manuscript, we will provide more discussions related to the polarimetric radar variable selection.

**Manuscript Update:**

No manuscript updates were made in reference to this comment. While we appreciated the reviewer's suggestion, we explained the limitations that prevented us from using the additional variables in the public discussion.

| Reviewer: | Comment Number: |
|---|---|
| CC2-Davide Ori | 50 |

**Comment:**

3) The issue of comparing ground measurements with radar volumes aloft is not discussed enough. From what I understood multiple parsivels on the ground are used to compare their simulated radar quantities with radar variables, those parsivels can be up to 70 km away. I did not understand what is the vertical separation between the radar volume and the parsivels. Is this small enough to ensure that the radar and the disdrometers are observing the same DSDs? I assume that different radar elevation angles are used for the comparison with the various disdrometers, is that taken into account in the T-Matrix calculations? if yes, isn't this causing the dataset to be inhomogeneous, what is the effect on the optimization method?

**Interactive Discussion Response Provided:**

Response: We acknowledge the challenge posed by the vertical separation between radar volumes and ground-based measurements, and this is the challenge for all radar-based DSD retrieval and radar-based QPE algorithms. There is no way to really eliminate the difference from these two observations, rather we can only attempt to mitigate it.

In order to mitigate the retrieval error caused by the DSD vertical variation, we added two constrains in our algorithm. First, our analysis was confined to data from the lowest elevation tilt of 0.5 degrees to minimize vertical separation, which in essence is all we can do. However, to enrich our dataset with more cases that align with our strict time synchronization criteria, we expanded our selection to include data from the next lowest elevation tilt of 1.4 degrees. Given the fact that only the lowest two elevation angles are used, the canting angle effects on the calculated radar variables is very limited. Furthermore, we limited our retrievals to distances under 70 km to avoid complications arising from less favorable geometric conditions at greater ranges. We recognize the importance of transparency in addressing this limitation and will ensure a thorough discussion of this aspect in the revised manuscript, emphasizing our efforts to balance data quality with practical constraints.

We will provide more details related to the measurement differences in the revised manuscript.

**Manuscript Update:**

We have included the effects of vertical separation in the discussion on error sources. Lines 104-119, redlines 114-129.

| Reviewer: CC2-Davide Ori | Comment Number: 51 |
|---|---|

**Comment:**

3. ) What is the effect of the 4km averaging window, isn't this causing the radar quantities to be affected by returns that are up to 4 km away? By judging from figure 4 it seems that the averaging window is not applied as a moving average but rather at discrete points every 4km, does this mean that the bin center of the radar range can be up to 2 km away from the disdrometer position?

**Interactive Discussion Response Provided:**

The implementation of a 4 km averaging window in our analysis was a deliberate decision. We are aware that this averaging causes radar returns from up to 2 km away to influence the local observation. However, we view this as an acceptable compromise. The nature of both reflectivity and phi_dp is such that they are inherently noisy, necessitating a certain degree of smoothing for meaningful analysis. The choice of a 4 km smoothing length also aligns with our KDP calculation methodology, which utilizes window lengths of 2.25 km and 6.25 km.

To provide further context, and in response to another reviewer's feedback, we have included a figure illustrating the level of noise typically present in the raw reflectivity observations.

Figure 4

[Figure]

*Figure 4 – Parameter Smoothing Example –* An approach that balances smoothing and information preservation is clear.

Figure 4 (of the manuscript) may have led the reader to believe filtering was applied only at 4 km points. The filtering was applied as a moving average, however for the along-beam retrieval, discrete points are chosen to operate on. We will add a brief note to the revised manuscript to avoid any confusion.

**Manuscript Update:**

We have included the effects of vertical separation in the discussion on error sources. Lines 104-119, redlines 114-129.

| Reviewer: | Comment Number: |
|---|---|
| CC2-Davide Ori | 52 |

**Comment:**

4) The metric used to test the DSD retrieval is again not clear. It appears that DSDs are compared only qualitatively. If this is the case I strongly recommend plotting DSD in semilog scale, otherwise it would be extremely difficult to evaluate them. In general, it would be nice to first establish what is the goal in terms of retrieval. Alternatively one might report retrieval errors concerning certain moments of the distribution, for example: total number of droplets, total liquid water content, mean size, and distribution width. To do so, one would need a statistically significant sample of retrieved and observed DSDs, I believe that one reviewer already reported on the lack of that.

**Interactive Discussion Response Provided:**

The main omission and criticism you have outlined is consistent with that of the other reviewers- the lack of a comprehensive performance assessment against an acceptable benchmark. This has prompted us to significantly expand the scope of our study.

To address this, we will update the manuscript to explicitly include following revision:

We quantitatively evaluated the performance of the proposed approach and look forward to including the results in the revised manuscript. In the quantitative evaluation, the rainfall rates were firstly estimated using three different approaches: i.) using the retrieved DSD parameters following equation $R = \frac{\pi}{6} \int_0^{Dmax} D^3 N(D) v(D) dD$ (Bringi 20002, Zhang 2001, etc); ii) using the S-band radar reflectivity ($Z$) following the WSR-88D $R$-$Z$ relationship, $Z = 300 R^{1.4}$ (Ulbrich and Lee 1999) and iii) using the DSD observed by the Parsivel disdrometer following equation $R = \frac{6 \pi \times 10^4}{\Delta t} \sum_{j=1}^{M} \frac{D_j^3}{S_j^{2DVD}}$ (Raupach and Berne, 2015). The rainfall rates from i and ii were then compared with the iii, which was treated as the ground truth. In the comparison, the relative absolute error (RAE) was calculated as.

$$\epsilon = \frac{|R_d - R|}{R_d}$$

where $R_d$ and R are the rainfall rate from iii and i/ii, respectively.

Total 167 cases were used in the analysis. The criteria of cases selection are:

1.) time difference between S- and C- band scan is within 1 minutes
2.) only the lowest two elevation angle (0.5° and 1.4°) are used.
3.) reflectivity>25
4.) 25<disdrometer range<70 km

The time series plot presented in the following figure illustrates the RAE results for two different approaches. Approach i, our proposed method, is represented by the blue line, while Approach ii, which employs the conventional R(Z) method, is indicated by the red line. The plot demonstrates that estimating rainfall rates using retrieved DSD parameters, as in our proposed approach, yields higher accuracy compared to the traditional Z-R relationship. Specifically, the median RAE for the Z-R approach stands at 0.72, which is notably reduced to 0.53 with our proposed method. This represents a significant improvement of 26.4% as observed in this study.

Figure 5

[Figure]

Figure 5 – Retrieval Error Evaluation– The retrieval algorithm's performance evaluated as RAE is shown in blue while the RAE associated with the Z-R derived rainrate is shown in red. Outliers are truncated at 8 in order to maintain a more useful vertical scale in the plot.

 In the revision, the quantitative evaluation results and discussions will be added.

**Manuscript Update:**
The Performance Evaluation section has undergone major revision in order to address this. Updates were made exactly as were outlined in the interactive discussion response.

Please see updated manuscript Section 3.2 beginning line 320 or redline version line 366.

| Reviewer: | Comment Number: |
|---|---|
| CC2-Davide Ori | 53 |

**Comment:**

5) Another implicit assumption (that should, at least be made explicit) is the fact that the T-matrix calculations are perfect and do not carry uncertainties. This requires at least some more details such as the refractive index model used. A better approach would have been to estimate the uncertainties in Z and Kdp given by the choice of refractive index (I believe that the reference temperature of 10 degrees C might not be correct at different altitudes). Furthermore, the assumption of null canting angle is quite extreme. Also when comparing with the disdrometer one might want to take into account the limited resolution and maximum observable size. While the integrals of equations 3,4, and 5 go from 0 up to infinity, the parsivel do not (nor natural raindrops), are the integrals truncated at a certain minimum and maximum value? What about the size resolution when computing the integrals?

**Interactive Discussion Response Provided:**

The dielectric constant of water is calculated using the formula proposed by Cole and Cole (1941), and the refractive index is then calculated as the square root of the obtained dielectric constant.

Regarding the temperature assumption for the radar volume, we selected 10 degrees Celsius based on the average ground temperature in Taiwan during June, which is around 25 degrees Celsius, and considering the radar volume's position beneath the melting layer. We acknowledge that temperature varies with altitude, but due to the beam's coverage of multiple altitudes, we determined that a single temperature estimate would be a practical and reasonable approximation for our purposes. We did ensure through T-matrix simulations that the difference in radar parameters is negligible when compared across the temperature range of 0 to 25 degrees.

On the topic of the canting angle, we are grateful for your attention to detail. The T-matrix formulations of the parameters we used contain terms relating to the canting angle that depend on the sine of the cant. This represents a very small value, however a brief note explaining the anticipated error addition in such an assumption should be included in the revised manuscript. Regarding the measurement capabilities of the Parsivel disdrometer, it categorizes drop sizes into 32 bins, ranging from 0.06 to 24.5 mm. For our numerical integrations, we conducted them with a resolution of 0.1 mm, spanning from 0.1 mm to 8 mm. While a narrower range could have been considered, we chose the upper limit of 8 mm as it effectively captures the spectrum of naturally occurring raindrop sizes. We agree that this aspect of the methodology merits a brief discussion in our updated manuscript to provide clarity on our decision-making process and we will update the integrals to state that integration occurs at these Dmin and Dmax values rather than the infinite upper limit.

**Manuscript Update:**

We have provided detailed clarification in the public discussion. As for the revision, we have added bounds on our integrals to show the maximum diameters used and have added substantiation on why the dieletric constant was referenced at 10 deg Celsius. Lines 200-205, redines 221-228.

---

## Referee Report (RR1)

The revised manuscript presents an iterative-based approach to retrieving DSD, reducing a number of conditions, from a unique setup that includes dual frequency dual-pol ground-based radar. In general, the manuscript is well-written and shows a good structure. However, the evaluation section needs to be clearer, some concluding sentences read too strong, and quantitative numbers are lacking.

I would suggest a round of major revisions:

**Section 2.2**

Radar calibration: Add one or two sentences related to the calibration status of the radars (reflectivity) as Z is an input to the PSO algorithm.

Lines 160-165: Are these conditions in range and Z values part of the methodology or is it meant only for validation purposes? Please clarify this in the manuscript and re-arrange accordingly. If these values are part of the methodology (in an operational context), then how should we proceed to have a complete "map" of retrieve DSD? Perhaps this needs to be discussed in the conclusions.

Lines 168-172: Remove it because all these were mentioned before in section 2.2.1

Lines 174-175: Clarify that Z is not yet corrected from attenuation.

Figure 4: Add the raw phidp to compare with the processed phip (FIR filtering). For easy comparison, the initial phase offset can be subtracted.

Lines 181-183: Clarify that Z (S-band) also suffers from attenuation. The way it read now, tell the reader that Z (C-band) is the only one sensitive to attenuation.

Line 230, Eq(7): Clarify if the Cost is estimated gate by gate or per radial, i.e., are the variables Z and phidp single values or vectors?

Eq(9): "four hundred iterations for each retrieval" . This may lead to a large computational time as this retrieval is gate by gate. However, I have not read anything related to this. If this is a burden, then please add a comment about it.

Line 243: "starting from the one nearest to the radar". Is that the case? Because I thought the minimum range is set to 25 km.

Line 245: "once a representative DSD" Is this the same as the best DSD or still under the optimization search. Please clarify it in the text.

On the same line: "the attenuation is calculated". Please add something like .. is calculated based on Equation (5). So far Equation (5) was not indicated yet.

Line 254-255: Replace "This process continues" by "This gate by gate process continues" or similar. Replace "at which point the final retrieval is performed" by "because the range criterion < 70 km" or something similar.

A related question: why the algorithm stops at the location of the disdrometer that is located 70 km from the radar site? One would like to continue till the end of the radial. Please clarify somewhere in the text.

A second related question: What happens to the algorithm when rain starts around 25 km, i.e., when the difference between phidp(C-band) and phidp(S-band) is not sufficient (see line 155). Then DSDs are not retrieved? Clarify this somewhere in the text.

Lines 267-269: Please clarify what is the relation between 10 days and 167 cases, how a case is defined?

**Section 3.1**

Line 278: "The mojority of the cases" … write how many, 5 out of 8?

Line 281: "Deviations can be found…" That is obvious. What reasons would the authors suggest for such deviations?

Figure 8: Add the resulting rainfall rate values from disdrometers (blue line) to each subplot so that we can associate them with a rain intensities.

Lines 290-294 Rewrite to something like this: This simulation involves calculating KDP using equation(4) for varying drop size distributions using a gamma-modeled DSD …

Line 300: Replace "C-band reflectivity excluded" by C-band differential phase excluded.

To strengthen the qualitative assessment. I suggest the following points:

- As the estimation of accurate phidp from raw phidp is not straightforward, especially at C-band,  show some radials of processed phidp (C and S-band) and compare them with optimal DSD-based phidp.
- Attenuation correction, Figure 11: Only showing one single radial is not convincing and difficult to suggest conclusions. Similar to my previous point, show some more radials with diverse Z radial profiles (or a PPI).

- Lines 305-310: It is not consistent with Line 290 ("key question"). I suggest you either i) make a strong analysis to address this question or ii) Rewrite Line 290 and move Equation (9) and Figures 9 and 10 to Appendix.

**Section 3.2**

This sections lack of statistical results. Showing a time series plot is not so relevant in this study. I would suggest using scatterplots and show metrics such as RMSE, Bias, correlation, etc. For instance, a scattering plot related to the attenuation correction: Z(retrieved DSD) vs Z(disdrometer). And two more related to KDP and Rainfall rate.

Lines 345-347: "Deviations from this correlation can mislead the algorithm" These lines are somehow "in the air". I suggest the authors to add some related references for the estimation of KDP that use self-consistency. For example:

- Park, S., V. N. Bringi, V. Chandrasekar, M. Maki, and K. Iwanami, 2005: Correction of Radar Reflectivity and Differential Reflectivity for Rain Attenuation at X Band. Part I: Theoretical and Empirical Basis. *J. Atmos. Oceanic Technol.*, **22**, 1621–1632, https://doi.org/10.1175/JTECH1803.1.
- Giangrande, S. E., McGraw, R., and Lei, L.: An application of linear programming to polarimetric radar differential phase processing, J. Atmos. Ocean. Tech., 30, 1716–1729, 2013.
- Gorgucci, E., Scarchilli, G., and Chandrasekar, V.: Specific Differ- ential Phase Estimation in the Presence of Nonuniform Rainfall Medium along the Path, J. Atmos. Ocean. Tech., 16, 1690–1697, 1999.
- Reinoso-Rondinel, R., Unal, C., and Russchenberg, H.: Adaptive and high-resolution estimation of specific differential phase for polarimetric X-band weather radars, J. Atmos. Ocean. Tech., 35, 555–573, 2018.

**Section 4:**

Discuss more the generalization of the proposed algorithm to retrieve DSD in terms of dual frequency systems and the range constraint ($> 25, < 75$ km).

Clarify if this algorithm can be applied in real-time operations. If not, then what aspects would need further work?

---

## Author Response (AR2)

Re: Drop Size Distribution Retrieval Using Dual-Polarization Radar Observations at C-Band and S-Band

Dear Editorial Team of AMT,

We continue to be grateful for the support of the editorial team and the efforts of the reviewers who have greatly helped improve our work.

Please find our enclosed point-to-point response. We have made substantial revisions to our manuscript to address each comment received.

Thank you,

Dan Durbin
Yadong Wang
Pao-Liang Chang

| Reviewer: | Comment Number: |
|---|---|
| 1 | R1 |

**Comment:**
The authors carefully answered reviewer's questions and revised the manuscript according to the reviwer's comments and suggestions. I suggest they should take a minor revision. Change the term "phase" in their interactive disscussion response and revised manuscript to either "specific differential phase" for KDP or "differential phase" for PhiDP.

**Reply:**
We have updated any generic "phase" references with either "differential phase" or "specific differential phase" as appropriate.

**Manuscript Update:**
Revision:
Line 395
Figure 4

Redline Version:
Line 451
Figure 4

| Reviewer: | Comment Number: |
|---|---|
| 2 | R2-1 |

**Comment:**
Section 2.2: Radar calibration: Add one or two sentences related to the calibration status of the radars (reflectivity) as Z is an input to the PSO algorithm.

**Reply:**
We have added the following sentences with respect to calibration of RCWF (S-band reflectivity source):

"RCWF is an S-band dual-polarimetric radar that is part of Taiwan's operational Multi-Radar-Multi-Sensor QPE system (Chang et al, 2021). To achieve a QPE accuracy within 10 percent, the reflectivity bias must be kept within 1 dBZ. This is accomplished by regularly calibrating RCWF's reflectivity using a self-consistency algorithm (Le Loh et al., 2022). In contrast, RCMD is a C-band radar used experimentally and for research purposes rather than operationally."

**Manuscript Update:**
Revision:
Lines 79-83

Redline Version:
Lines 81-85

| Reviewer: | Comment Number: |
|---|---|
| 2 | R2-2 |

**Comment:**
Section 2.2: Lines 160-165: Are these conditions in range and Z values part of the methodology or is it meant only for validation purposes? Please clarify this in the manuscript and re-arrange accordingly. If these values are part of the methodology (in an operational context), then how should we proceed to have a complete "map" of retrieve DSD? Perhaps this needs to be discussed in the conclusions.

**Reply:**
These conditions are primarily used for validation purposes to ensure we had "good" data to operate on. However, the range limitation would be critical in minimizing vertical separation between the observation volume and ground position as well as minimizing the effects of accumulated error.

We have updated this paragraph accordingly:
"This set of criteria is designed to strike a balance between creating sufficient deviation between C and S-band differential phases for an accurate DSD retrieval, while also preventing excessive error accumulation as well as minimizing the vertical separation between the radar observation volume and ground location. Additionally, a reflectivity threshold of 25 dBZ is imposed to ensure that there is enough observable precipitation in the terminal gate where the disdrometer is located. These criteria are primarily chosen to increase the quality of the data for development and validation, however similar standards would need imposed if the algorithm were to be operationally applied to address the error accumulation and elevation differences."

**Manuscript Update:**
Revision:
Lines 176-181

Redline Version:
Lines 180-185

| Reviewer: | Comment Number: |
|---|---|
| 2 | R2-3 |

**Comment:**
Section 2.2: Lines 168-172: Remove it because all these were mentioned before in section 2.2.1

**Reply:**
We have removed the duplicated content.

**Manuscript Update:**
Revision:
N/A
Redline Version:
Lines 186-193

| **Reviewer:** | **Comment Number:** |
|---|---|
| 2 | R2-4 |
| **Comment:** | |
| Section 2.2: Lines 174-175: Clarify that Z is not yet corrected from attenuation. | |
| **Reply:** | |
| We have updated the line accordingly:

"Figure 4 shows the pre-processed (dashed) and post-processed $Z$ (solid), without any attenuation correction, and $\phi_{DP}$ fields along the yellow arrow of Figure 3, contrasting the difference between the post-processed differential phase profiles (solid) with the profiles prior to unfolding and smoothing (dashed)." | |
| **Manuscript Update:**
Revision:
Lines 142-146

Redline Version:
Lines 145-149 | |

| Reviewer: | Comment Number: |
|---|---|
| 2 | R2-5 |

**Comment:**

Section 2.2: Figure 4: Add the raw phidp to compare with the processed phip (FIR filtering). For easy comparison, the initial phase offset can be subtracted.

**Reply:**

We have updated the figure to add the raw phidp. We have also added the raw reflectivity for completeness.

**Figure 4.** The data along the radial indicated in Figure 3 before and after processing. Raw radar data (dashed lines) are contrasted with post-processed data (solid lines). Processed C-band reflectivity is not used as a retrieval input but is useful for validation purposes as discussed in Section 3.1

**Manuscript Update:**

Revision:
Figure 4

Redline Version:
Figure 4

| Reviewer: | Comment Number: |
|---|---|
| 2 | R2-6 |

**Comment:**
Section 2.2: Lines 181-183: Clarify that Z (S-band) also suffers from attenuation. The way it read now, tell the reader that Z (C-band) is the only one sensitive to attenuation.

**Reply:**
We have stated more clearly that S-band also suffers from attenuation:

"Although S-band reflectivity does experience atmospheric attenuation, $Z^C$ is much more vulnerable to this effect and is therefore not used as an input to the algorithm."

**Manuscript Update:**
Revision:
Line 187

Redline Version:
Line 200

| Reviewer: | Comment Number: |
|---|---|
| 2 | R2-7 |

**Comment:**
Line 230, Eq(7): Clarify if the Cost is estimated gate by gate or per radial, i.e., are the variables Z and phidp single values or vectors?

**Reply:**
We have added the following note to state that the retrieval is for each gate, implying the values are singular rather than vectored:

"For a gate's retrieval, the cost of every particle is calculated, and the iteration's current best solution as well as the global best solution of all iterations are recorded."

**Manuscript Update:**
Revision:
Line 237

Redline Version:
Line 251

| Reviewer: | Comment Number: |
|---|---|
| 2 | R2-8 |

**Comment:**

Section 2.2: Eq(9): "four hundred iterations for each retrieval" . This may lead to a large computational time as this retrieval is gate by gate. However, I have not read anything related to this. If this is a burden, then please add a comment about it.

**Reply:**

For this prototyping phase, minimizing run times was not a priority and the dataset could typically be processed overnight on a desktop computer. Any embedded solution should take advantage of speed improvements gained from parameter tuning or even substituting a more efficient technique in place of PSO.

We have added the following:

"It should be noted that the setting of four hundred iterations is intended solely for prototype algorithm development, and computational efficiency has not yet been addressed. A simple iteration control algorithm could be implemented to terminate the computation once the root mean square error reaches the predefined threshold. Any embedded solution should take advantage of speed improvements gained from parameter tuning or even substituting a more efficient technique in place of PSO."

**Manuscript Update:**

Revision:
Lines 245-249

Redline Version:
Lines 260-264

| Reviewer: | Comment Number: |
|---|---|
| 2 | R2-9 |

**Comment:**
Section 2.2: Line 243: "starting from the one nearest to the radar". Is that the case? Because I thought he minimum range is set to 25 km.

**Reply:**
Thank you for highlighting this point. We acknowledge that the manuscript lacked clarity, which led to the confusion. We have now clarified that the specified criteria pertain to the terminal gate containing the truth data. The retrieval process still begins at the closest gate to accurately account for all attenuation along the beam. The final (destination) gate is at least 25 km from the radar. To address this issue, we have made the following modifications in the manuscript:

"To achieve reasonable results, the following criteria for candidate data are therefore used for the terminal gate of the retrieval which contains the disdrometer:
$\quad$ 25 km< Range< 70 km
$\quad$ $Z^S$> 25 dBZ
This set of criteria is designed to strike a balance between creating sufficient deviation between C and S-band differential phases for an accurate DSD retrieval, while also preventing excessive error accumulation as well as minimizing the vertical separation between the radar observation volume and ground location. Additionally, a reflectivity threshold of 25 dBZ is imposed to ensure that there is enough observable precipitation in the terminal gate where the disdrometer is located. These criteria are primarily chosen to increase the quality of the data for development and validation, however similar standards would need imposed if the algorithm were to be operationally applied to address the error accumulation and elevation differences."

**Manuscript Update:**
Revision:
Line 172
Lines 251-256

Redline Version:
Lines 176-179
Lines 270-275

| Reviewer: | Comment Number: |
|---|---|
| 2 | R2-10 |

**Comment:**
Section 2.2: Line 245: "once a representative DSD" Is this the same as the best DSD or still under the ptimization search. Please clarify it in the text.

**Reply:**
We have removed this ambiguity to clearly state the optimization search has ceased:

"After completing the retrieval for a gate, attenuation is calculated using Equation 5, and the reflectivity for the next farthest gate is adjusted."

**Manuscript Update:**
Revision:
Line 253
Redline Version:
Lines 271

| Reviewer: | Comment Number: |
|---|---|
| 2 | R2-11 |

**Comment:**
Section 2.2: On the same line: "the attenuation is calculated". Please add something like .. Is calculated based on Equation (5). So far Equation (5) was not indicated yet.

**Reply:**
We have updated to include reference to Eq 5:
"After completing the retrieval for a gate, attenuation is calculated using Equation 5, and the reflectivity for the next farthest gate is adjusted."

**Manuscript Update:**
Revision:
Line 254
Redline Version:
Line 272

| Reviewer: | Comment Number: |
|---|---|
| 2 | R2-12 |

**Comment:**
Section 2.2: Line 254-255: Replace "This process continues" by "This gate by gate process continues" or similar. Replace "at which point the final retrieval is performed" by "because the range criterion < 70 km" or something similar.

**Reply:**
We have updated similarly to the suggestion:
"In this manner, each gate's input reflectivity accounts for attenuation experienced between the radar and that gate. This process, illustrated in Figure 7, continues at 4 km intervals until reaching the disdrometer location."

**Manuscript Update:**
Revision:
Lines 255-256

Redline Version:
Lines 273-275

| Reviewer: | Comment Number: |
|---|---|
| 2 | R2-13 |

**Comment:**
Section 2.2: A related question: why the algorithm stops at the location of the disdrometer that is located 70 km from the radar site? One would like to continue till the end of the radial. Please clarify somewhere in the text.

**Reply:**

Thank you for pointing this out. While the algorithm can indeed extend to the end of each radial, we limited the range to within 70 km in this prototype algorithm for two primary reasons:

1) Since the retrieval result from each gate highly depends on the results from previous gates, errors can accumulate up to the final gate. Limiting the range to 70 km helps to effectively mitigate these accumulated errors. The 70 km range was chosen based on the ranges of available disdrometers used for validation.

2) The height of the radar beam increases monotonically along the radar beam. Consequently, at longer distances, the deviation between radar observations (in the air) and disdrometer observations (on the ground) could be significant due to vertical variations in the DSD. For validation purposes, limiting the radar's final gate within a reasonable range helps to minimize differences between retrieval results and disdrometer observations, which serve as the ground truth.

The manuscript has been updated as outlined in response to comment R2-2.

**Manuscript Update:**
Revision:
Lines 176-181

Redline Version:
Lines 180-185

| **Reviewer:** | **Comment Number:** |
|---|---|
| 2 | R2-14 |

**Comment:**
Section 2.2: A second related question: What happens to the algorithm when rain starts around 25 km, i.e., when the difference between phidp(C-band) and phidp(S-band) is not sufficient (see line 155). Then DSDs are not retrieved? Clarify this somewhere in the text.

**Reply:**
Please see reply to R2-13.

**Manuscript Update:**
Revision:
Lines 176-181

Redline Version:
Lines 180-185

| **Reviewer:** | **Comment Number:** |
|---|---|
| 2 | R2-15 |

**Comment:**
Section 2.2: Lines 267-269: Please clarify what is the relation between 10 days and 167 cases, how a case is defined?

**Reply:**
We have updated the manuscript to include the definition of a case (one 360 degree scan of each radar) and specified that only 167 of the cases met our strict synchronization and data quality requirements.

"A potential case is defined as one plan position indicator (PPI) scan of each radar. The stringent criteria for time synchronization narrowed the dataset to 167 cases for time synchronization narrowed the dataset to 167 cases that not only met the synchronization requirements of the radar scans but also had the requisite data quality and fell within the disdrometer range requirement."

**Manuscript Update:**
Revision:
Lines 313-316

Redline Version:
Lines 340-343

| Reviewer: | Comment Number: |
|---|---|
| 2 | R2-16 |

**Comment:**
Section 3.1 Line 278: "The mojority of the cases" … write how many, 5 out of 8?

**Reply:**
We have updated as follows, although it is a matter of opinion since this is still for our subjective comparison:

"Six of these eight cases show a high degree of agreement across the spectrum of drop sizes, with particularly strong correlation observed for the measurements of moderate drop sizes."

**Manuscript Update:**
Revision:
Line 325

Redline Version:
Line 352

| Reviewer: | Comment Number: |
|---|---|
| 2 | R2-17 |

**Comment:**
Section 3.1 Line 281: "Deviations can be found…" That is obvious. What reasons would the authors
suggest for such deviations?

**Reply:**
We have combined this paragraph with the following paragraph which primarily focuses on our rationale for the deviations, namely the relative importance of different drop sizes. This update provides a better flow from the statement to the underlying reasons.

"Deviations can be found in cases such as 20170613 - 08:04:49 and 20180107 - 09:26:32 for smaller drops ($D < 0.5$ mm). This is a predictable result given the retrieval input parameters are heavily dominated by contributions of larger diameters. Not all disdrometer sizes need to be equally prioritized for accurate fitting. Examination of Equations 3 and 4 reveals that the contribution to radar parameters from each diameter increases with the size of the drop. Furthermore, previous research evaluating the accuracy of disdrometers across various drop sizes indicated that drops of 0.6 mm and larger are the first reliably measured sizes by optical disdrometers (Tokay et al., 2001). This finding supports the notion that inaccuracies in measuring smaller drop sizes do not significantly impact the calculations of reflectivity or DSD-derived metrics such as rain rate and attenuation."

**Manuscript Update:**
Revision:
Lines 327-334

Redline Version:
Lines 354-362

| Reviewer: | Comment Number: |
|---|---|
| 2 | R2-18 |

**Comment:**

Section 3.1 Figure 8: Add the resulting rainfall rate values from disdrometers (blue line) to each subplot so that we can associate them with a rain intensities.

**Reply:**

We have updated Figure 8 (now Figure 10) to include rainfall rates.

**Figure 10.** The retrieved DSD is shown in black on each plot with its gamma distribution equation. The DSD of the closest disdrometer record is plotted in blue while the previous record and next record time are shown in red and green, respectively. The median rainfall rate of the three disdrometer collections is indicated in mm/hr.

**Manuscript Update:**

Revision:

Figure 10

Redline Version:

Figure 10

| Reviewer: | Comment Number: |
|---|---|
| 2 | R2-19 |

**Comment:**
Section 3.1 Lines 290-294 Rewrite to something like this: This simulation involves calculating KDP using equation(4) for varying drop size distributions using a gamma-modeled DSD …

**Reply:**
This section has been updated per the reviewer's comments to include which equations/models are being utilized. Additionally, the simulation section has been expanded to include a much more thorough evaluation of the algorithm under simulated ideal conditions.

**Manuscript Update:**
Revision:
Lines 258-301

Redline Version:
Lines 281-328

| Reviewer: | Comment Number: |
|---|---|
| 2 | R2-20 |

**Comment:**
Section 3.1 Line 300: Replace "C-band reflectivity excluded" by C-band differential phase excluded.

**Reply:**
Thank you for catching this error. We have completely removed this example in lieu of a more thorough simulation that addresses one vs. two frequency retrievals.

**Manuscript Update:**
Revision:
Lines 258-301

Redline Version:
Lines 281-328

| Reviewer: | Comment Number: |
|-----------|-----------------|
| 2 | R2-21 |

**Comment:**
Section 3.1 To strengthen the qualitative assessment. I suggest the following points:
• As the estimation of accurate phidp from raw phidp is not straightforward, especially at C-band, show some radials of processed phidp (C and S-band) and compare them with optimal DSD-based phidp.
• Attenuation correction, Figure 11: Only showing one single radial is not convincing and difficult to suggest conclusions. Similar to my previous point, show some more radials with diverse Z radial profiles (or a PPI).
• Lines 305-310: It is not consistent with Line 290 ("key question"). I suggest you either i) make a strong analysis to address this question or ii) Rewrite Line 290 and move Equation (9) and Figures 9 and 10 to Appendix.

**Reply:**
We have addressed the reviewer's comments through the following updates:
1) We updated this figure to include more examples. A PPI plot is not feasible with our processed data, but we have expanded the figure to include 3 examples with the space available.
2) We have included the input and retrieved phi_dp profiles for these examples.
3) We have provided a stronger analysis on one of the "key questions" (one vs two frequency) through a more thorough simulation.

**Manuscript Update:**
1) 3 examples
Revision:
Figure 11
Redline Version:
Figure 11

2) Inclusion of phidp
Revision:
Figure 11
Redline Version:
Figure 11

3) Updated simulation analysis
Revision:
Lines 258-301
Redline Version:
Lines 281-328

| Reviewer: | Comment Number: |
|---|---|
| 2 | R2-22 |

**Comment:**

Section 3.2 This sections lack of statistical results. Showing a time series plot is not so relevant in this
study. I would suggest using scatterplots and show metrics such as RMSE, Bias,
correlation, etc. For instance, a scattering plot related to the attenuation correction:
Z(retrieved DSD) vs Z(disdrometer). And two more related to KDP and Rainfall rate.

**Reply:**

We appreciate the reviewer's suggestions. To address the reviewer's comment, we have updated the manuscript to include scatter plots of retrieved vs. input values for $Z$, $K_{DP}^s$, and $K_{DP}^c$. We have also included the calculated correlation coefficients in the manuscript text. We recognize that the statistical results of rainfall are normally evaluated either in the accumulated (e.g., hourly based or days based) format (e.g. Wang et al. 2013, Zhang et al. 2016, Zhang et al. 2020), or in a relatively large scale of rain rate (Scarchilli et al. 1993). However, based on our current available data, such statistical analysis is not feasible. Currently, we are collecting more data for such a goal, and will present additional statistical results in future work. The time-series data (relative absolute error) has been reformatted in terms of a cumulative error distribution which is much more interpretable.

In the revised manuscript, new figures and discussions are included.

Wang, Y. J. Zhang, A. Ryzhkov, L. Tang, 2013: C-band polarimetric radar QPE based on specific differential propagation phase for extreme typhoon rainfall. JTECH.
https://doi.org/10.1175/JTECH-D-12-00083.1

Jian Zhang, et al., 2016: Multi-Radar Multi-Sensor (MRMS) Quantitative Precipitation Estimation: Initial Operating Capabilitie. BAMS

Gianfranco Scarchilli, et al. 1993: Rainfall estimation using polarimetric techniques at C-band frequencies. JAMC.https://doi.org/10.1175/1520-0450(1990)029<0561:MRTSUC>2.0.CO;2

Jian Zhang, et al. 2020: A dual-polarization radar synthetic QPE for operations.
JHM.https://doi.org/10.1175/JHM-D-19-0194.1

**Figure 12.** The retrieved calculated value of each parameter (y-axis) is shown relative to the algorithm's input (x-axis) based on the radar measurements at the final gate (disdrometer location).

**Figure 13.** The performance of the dual frequency approach (blue circles) shows roughly equivalent performance to the single frequency approach employing a relevant $\mu - \Lambda$ constraint (purple triangles). Both methods utilizing DSD information outperform the region-specific Z-R derived rainrates (black dash).

**Manuscript Update:**

Scatter Plots
Revision:

Figure 12
Redline Version:
Figure 12

Cumulative Error Distribution
Revision:
Figure 13
Redline Version:
Figure 13

| Reviewer: | Comment Number: |
|---|---|
| 2 | R2-23 |

**Comment:**
Section 3.2 Lines 345-347: "Deviations from this correlation can mislead the algorithm" These lines
are somehow "in the air". I suggest the authors to add some related references for the
estimation of KDP that use self-consistency. For example:
• Park, S., V. N. Bringi, V. Chandrasekar, M. Maki, and K. Iwanami, 2005: Correction of
Radar Reflectivity and Differential Reflectivity for Rain Attenuation at X Band. Part I:
Theoretical and Empirical Basis. J. Atmos. Oceanic Technol., 22, 1621–
1632, https://doi.org/10.1175/JTECH1803.1.
• Giangrande, S. E., McGraw, R., and Lei, L.: An application of linear programming to
polarimetric radar differential phase processing, J. Atmos. Ocean. Tech., 30, 1716–1729,
2013.
• Gorgucci, E., Scarchilli, G., and Chandrasekar, V.: Specific Differ- ential Phase
Estimation in the Presence of Nonuniform Rainfall Medium along the Path, J. Atmos.
Ocean. Tech., 16, 1690–1697, 1999.
• Reinoso-Rondinel, R., Unal, C., and Russchenberg, H.: Adaptive and high-resolution
estimation of specific differential phase for polarimetric X-band weather radars, J.
Atmos. Ocean. Tech., 35, 555–573, 2018.

**Reply:**
We have updated the manuscript to include the need for a self-consistency criteria and the
suggested references have been added into the revised manuscript.

"Therefore, incorporating self-consistency relations between the input variables stands out as a
promising direction for enhancing algorithmic accuracy in future research endeavors. (Park et al.,
2005; Giangrande et al., 2013; Gorgucci et al., 1999; Reinoso-Rondinel et al., 2018)"

**Manuscript Update:**
Revision:
Lines 376-381

Redline Version:
Lines 435-436

| Reviewer: | Comment Number: |
|---|---|
| 2 | R2-24 |

**Comment:**
Section 4 Discuss more the generalization of the proposed algorithm to retrieve DSD in terms of
dual
frequency systems and the range constraint (> 25, < 75 km).

**Reply:**
Please see reply to RC2-2.

**Manuscript Update:**
N/A

| Reviewer: | Comment Number: |
|---|---|
| 2 | R2-25 |

**Comment:**
Section 4 Clarify if this algorithm can be applied in real-time operations. If not, then what aspects would need further work?

**Reply:**

Yes, this algorithm is developed to applied in real-time operation. We clarify this and point out some proposed improvements for doing such in the conclusion.

"The optimization approach involves various adjustable parameters, including swarm size, number of iterations, and acceleration coefficients. Fine-tuning these parameters could result in faster and more optimal results. Moreover, adapting the algorithm to function as an embedded application for field testing is also a promising area for further development."

**Manuscript Update:**
Revision:
Lines 402-404

Redline Version:
Lines 458-460

| Reviewer: | Comment Number: |
|---|---|
| 3 | R3-1 |

**Comment:**

1. The authors addressed my question #5 in their Response (See their Reply to RC1, Comment Number 6), which suggested that the proposed method be compared with single-wavelength radar observations and a mu-lambda constraint. The Response states that the proposed method and the mu-lambda constraint method produce similar results. Thus, the proposed method is not better than a single frequency method using a mu-lambda constraint. However, this comparison is not included in the manuscript. The manuscript should include the single frequency mu-lambda constraint method as a benchmark to show that the proposed method produces similar results.

**Reply:**

A comparison with the mu-Lambda constraint is now included with both simulated data and the measured/truthed data. The errors of each are much more clearly demonstrated in the form of a cumulative distribution rather than a time series plot.

Update for simulated data:

"Figure 8 illustrates the cumulative distribution of errors for each method. In this plot, the cumulative portion of errors is shown on the y-axis, while the sorted error values of each method are displayed on the x-axis. As an example interpretation of the plot, 90% of current study's errors (blue) are less than 0.2 in terms of RAE, while 90% of the Tawain Z- R based errors(black dashed) are not contained until an error level of 0.78 RAE. Interpreting the plot from the constant RAE perspective, 65% of the current study's errors are below 0.1, while 60% of $\mu$ - $\Lambda$ (purple) method's errors and 30% of the Tawain Z- R (black) method's errors are below 0.1 RAE. The conclusion of the simulations can be drawn from the plots: the dual frequency approach provides a modest improvement over single frequency retrievals, even with a relevant $\mu$ - $\Lambda$ constraint, and a significant improvement compared to all Z-R based rates. Under these ideal simulated conditions, the median RAE of the dual-frequency approach was 0.0623 while the $\mu$ - $\Lambda$ single-frequency approach was 0.0725. The median RAE of the best performing (at the 50th percentile) Z-R relationship ($Z = 207R^{1.45}$), was 0.1861."

**Figure 8.** The distribution of errors (cumulative) associated with each approach are plotted. The results of this study's proposed methodology using two frequencies (blue circles) shows modest improvement when compared to single frequency retrievals with (purple triangle) and without (red square) assumed $\mu - \Lambda$ constraints. A more significant improvement is seen when compared to the various errors associated with $Z$-$R$ rain rate estimations are indicated with dashed lines.

Evaluation data:
"Figure 13 features a plot of the cumulative distribution of RAE for all three methods. The retrieval algorithm's accuracy is depicted by the blue-circles line, while the $\mu$ - $\Lambda$ constrained retrieval is shown in purple, and the Z-R relationship benchmark is indicated with the dashed black line. This visualization highlights that the proposed method of estimating rainfall rates using retrieved DSD parameters significantly enhances accuracy over a region-specific Z-R relationship (Chang et al., 2021). Specifically, the median RAE for the Z-R method is 0.76, while the retrieval results correspond to a median of 0.53, marking a notable improvement of 30.3 % in this study's context. The dual-frequency approach is comparable to the single-frequency constrained approach and the only claimed relative benefits are not needing to ascertain a regional $\mu$ - $\Lambda$ relationship or cases in which the precipitation deviates from the assumed relationship."

**CDF of RAE for Comparative Approaches**
**Measured Data**

**Figure 13.** The performance of the dual frequency approach (blue circles) shows roughly equivalent performance to the single frequency approach employing a relevant $\mu - \Lambda$ constraint (purple triangles). Both methods utilizing DSD information outperform the region-specific *Z-R* derived rainrates (black dash).

**Manuscript Update:**
Revision:
Lines 287-296 and 366-373

Redline Version:
Lines 314-323 and 421-429

| Reviewer: | Comment Number: |
|---|---|
| 3 | R3-2 |

**Comment:**
2. Comparing the proposed dual wavelength retrieved rain rates to a generic z = 300R^1.4 power law relationship is not a valid comparison. The comparison should be performed using a regionally tuned Z-R relationship produced from the disdrometer data. The manuscript should compare the rain rates estimated from the proposed retrieval method with rain rates estimated from a disdrometer derived regional Z-R relationship.

**Reply:**
We have reevaluated the results using a region-specifc Z-R relationship. Additionally, a more comprehensive simulation section has incorporated three Z-R relationships for comparison.

"Equation 14 represents perhaps the most common meteorological radar relationship Ulbrich and Lee (1999), Equation 15 is specific to Taiwan as proposed by Chang et al. (Chang et al., 2021), and Equation 16 is derived from fitting calculated Z and R values found in the initial steps of the simulation."
Z=300R^1.4        (14)
Z=207R^1.45        (15)
Z=324R^1.35        (16)

Equation 15 (Taiwan specific) is used as the Z-R relationship for the measured data evaluation.

**Manuscript Update:**
Revision:
Lines 280-286

Redline Version:
Lines 307-313

| Reviewer: | Comment Number: |
|---|---|
| 3 | R3-3 |

**Comment:**

3. The simulations shown in Figure 9 are misleading because they are incomplete and do not represent the observed rain systems. For example, the simulation values of N0 = 8000, mu = 2, and lambda = 2 will produce reflectivity factors over 55 dBZ (I am using formulas given in Ulbrich 1983 J. Clim. Appl. Meteor.). The manuscript should either remove the simulations (and Figure 9) or add a more thorough simulation analysis that includes rain systems observed by the radars and the possible correlation between DSD parameters.

**Reply:**

We have updated the manuscript to include a dedicated simulation section which involves performing retrievals on simulated data in order to examine the performance in an ideal setting.

We believe the simulation at the parameter level is still valuable and have thus updated it to include values that are more representative of our data set. The reflectivity factor for the DSDs generated by the combined gamma parameters of the plot should not exceed 45 dBZ.

**Figure 9.** $K_{DP}$ is simulated for DSDs of varying $\mu$ and $\Lambda$ values with fixed $N_0 = 8000$. A separation between the $K_{DP}$ values is clear at the two radar bands.

**Manuscript Update:**

Revision:
Lines 257-301 and Figure 9

Redline Version:
Lines 281-328 and Figure 9

---

## Author Response (AR3)

Re: Drop Size Distribution Retrieval Using Dual-Polarization Radar Observations at C-Band and S-Band

Dear Editorial Team of AMT,

We would like to again extend our sincere gratitude to you and the reviewers who have been very generous with their time and effort in helping us improve our manuscript.

The few suggested minor revisions have been incorporated.

Thank you,

Dan Durbin
Yadong Wang
Pao-Liang Chang

| Comment Number: |
|---|
| 1 |

| Comment: |
|---|
| Figure 4: Please set the y-lim of the differential phase to [-50 - 200] |

| Reply: |
|---|
| We have updated the y-axis limits as suggested:
[Figure]
 Figure 4. The data along the radial indicated in Figure 3 before and after processing. Raw radar data (dashed lines) are contrasted with post-processed data (solid lines). Processed C-band reflectivity is not used as a retrieval input but is useful for validation purposes as discussed in Section 3.1 |

| Manuscript Update: |
|---|
| Revision:
Figure 4

Redline Version:
Figure 4 |

| |
|---|
| **Comment Number:** |
| 2 |
| **Comment:** |
| Figure 12: Please add some quantification metrics to each subplot. |
| **Reply:** |

We have updated Figure 12 to include RMSE, mean bias, and the correlation coefficients. The referenced standard formulas for each have been added to the main body text.

[Figure]

**Figure 12.** The retrieved calculated value of each parameter (y-axis) is shown relative to the algorithm's input (x-axis) based on the radar measurements at the final gate (disdrometer location). The results in terms of MB, CC, and RMSE are included.

| |
|---|
| **Manuscript Update:** |
| Revision: |
| Figure 12 |
| Lines 354-358 |
| |
| Redline Version: |
| Figure 12 |
| Lines 355-360 |

| |
|---|
| **Comment Number:** |
| 3 |
| **Comment:** |
| Conclusions: Make sure to point out that the multifrequency approach brought some modest improvement compared to that of single frequency with mu-lambda constrain [reviewer #3].. |
| **Reply:** |
| We have updated the conclusion to include this statement:

A novel approach to retrieving the DSD using PSO has been discussed. The outlined dual-frequency dual-polarization method can provide significantly improved estimates for rainfall compared to $Z - R$ relationship estimates and offers modest improvement to retrievals utilizing relevant $\mu - \Lambda$ constraints. While the retrieval is unconstrained regarding the gamma distribution parameters, the price is the additional data needed. The authors predict that radar systems utilizing more than one frequency will continue to become more commonplace and producing the required synchronized measurements will be more feasible with teh the adoption of phased array systems. The value of dual-polarization in radars is already universally accepted. Algorithms such as the one prototyped in this work will become more valuable as radar systems produce data with this type of increased diversity. |
| **Manuscript Update:**
Revision:
Lines 386-388

Redline Version:
Lines 387-390 |